# Motion direction is represented as a bimodal probability distribution in the human visual cortex

Andrey Chetverikov [1,2] ✉ & Janneke F. M. Jehee [1] ✉

Humans infer motion direction from noisy sensory signals. We hypothesize that to make these inferences more precise, the visual system computes motion direction not only from velocity but also spatial orientation signals – a 'streak' created by moving objects. We implement this hypothesis in a Bayesian model, which quantifies knowledge with probability distributions, and test its predictions using psychophysics and fMRI. Using a probabilistic pattern-based analysis, we decode probability distributions of motion direction from trial-by-trial activity in the human visual cortex. Corroborating the predictions, the decoded distributions have a bimodal shape, with peaks that predict the direction and magnitude of behavioral errors. Interestingly, we observe similar bimodality in the distribution of the observers' behavioral responses across trials. Together, these results suggest that observers use spatial orientation signals when estimating motion direction. More broadly, our findings indicate that the cortical representation of low-level visual features, such as motion direction, can reflect a combination of several qualitatively distinct signals.

Estimating the direction of motion of an object is arguably one of the most ubiquitous tasks. Whether it is to catch a ball, cross a busy street, or make sure that your toddler does not run into something, the visual system needs to quickly and efficiently parse the retinal input and infer the direction in which things are moving. Yet, this task is also very difficult because of noise. The visual system needs to rapidly infer an object's direction of motion, despite occlusion of motion paths, changes in motion speed and direction, and additional variability in neural signals. As a result of all these sources of variance, estimates of motion direction are necessarily uncertain – any given pattern of neural activity is almost always consistent with multiple interpretations.

Are there ways in which the nervous system could reduce this uncertainty in the interpretation of its sensory signals? One potential strategy for reducing uncertainty could be to use additional visual cues when inferring motion direction. Using multiple cues to obtain a more precise estimate of a visual feature is a well-known strategy observed for slant, depth, and shape perception, among others[1]. Although motion direction is often thought of as a "simple" visual feature, the underlying inference steps need not be simple from a computational

perspective and could involve cues other than velocity. For example, the nervous system could rely on "motion streaks" in its estimates of direction[2] – a "streak" is a smeared representation of a fast-moving object due to the temporal integration of signals along its motion path. From a computational standpoint, combining such spatial orientation signals with the information provided by velocity-tuned neurons would decrease the uncertainty in inferred motion direction (as also illustrated in simulations below). Although the noise in velocity and spatial orientation signals is probably correlated (e.g., due to common retinal factors), integrating the information provided by both sources would still be beneficial as long as the correlation between them is not perfect (see[1,3,4] for similar rationale). While behavioral studies suggest that observers are sensitive to motion streaks (e.g., [2,5,6]), direct neural evidence for streak-based computations in human motion perception is currently lacking.

Here, we investigate the neural and computational basis of human motion perception, using a combination of computational modeling, psychophysics, and functional MRI. To arrive at a set of quantitative predictions, we first implemented a Bayesian observer model that uses

[1]Donders Institute for Brain, Cognition, and Behavior, Radboud University, Nijmegen, The Netherlands. [2]Department of Psychosocial Science, Faculty of Psychology, University of Bergen, Bergen, Norway. ✉e-mail: andrey.chetverikov@uib.no; janneke.jehee@donders.ru.nl

both velocity and spatial orientation signals in its estimates of motion direction. The model quantifies its beliefs about direction of motion as a probability distribution, wherein each direction of motion (interpretation) is assigned a probability of being true. As we will illustrate below, the observer model makes a surprising prediction: when an observer relies on both velocity and orientation signals, their belief should be represented as a bimodal probability distribution, with peaks around the true and opposite motion direction. We tested this prediction using fMRI in combination with a generative model-based decoding technique to extract probability distributions from cortical activity[7]. Interestingly, this revealed that motion direction is represented as a bimodal probability distribution in visual areas V1, V2, V3, and V4, as well as motion-sensitive middle temporal cortex (hMT+). Notably, the shape of the decoded distribution (peak locations and entropy) predicted both the direction and magnitude of the behavioral errors made by the participants. The observer model furthermore predicted a perceptual illusion that we subsequently tested and found support for in a follow-up behavioral study: when sensory information is particularly noisy, participants sometimes perceive the stimulus as if it is moving in the direction opposite to the true direction of motion. Taken together, our findings provide strong evidence that spatial orientation signals are used by the nervous system in its judgments of motion direction and demonstrate complexity in the visual processing of a seemingly simple visual feature.

## Results

### Bayesian observer model

Visual perception is necessarily uncertain. Because of noise and ambiguity, it is impossible to infer with absolute precision the stimulus from the sensory response. Instead, any sensory measurement is consistent with a whole *range* of different interpretations. What strategies could the brain employ to reduce this uncertainty and improve the precision of its sensory estimates?

One well-known strategy to reduce uncertainty is to use additional sources of information. For example, when the observer's task is to determine an object's direction of motion from noisy sensory measurements, combining velocity signals with spatial orientation signals could help to decrease ambiguity[2]. Orientation signals are useful because they convey information about the trajectory of a moving object. That is, if the observer integrates the object's position over time, the orientation of the resulting path (a motion "streak") will be aligned with motion direction. Assuming noise is sufficiently independent between the signals, orientation signals could provide additional information about an object's motion direction and improve the

observers' estimates. Indeed, given known neural response properties in visual areas[8–11], it seems likely that velocity and spatial orientations signals remain largely independent in cortex. Here, we develop a Bayesian observer model that implements this strategy. The model results in a set of concrete predictions that we will test in experiments.

The observer's task is to infer the direction of motion of a stimulus $s$ using two types of signals: velocity and spatial orientation. These signals are corrupted by noise (Fig. 1a). Thus, across trials, the sensory measurements $x_V$ (based on velocity) and $x_O$ (based on orientation) can take different values and are best described by a probability distribution (the generative distribution). For the velocity measurements $x_V$, we assume that the probability distribution of their values, $p(x_V \mid s)$, is a von Mises (circular normal) distribution with variance $\sigma_V^2$. The values of the spatial orientation measurements, $p(x_O \mid s)$, also follow a von Mises distribution but wrapped in the 180° orientation space and with variance $\sigma_O^2$.

To infer the object's direction of motion from the sensory signals, the Bayesian observer uses knowledge of these generative distributions to compute a likelihood function. Given the velocity measurements alone, the likelihood function $L_V(s \mid x_V) = p(x_V \mid s)$. When computed as a function of $s$, the likelihood reflects the range of possible motion directions that are consistent with the velocity measurement $x_V$. The observer similarly computes a likelihood function from the orientation signals. Notably, compared to the velocity measurements, the orientation signals are even more ambiguous with respect to motion direction, because a given orientation is consistent with *two* ranges of opposite motion directions. For example, a snowflake moving left and a snowflake moving right move along the same horizontal motion path. This is why the likelihood function given the orientation measurement is bimodal, with two peaks that indicate opposite motion directions:

$$L_O(s \mid x_O) = \frac{1}{2}p(x_O \mid s) + \frac{1}{2}p(x_O + \pi \mid s) \tag{1}$$

How should the observer make use of all this information so as to determine the object's direction of motion? Both likelihood functions provide information about the stimulus, and the Bayesian observer combines these to arrive at a better estimate. Specifically, under the assumption that the noise is independent, the observer simply multiplies the two likelihoods (see Supplementary Fig. 1 for predictions when noise is correlated). The observer then uses Bayes rule to infer the posterior distribution, $p(s \mid x_O, x_V)$, which describes the range of possible directions of motion given the two measurements. Assuming

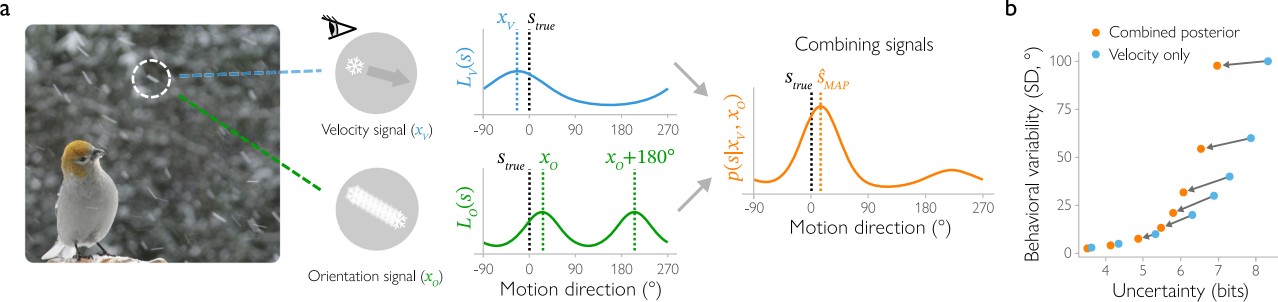

**Fig. 1 | Bayesian observer model combining velocity and orientation signals to determine motion direction. a** The observer (a bird) receives velocity and orientation signals ($x_V$ and $x_O$) elicited by the falling snowflakes. Orientation signals provide cues about the trajectory of the snowflake. Based on each of these noisy measurements, the observer computes the likelihood of motion direction, which is described by the likelihood functions $L_V(s)$ and $L_O(s)$. Notably, the orientation cue is consistent with two opposite motion directions (i.e., a snowflake moving left and one moving right have the same trajectory), and therefore has a bimodal likelihood function. The observer combines the two likelihood functions to compute the

posterior distribution $p(s \mid x_V, x_O)$, which describes how probable different motion directions are given the combination of cues. The observer selects the most likely stimulus (maximum a posteriori, $\hat{s}_{MAP}$) from the posterior distribution as their final estimate of motion direction. Photo courtesy of Bill Geisler. **b** Compared to an observer who uses velocity measurements only (blue), combining orientation and velocity cues (orange) reduces both uncertainty and behavioral variability. The entropy (uncertainty) of the velocity likelihood is the same for each pair of dots connected with an arrow, but overall uncertainty and behavioral variability are not.

that the prior $p(s)$ is flat, $p(s|x_O, x_V) \propto L_V(s|x_V)L_O(s|x_O)$. While the prior is likely not flat for orientation signals[12], this assumption does not affect any of our conclusions. The peak of the posterior function is the most likely direction of motion, and we assume that this is the observer's decision (maximum a posterior estimate, $\hat{s}_{MAP}$; please see Methods, Bayesian observer models, for a discussion of other readout strategies). We can take the entropy of the distribution (the Shannon information for a given probability distribution; see Methods) as a measure of the degree of uncertainty in this estimate. Because the sensory measurements vary from trial to trial, so do the observer's estimates of the direction of motion. Across trials, this results in a distribution, $p(\hat{s}_{MAP}|s)$, which we will call the behavioral response distribution. As we will demonstrate in simulations, uncertainty is reduced when combining the two sources of information. For behavior, the integration of velocity and orientation signals is beneficial as well, but will also lead to a surprising bimodal pattern of responses under some conditions that we discuss below.

We simulated the trial-by-trial decisions of the Bayesian observer when presented with noisy stimuli. Specifically, we used different levels of noise in the observer's measurements to illustrate the predictions. We computed the posterior distribution, quantified uncertainty, and obtained the observer's behavioral responses. In what follows, we start with the description of the simulations in the low-stimulus-noise regime, which corresponds to our fMRI study design. Later, we will describe the results of the simulations in a high-stimulus-noise regime, which matches the design of a follow-up behavioral experiment.

First, we show that combining velocity and orientation signals is indeed beneficial for observers. To this end, we compare two different model observers. The first observer infers motion direction from velocity signals only, while the second one uses both orientation and velocity signals. We analyzed the relationship between behavioral variability and uncertainty for each model observer, with the velocity and orientation standard deviation ($\sigma_V$ and $\sigma_O$) parameters spanning the range from 3° to 100° to ensure that our predictions hold for different parameter settings. We found that, in general, behavioral variability increases when uncertainty increases (Fig. 1b). In addition, the posterior distribution computed by the velocity-only observer always indicates greater levels of uncertainty than the posterior obtained by the observer who combines velocity and orientation likelihoods. This is because the orientation signals provide additional information as to which motion directions are likely. This reduction in uncertainty also results in improved behavior. That is, the more information (the less uncertainty) there is, the better the observer's estimates of motion direction. Thus, behavioral variability decreases when both velocity and orientation signals are used. Overall, the simulations demonstrate that combining the velocity and spatial orientation signals is beneficial for the observer's behavior: it decreases uncertainty, which reduces the variability of behavioral responses.

A second prediction of the model is that the posterior distribution could become bimodal when the orientation likelihood is combined with a sufficiently uncertain velocity likelihood. In other words, the inference process results in a posterior distribution that has two peaks (see Fig. 1c). In our simulations, we varied uncertainty by manipulating the amount of noise in the velocity measurements ($\sigma_V > 30°$ for our set of simulations; see Methods for details). When velocity signals indicate high uncertainty (so the likelihood function $L_V(s|x_V)$ is wide), the orientation signal dominates in the posterior, resulting in a bimodal distribution. At the extreme, when the velocity signals provide no information at all (the velocity likelihood function is flat), the posterior becomes fully proportional to a bimodal orientation likelihood function (Supplementary Fig. 2). Of course, this is an extreme scenario in which the internal representation strongly deviates from the physical stimulus (for example, a low coherence stimulus combined with strong attentional effects on motion streaks might create a representation

that is dominated by orientation signals). But even in less extreme noise scenarios, when velocity signals do provide some imprecise clues to motion direction (i.e., the velocity likelihood function is wide, but not flat), bimodality stemming from the presence of orientation signals is still visible, because the ambiguity of the orientation signal cannot be fully resolved by the velocity signals. This bimodal shape is exclusively tied to the presence of orientation signals, as the velocity-only observer always arrives at unimodal posteriors, even with increased velocity uncertainty. Notably, bimodal posteriors can also be observed when stimulus noise is low but there is additional noise from other sources, such as fMRI noise (see details in Methods). That is, the noise incurred by fMRI measurements further increases the uncertainty associated with each of the two cues (Supplementary Fig. 3) and can result in bimodality at the level of voxels – even when at the neural level the posterior is unimodal. To account for this in our predictions, we included fMRI noise in our simulations of this experiment (see Eq. 30). We verified that our predictions are qualitatively the same when the posterior distribution is estimated directly from voxel population activity (see Supplementary Methods). The bimodality in the observed posterior is particularly evident when averaging distributions across trials: When orientation signals are present, the resulting mean posterior distribution is bimodal with peaks around the true and opposite motion directions (Fig. 2a).

The bimodality in the shape of the posterior can be observed not only when averaged across trials, but also on a trial-by-trial basis for the observer who uses both sources of information. The exact shape of the posterior varies from trial to trial and can be quantified by fitting a mixture of two von Mises components (basis functions) to the posterior, in line with its analytic description (see Methods). Parameters of the von Mises basis functions provide a convenient quantification of the posterior shape and allow us to trace the bimodality in the posterior at the single-trial level. Specifically, we analyzed the location of the peaks of the two components and plotted for each possible combination of locations the probability of observing a trial with this particular combination (i.e., the joint probability distribution of the component locations, Fig. 2b). Three clusters of trials are readily observed when looking at this plot. The first large cluster has two components co-located around the true motion direction. This combination of components corresponds to a unimodal posterior. In the second large cluster, the larger component is located around the true direction and a smaller one is located around the opposite direction of motion. This cluster is important for us, as it describes a bimodal posterior with a larger peak around the true direction. Finally, a small third cluster also corresponds to a bimodal posterior, but in this case the velocity measurement just happened to be closer to the opposite motion direction because of random noise. Accordingly, the larger component is located around the opposite direction, and the smaller fitted peak lies closer to the true direction of motion. This cluster of trials also signifies that orientation signals are present, but it might be difficult to detect in our empirical analyses, because it comprises only a small number of trials (about 0.3% in our simulations). In sum, when the observer computes a posterior distribution from two likelihoods, one obtained from velocity and the other from spatial orientation signals, then we should observe a large cluster of bimodal trials.

For the third prediction, we turned to the observer's behavioral errors. We found that the direction and magnitude of the observer's behavioral errors should be correlated with the location of either of the two peaks in a bimodal posterior. Fig. 2c shows the error in the observer's simulated behavioral estimates of direction of motion (i.e., the difference between the MAP estimate and the true motion direction) as a function of the location of each peak in the posterior distribution. For the observer who uses both sources of information, there is a positive relationship between either peak location and the direction and magnitude of the behavioral error (Fig. 2c, left). That is, if the first peak is shifted clockwise relative to the true direction of

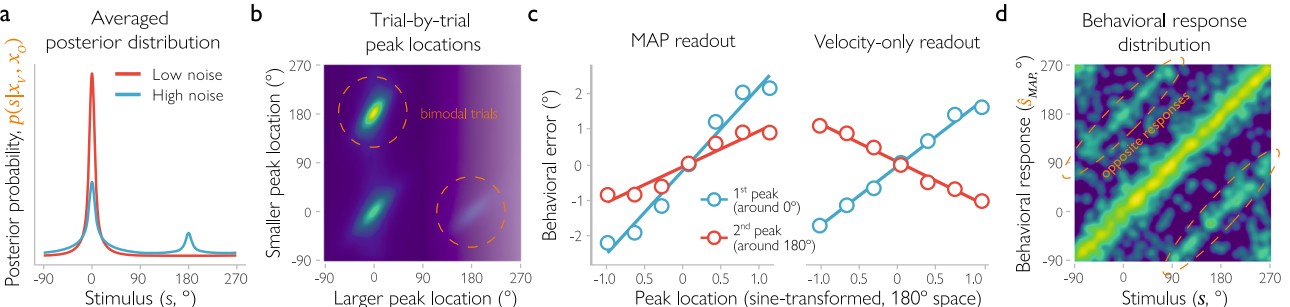

**Fig. 2 | Predictions of the Bayesian observer model. a** For low uncertainty (entropy) levels of the velocity likelihood ($\sigma_V \leq 30°$), the posterior distribution (averaged across trials) is unimodal, centered at the presented stimulus (at 0°). However, for high uncertainty levels ($\sigma_V > 30°$), it becomes bimodal with a second peak at the opposite motion direction (at 180°). **b** The trial-by-trial posterior distribution can be described as a weighted combination of two von Mises components. At high noise levels, the joint distribution of the locations of these components has three clusters (brighter colors indicate a higher probability of observing a certain combination of component locations). The largest cluster consists of bimodal trials with a larger component located around the presented motion direction (0°) and a smaller one around the opposite direction (180°). A smaller cluster contains unimodal trials (both components around the presented direction). The third cluster corresponds to trials with the larger component located around the opposite and the smaller around the presented direction. Noise levels here and in the next panel were based on empirically observed distributions

under the assumption of additive noise in the MRI measurements, see Methods. **c** For the Bayesian observer model inferring motion direction from both velocity and spatial orientation signals ("MAP readout"), the location of either of the two peaks correlates positively with the direction and magnitude of errors in the observer's behavioral response. When the first peak is shifted clockwise relative to the true direction of motion (or the second peak relative to the opposite direction), the observed response should also be shifted clockwise relative to the true direction. In contrast, for an observer with a bimodal probabilistic representation of motion direction but who uses only velocity and not spatial orientation signals ("velocity-only readout"), the correlation between the second peak location and behavioral errors has the opposite sign. **d** In a high-uncertainty regime, the distribution of behavioral responses for the Bayesian observer becomes bimodal as well (brighter colors indicate a higher probability of a given behavioral response for a given stimulus). That is, the Bayesian observer sometimes mistakes the presented motion direction for the opposite one.

motion (or the second peak is shifted clockwise relative to the opposite direction), the observed response is also shifted clockwise relative to the true direction. In contrast, for an observer whose representations are bimodal, but who nonetheless uses only velocity (and not orientation) signals, the relationship between the second peak location and behavioral errors has the opposite sign (Fig. 2c, right). This inverse relationship arises because the second orientation peak is "pulled" towards the velocity peak in the posterior. This gives rise to a negative circular correlation between the location of the second peak and the velocity estimate: when the velocity estimate shifts clockwise relative to the true stimulus, the second peak shifts towards it, counterclockwise. When the observer's behavioral response is based on velocity signals alone, the behavioral response also has this inverse relationship with the second peak of the integrated posterior. Thus, the observed relationship between the second peak location and behavioral errors should enable us to determine whether or not orientation signals are used in the observer's estimates of direction of motion; in other words, whether the bimodality in brain signals is behaviorally relevant.

Finally, we turned to a high-noise regime, in which the level of noise associated with the observer's measurements is high (note that this refers to noise in the neural signals, and not the additional noise due to fMRI recordings). This noise regime matches that of our follow-up behavioral experiment. The model predicts that under these conditions, the behavioral response distribution should become bimodal. Specifically, the model shows that if the observer uses both sources of information, then bimodality should become more pronounced under high levels of noise. Fig. 2d shows the expected distribution of behavioral responses across trials for this condition; that is, the probability of an estimated motion direction given the true direction of motion. Interestingly, we observed a striking bimodal response distribution across trials. In other words, the observer cannot always reliably tell apart the true and the opposite motion direction in the posterior distribution, and sometimes mistakes the opposite direction for the true one. Altogether, this means that if the task becomes sufficiently difficult, we should sometimes observe a behavioral response that matches the stimulus in its orientation but not its direction of motion. We additionally analyzed the relationship between the shape of the

posterior distribution and the observer's behavioral responses, but found that it would not allow us to further adjudicate between the models (Supplementary Fig. 4). For this reason, we focus exclusively on the behavioral response distribution here, as these data are most informative.

In sum, when the observer uses velocity and spatial orientation signals, the model makes the following predictions:

1. Uncertainty (posterior entropy) should be linked to behavioral variability.
2. The posterior distribution should be of bimodal shape (with the peaks at the true and the opposite motion direction), both when averaged across trials and for a large portion of trials in trial-by-trial analyses.
3. Both peaks of the posterior should predict behavioral responses; that is, the direction and the magnitude of the observer's errors.
4. High levels of uncertainty should result in a bimodal response distribution in behavior.

With these predictions in hand, we now turn to the experimental data to see if they hold true.

### fMRI study

How do human observers represent and estimate motion direction? We ran an fMRI study and a follow-up behavioral experiment to address this question. In the fMRI study, we used random-dot kinematograms with 100% coherent motion. Because of relatively low levels of neural variability, we predicted that, behaviorally, this would correspond to the low-noise motion regime of the simulations. Participants viewed dots moving in a single direction in an annular window for 1.5 seconds. After a brief delay period, they reported the direction of motion of the stimulus by rotating a bar presented at fixation. Under these conditions, the task was relatively easy for the participants, whose behavior showed a mean absolute error of $M = 6.18°$, 95% $CI =$ [5.72°, 6.62°]. For each trial of fMRI data, we decoded a probability distribution over motion direction from patterns of BOLD activity in visual areas V1, V2, V3, hV4 and hMT+ combined, using a probabilistic decoding technique[7,17]. We took the most likely value of the decoded distribution as the decoded direction of motion, and its entropy

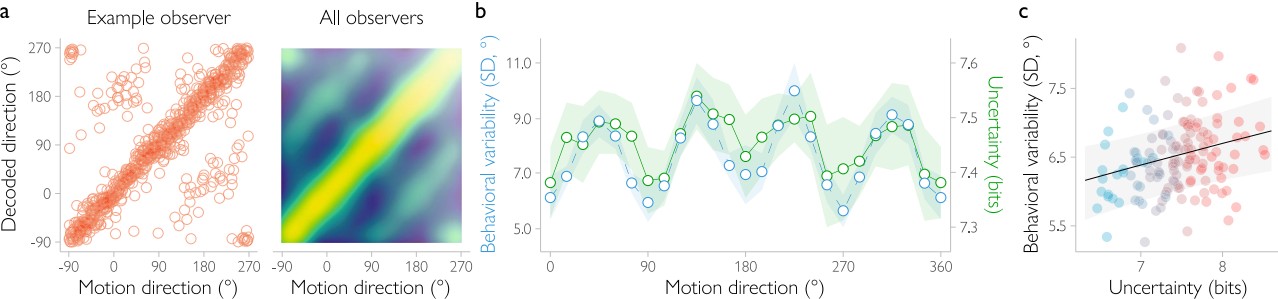

**Fig. 3 | Direction decoding accuracy and decoded uncertainty in the main fMRI experiment with low-noise (100% coherence) stimuli. a** Decoded motion directions for an example observer (left) and a joint probability distribution of decoded and presented motion directions across all observers (right). For the example observer, each dot is a single trial. The dots along the main diagonal correspond to decoded directions that are relatively accurate, while dots along the lines parallel to the main diagonal with an offset of ±180° degrees show trials on which the decoded direction is opposite to the true direction. When aggregated across observers, each pixel shows the probability of observing a certain decoded direction for a given presented motion direction. Most of the trials are decoded accurately ($r = 0.72$), but there is a noticeable increase in the probabilities of opposite directions of motion. **b** Both behavioral variability (in blue) and decoded uncertainty (in green) increase

as a function of distance from cardinal motion directions (trials split into 15° bins for illustrative purposes, $b = 9.17$, 95% HPDI = [4.73, 13.65], $BF = 8.54 \times 10^5$). Circles denote mean across observers. Shaded regions show 95% confidence intervals. **c** Behavioral variability on a trial-by-trial basis increases as a function of decoded uncertainty after controlling for motion direction ($b = 5.93$, 95% HPDI = [1.98, 9.96], $BF = 157$, with the effect of motion direction on behavioral variability included as a regressor in the multiple regression model). Dots show the variability across trials computed for eight equal-sized bins (indicated by color) and for each individual observer (data split into bins for illustrative purposes only). The line shows the effect of uncertainty estimated with a hierarchical Bayesian regression (without binning) with 95% HPDI range indicated by the gray area.

reflected the degree of decoded uncertainty. Please note that the reliability of the two cues likely changes with stimulus parameters such as motion speed and dot size, so our results likely depend on the specific dot size and chosen speed of 7 deg/s.

To benchmark the decoding approach, we first tested the degree to which the decoded direction of motion matched the true motion direction of the stimulus (Fig. 3a). The decoded and true directions were significantly correlated, with a mean circular correlation coefficient across participants of $r = 0.72$, 95% $CI = [0.62, 0.80]$, $BF = 2.82 \times 10^5$ (Bayes factors above 3.2 are usually taken to indicate substantial evidence, above 10 – strong, and above 100 – decisive evidence[13]). We then tested the decoder's assumptions about the covariance structure of the data. To the extent that the model assumptions match the true generative structure of the data, the decoded uncertainty should be correlated with the magnitude of the error in the decoded direction of motion. Indeed, we found this to be the case ($BF > 10^{170}$). Together, these findings indicate that the decoder provides a reasonable estimate of the aggregated sources of uncertainty in the data.

To test the degree to which the algorithm also captured neural sources of uncertainty (as opposed to imprecision due to the fMRI measurements), we then turned to behavior. Specifically, we reasoned that a more precise neural representation in cortex should result in more precise behavior (as also quantified by our simulations, see Fig. 1b). To test this relationship and benchmark the degree to which the decoding technique was able to catch neural sources of uncertainty in particular, we first investigated the link between decoded uncertainty and behavioral variability across motion directions, using Bayesian hierarchical regression to estimate the within-subject effect of uncertainty on behavioral variability while accounting for individual differences between participants (Fig. 3b). The judgments of motion direction of our participants showed a classical oblique effect[14–16]: observer responses were more variable at oblique compared to cardinal directions. Specifically, behavioral variability increased from 6.18° at cardinal to 9.42° at oblique directions ($BF = 2.1 \times 10^{15}$). Importantly, decoded uncertainty also increased from cardinal to oblique directions of motion (entropy changed from 7.40 bits for cardinal to 7.49 bits for oblique, $BF = 1.6 \times 10^{13}$), and this change in entropy across directions was significantly correlated with behavioral variability, $b = 9.17$, 95% HPDI = [4.73, 13.65], $BF = 8.54 \times 10^5$ ($b$ is the regression coefficient). This indicates that across motion directions, the decoded

distributions reflect the precision of the information contained in underlying neural activity.

Even when the stimulus is held constant, uncertainty varies on a trial-by-trial basis due to random fluctuations in cortical activity. Is there a relationship between uncertainty and behavior when between-stimulus variability is accounted for? We tested if our approach captures trial-by-trial fluctuations in the fidelity of the cortical representation by quantifying the effect of uncertainty on behavioral variability in a hierarchical Bayesian regression model that accounted for the oblique effect in addition to between-subject differences. Decoded uncertainty again reliably predicted behavioral variability ($b = 5.93$, 95% HPDI = [1.98, 9.96], $BF = 157$). Control analyses showed that these results cannot be explained by mean BOLD amplitude, head motion, gaze fixation position, and variability of gaze fixation positions (Supplementary Fig. 5). This suggests that our decoder captures not only between-stimulus uncertainty, but also fluctuations in the quality of the underlying cortical representation on a trial-by-trial basis (Fig. 3c).

Having established that the decoded distributions are meaningful and capture the degree of uncertainty in the underlying cortical representation, we then turned to the shape of the distribution. To what degree do the decoded distributions show evidence of an advantageous estimation process in which velocity and orientation signals are combined for estimates of motion direction? Our simulations suggest that for such an advantageous decision process, the mean decoded posterior across trials should be bimodal, with a larger peak on the true direction and a smaller peak at the opposite direction of motion (Fig. 2a). Indeed, this is what we found when we analyzed the shape of the decoded posterior (Fig. 4a). When averaged across trials, the decoded distributions had peaks located around the true ($M = -0.01°$, 95% $CI = [-0.13, 0.10]$) and opposite ($M = 178.69°$, 95% $CI = [177.04, 180.00]$) motion directions for 16 out of 18 participants (for the remaining two participants, the average posterior was unimodal and located at the true direction of motion). This similarly held for individual visual areas, including hMT+ (Supplementary Fig. 6). That is, on average, the decoded posterior is bimodal, matching our predictions.

Our decoding approach further allowed us to more formally test for the presence of bimodality and demonstrate that the same pattern is observed on a trial-by-trial basis. This is important because, when evaluated across trials, the bimodality in the averaged decoded

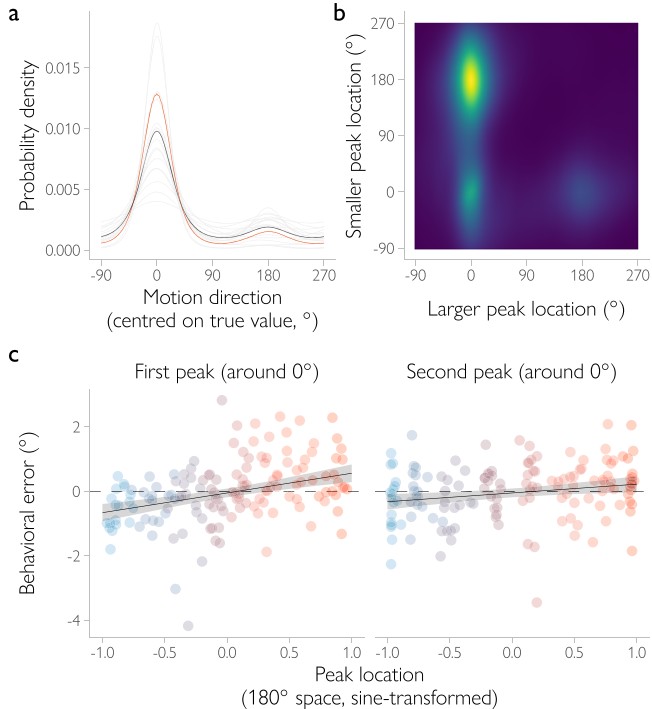

**Fig. 4 | Bimodality in the posterior distribution decoded from fMRI signals. a** When averaged across trials, the decoded posterior distribution is visibly bimodal for the majority of observers (gray lines show the across-trials average for each observer, black line shows the mean posterior across observers, the example observer from Fig. 3A is highlighted in orange). **b** Joint distribution of trial-by-trial peak locations aggregated across observers. Brighter colors correspond to higher probabilities of observing a certain combination of the larger and smaller peak location. A large fraction of trials (52%) has the larger peak located around the presented motion direction (0° on the abscissa) and the smaller peak around the opposite direction (180° on the ordinate). **c** Focusing on bimodal trials only (i.e., the peaks of the decoded distribution are located around the presented (between −90° and 90°) and opposite motion directions (between 90° and 270°)), the location of either peak is correlated with behavioral errors ("first" peak, located closer to the true direction of motion: $b = 0.63$, 95% HPDI = [0.40, 0.88], $BF = 2.46 \times 10^5$; "second" peak, located closer to the opposite one: $b = 0.30$, 95% HPDI = [0.10, 0.50], $BF = 19.05$). For each observer, data were divided into eight equal-sized bins (illustrated by color). Dots show the mean behavioral error across all trials in each of the bins. Please note that data were binned for illustrative purposes only. The lines show the estimated effect (i.e., the regression coefficient) of peak location on behavioral error based on a hierarchical Bayesian model, with the 95% credible intervals in gray.

posterior might result from an aggregation of unimodal trials with peaks concentrating either at the true or at the opposite direction. To address this concern, we quantified the shape of the posterior for each individual trial as a mixture of two von Mises components (basis functions) and estimated the location of each component (i.e., its mean). Fig. 4b shows how these locations are distributed across trials (their joint distribution). This pattern of results is qualitatively very similar to what is predicted when the posterior reflects both velocity and spatial orientation signals, and rather distinct from when it would be based on orientation or velocity signals alone (Supplementary Fig. 7). To quantify these results, we estimated the number of clusters in the observed pattern of location combinations. We modeled the joint distribution of the peak locations with models that assumed one to three independent clusters of peaks (see Methods). For example, a model with a single cluster assumed that the peak locations on all trials are similar (belong to a single bivariate von Mises distribution). The best fit was provided by a two-cluster model (ΔWAIC against the single-cluster model = 9741, $BF > 10^{20}$; adding a third cluster did not

significantly improve the fit, ΔWAIC = −0.2) with the first cluster corresponding to unimodal trials (the two peaks overlap and are located at the true direction) and a second cluster corresponding to bimodal trials (a larger peak at the true and a smaller peak at the opposite direction). This bimodality in the posterior distribution was observed for a significant portion of the trials (52% were allocated to the second cluster). These results match the model predictions (Fig. 2b) and show that the bimodality in the decoded posterior is not an artifact of aggregation, but rather is a property of the single-trial posterior. This provides further support for our theoretical predictions: the presence of orientation signals leads to a bimodal posterior distribution both on average and in trial-by-trial analyses.

Next, we turned to the third prediction of the Bayesian observer model: if the two peaks of the decoded distribution are relevant for behavior (which would suggest that the observer uses spatial orientation signals to estimate direction of motion), then there should be a relationship between peak location and the direction and magnitude of the error in the participant's behavioral response. More specifically, the relationship should be positive for either peak of the distribution. Thus, when a first peak is shifted clockwise relative to the true direction of motion (or a second peak is shifted clockwise relative to the opposite direction), the observed response should also be shifted clockwise relative to the true direction. Testing this prediction revealed that the location of either peak is indeed positively and reliably correlated with the behavioral errors of our participants (the peak located closer to the true direction: $b = 0.63$, 95% HPDI = [0.40, 0.88], $BF = 2.46 \times 10^5$; the peak located closer to the opposite one: $b = 0.30$, 95% HPDI = [0.10, 0.50], $BF = 19.05$; Fig. 3c). Control analyses showed that these results cannot be explained by the direction in which saccades are made (Supplementary Fig. 8). Thus, the decoded direction of motion (i.e., the location of the first peak) reliably predicts the trial-by-trial behavioral responses of our participants. Crucially, also the second peak of the decoded distribution appears to be behaviorally relevant, providing further support for the hypothesis that human observers use orientation signals when they are estimating direction of motion.

## Follow-up psychophysical experiment

We then tested the final prediction of our model: when uncertainty is high, also the behavioral estimates of the participants should follow a bimodal distribution, much like their internal representation. In a follow-up behavioral experiment, we increased levels of uncertainty using stimuli in which only 18% of dots were moving in a single direction while all other dots moved in random directions. The conditions were otherwise identical to the fMRI study and the participants performed the same task. We first confirmed that our manipulation increased uncertainty. Indeed, participants performed on average much worse than in the main study (a mean absolute error of $M = 41.02°$, 95% $CI = [33.42°, 49.44°]$ in this experiment against $M = 6.18°$, 95% $CI = [5.72°, 6.62°]$ in the main study, Supplementary Fig. 9; but note that the mean error estimates are less informative for this experiment because of the shape of the response distribution, as we discuss next). Further analysis of the behavioral data revealed clear bimodality in the behavioral estimates of our participants. Their responses were clustered around not only the true direction (the main diagonal in Fig. 5a, b), but also the opposite direction of motion (the dashed lines parallel to the main diagonal in Fig. 5a, b). To quantify the degree of bimodality in the behavioral responses, we fitted and then compared two models equivalent to the descriptive models we used in the analysis of the decoded posterior shape: 1) a unimodal (von Mises) distribution centered on the true direction, and 2) a bimodal distribution mixture (two von Mises) with peaks centered at the true and the opposite direction. Both models additionally included a uniform component to account for random guesses. The models were first fitted to each observer's responses separately, and the results were

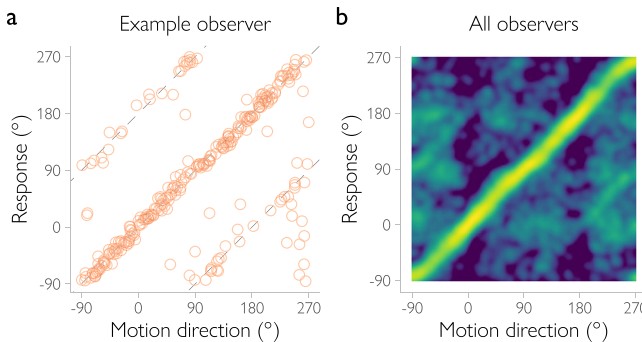

**Fig. 5 | Follow-up behavioral experiment with decreased motion coherence in the presented stimulus. a** Estimates of motion direction for an example human observer (each dot represents a single trial). While the observer is mostly accurate (most dots lie along the diagonal), they also show a tendency to perceive dots as if they were moving in the opposite direction (dots along the dashed lines). **b** Joint probability distribution of the presented motion direction and the observer's judgment, aggregated across human observers. The overall pattern of responses is the same as for the example observer: most responses match the presented directions, but there is a noticeable increase in probability density along the lines ±180° off the diagonal (ΔBIC = 322 for a model comparison between a unimodal (von Mises) distribution with responses centered on the true direction, and a bimodal distribution mixture (two von Mises) with peaks centered at the true and the opposite direction).

then combined across participants. The bimodal model provided a significantly better fit than a model consisting of a single peak (group ΔBIC = 322). This indicates that across participants, the distribution of behavioral responses is indeed bimodal with peaks centered at the true and opposite motion direction. That is, for higher levels of uncertainty, the participants reported seeing dots moving in a direction *opposite* from the true direction of motion. Altogether, this shows that human participants use spatial orientation signals when estimating motion direction, leading to a surprising bimodality not only in the representations decoded from early visual cortex, but also in behavior.

## Discussion

What neural computations enable human observers to infer the direction in which an object is moving? Here, we argue that the nervous system uses not only velocity but also spatial orientation signals to estimate motion direction[2]. We implemented this hypothesis in a Bayesian observer model and tested its predictions using a combination of psychophysics and fMRI. Using a generative model-based fMRI analysis technique[7,17], we decoded probability distributions of motion direction from activity in areas V1-V4 and hMT+. Corroborating the predictions of the Bayesian observer model, we discovered that the decoded distribution of motion has a bimodal shape. Moreover, the two peaks of the distribution predicted the magnitude and direction of the participants' behavioral errors. In a follow-up behavioral experiment, we furthermore showed that this bimodal shape is also observed in the distribution of the participants' behavioral responses when analyzed across trials, indicating that the participants sometimes reported the direction of motion opposite to the true one, as predicted by the Bayesian model. Altogether, this suggests that the nervous system uses not only velocity signals, as assumed by dominant models of motion perception (e.g.,[18]), but also spatial orientation cues to infer motion direction.

It is interesting that the decoded direction of motion predicted not only the presented direction of dot motion with relatively high accuracy across the 360-degree motion feature space, but also the subjective judgments of the observer, irrespective of the presented stimulus. That is, the direction decoded from cortical activity predicted the participants' behavioral errors on a trial-by-trial basis. This suggests that the decoded representations are behaviorally relevant

and not an artifact of the fMRI measurements. This relationship between cortical activity and behavior is reminiscent of previous neurophysiological results showing a correlation between neural activity and behavioral choices ("choice probability"[19–23]) – a link that is often taken to indicate that the signals are used by the animal to determine its decision. Our ability to predict the participant's judgments surpasses that of previous fMRI findings, and is likely due to our use of the TAFKAP decoder, which improves decoding precision by estimating not only voxel tuning properties, but also the voxel (co) variance induced by the fMRI measurements and neural variability[7,24].

Previous work has shown that the degree of imprecision in the cortical representation of orientation[17,25,26] and location[27] can reliably be decoded from fMRI activity patterns. The current study extends these earlier findings by showing that also the fidelity of the cortical representation of motion can be successfully characterized with fMRI. That is, the decoder's trial-by-trial estimates of uncertainty predicted behavioral variability – a measure of perceptual imprecision. This illustrates the versatility of the probabilistic decoding approach and suggests that the neural code for uncertainty may be similar across different visual features, such as motion, location and orientation.

As integrating fully correlated signals would make little sense from a computational standpoint (no additional information would be gained from doing so), our Bayesian model assumes that orientation and velocity signals are conditionally independent. While full independence is likely too strong an assumption for visual cortical neurons (e.g., due to common retinal input), it seems nonetheless likely that a good fraction of the noise added by post-retinal stages of processing will not be shared – for example, because orientation and velocity signals are processed in segregated visual pathways[9,11] and by neurons with different spatiotemporal receptive fields[28–30]. Computationally, as long as some of the noise is independent, the estimate will be better when signals are combined. Indeed, our results suggest that human participants use such integration strategies, even when the cues are likely partially correlated.

It is well known that fMRI signals reflect many forms of (correlated) noise, in addition to the correlated sources of noise in the orientation and velocity signals. For example, the amount of noise in the orientation and velocity signals could fluctuate jointly due to factors such as attention or alertness. Voxel responses could also be correlated due to non-neural sources of noise that are associated with the fMRI measurements themselves, such as participant head motion. However, none of these correlations can explain the observed relationship between the locations of the peaks in the decoded posterior and the direction and magnitude of behavioral errors (Supplementary Figs. 10 and 11). That is, for both scenarios, the predicted correlation between the second peak location and behavioral errors would be negative, much like the pattern shown in Fig. 2c ("velocity-only readout"). This is opposite to what we find in the data, further supporting the conclusion that orientation signals are used by the observers in their direction judgments.

Previous behavioral work has also suggested that the nervous system might use spatial orientation signals to determine motion direction[2,5,6,31–35]. For example, prolonged exposure to moving dot stimuli creates aftereffects similar to the effects produced by static gratings[33], and removing information about streaks increases thresholds for motion detection[5]. Other studies have highlighted potential neural mechanisms that could give rise to motion streak sensitivity[9–11,29,36–40] or found preliminary evidence to suggest that streak-based signals are represented in the human visual cortex[41]. Our work extends these previous findings in several ways. Our normative model explains why observers should use both velocity and spatial orientation cues, as integrating these signals reduces uncertainty and improves direction estimates. The model furthermore made a number of quantitative and qualitative predictions that we tested in experiments. This revealed that motion direction is represented in cortex as a

bimodal probability distribution – a level of complexity that stands in sharp contrast to previously observed probabilistic representations, such as those for orientation and location[17,25–27]. No less important, we discovered that the shape of the bimodal distribution is linked to the participants' behavioral estimates in various ways. Altogether, this suggests that the human visual system uses spatial orientation signals for determining direction of motion and reveals the hidden complexity of probabilistic feature representations in cortex.

We also considered several alternative strategies to judging direction of motion, in addition to the Bayesian observer and velocity-only models (Supplementary Fig. 12). The first alternative model assumed that the observer only uses velocity signals to infer motion direction, with an arbitrary constant bias away from the velocity-based estimate. However, this strategy cannot capture bimodality in the decoded posterior distribution, nor does it explain the bimodal behavioral response distribution as observed here. The second model assumed that observers combine spatial orientation and velocity signals while ignoring uncertainty. That is, the response is a weighted average of the velocity-based and orientation-based estimates, where the weights are assigned randomly. While this model does capture the bimodal behavioral response distribution for high levels of orientation noise, it wrongfully predicts a very wide behavioral response distribution when the precision of velocity and orientation signals is, respectively, high and low. This is inconsistent with behavioral data from previous studies showing that observers perform relatively well at slow motion speeds when orientation information is presumably very noisy or even absent (e.g., [5]). The third model assumed that observers randomly switch between the orientation and motion likelihoods when making the decision[42]. This model predicts that bimodality is always observed, regardless of the degree of uncertainty. Critically, this is not what we observe in our data, where bimodality in the behavioral response distribution clearly depends on stimulus reliability. Finally, we considered a strategy based on the motion aftereffect. Here, the hypothesis is that observers experience and report aftereffects after viewing the stimulus, which results in behavioral responses that are opposite to the true direction of motion. However, such a model would predict stronger aftereffects with greater motion coherence in the stimulus (so lower uncertainty), because the strength of aftereffects is positively related to signal strength (e.g., [43,44]). Therefore, greater bimodality in behavior is expected with greater certainty, which is opposite to our results. In sum, none of the alternative models considered can explain the full scope of our findings.

While our data suggest that observers combine velocity and orientation cues when inferring motion direction, we do not argue that these cues are necessarily integrated *optimally* – that is, that each estimate is perfectly weighted by its uncertainty. We believe it will be difficult, if not impossible, to argue and test for optimality in this particular situation. The main reason for this is that it will be difficult to infer the likelihood for the spatial orientation and velocity-based signals alone. That is, in a typical cue integration experiment, the likelihood of each cue is manipulated by the experimenter and therefore (roughly) known. This makes it possible to predict what the behavioral response should be for both the optimal integration strategy and alternative strategies that ignore uncertainty. For the integration problem considered here, however, the likelihoods are not a priori known to the experimenter, and would have to be inferred from brain data. An fMRI voxel, however, reflects the aggregate response of many neural populations, where the responses from the individual populations are unknown. Without knowledge of the individual signals for spatial orientation and velocity, their likelihoods cannot be calculated, which makes it impossible to predict and compare between the Bayesian and alternative integration strategies.

What neural mechanisms might underlie the observed bimodal distribution in visual cortex? Studies in non-human primates[9,10,29,36,37],

mice[8] and cats[10] have shown that many orientation-tuned neurons in primary visual cortex respond to dots moving parallel to their spatial orientation receptive field. It seems likely that similar neural response properties could have led to the bimodal posterior distribution observed here. Interestingly, also many direction-selective neurons in V1 are tuned somewhat bimodally, with strong responses to one motion direction and a weaker response to the opposite direction[9,10]. To address whether these tuning properties could similarly give rise to a bimodal posterior distribution, we simulated neural population activity using a realistic range of direction selectivities (see Supplementary Fig. 13). We found that the posterior distribution decoded from the obtained population response is always unimodal and never bimodal. This strengthens the hypothesis that the empirically observed posteriors reflect the combined responses of direction-selective neurons and orientation-tuned cells whose spatial orientation receptive field runs parallel to the presented motion direction.

Interestingly, we observed bimodal posterior distributions throughout visual cortex (i.e., areas V1, V2, V3, hV4, and hMT+, see Supplementary Fig. 6), with no clear differences between areas. It is important to keep in mind, however, that the signal-to-noise ratio is likely not constant across these regions, which makes it difficult to draw firm conclusions from this finding. The signal-to-noise ratio also makes it difficult to test which cortical areas integrate the velocity and spatial orientation signals, as a unimodal posterior at the level of voxels does not necessarily imply unimodality in the underlying populations (Supplementary Fig. 14). Notwithstanding these considerations, it does seem likely that all the areas investigated here should show at least some degree of bimodality, as they all contain orientation and velocity sensitive neurons (e.g., [11,28]).

Our stimulus consisted of dots moving at 7 degrees visual angle per second. Interestingly, this speed roughly matches that of optic flow in the natural environment. That is, for a person with eyes 1.5 m above the ground who is walking at 1.4 m/s (the average walking speed), the optic flow in the ground plane will be 7 deg/s at 5 m to the left and right. This suggests that the observed motion streak signals may be highly relevant for the encoding of optic flow and other forms of real-world motion. To capture more complex real-world scenarios, our Bayesian model would have to be extended so as to also include, for example, mechanisms of causal inference[45]. It would enable the observer to determine whether or not the motion and orientation signals share a common cause and should be integrated or rather segregated, much like earlier mechanistic models have proposed before[2].

Our Bayesian decoding approach differs from previous methods in that we explicitly describe the generative structure of the data – that is, we model the effects on the cortical response for each stimulus. To infer the range of motion directions that could have caused a given cortical response (i.e., the posterior probability distribution), we simply invert this model using Bayes' rule (see Methods). This contrasts strongly with other decoding methods, such as SVM[46] or IEM[47], which merely focus on the response's single most likely interpretation. This methodological difference may explain why previous fMRI decoding studies[46,48–50] did not observe bimodality in the across-trial histogram of decoded motion directions: under low levels of uncertainty, the best guess estimate usually falls within a narrow range of the true direction of motion, and hardly ever on the opposite motion direction. Altogether, our findings show how fully characterizing the full probability landscape can improve our understanding of the computational mechanisms of cortical feature extraction.

Our work furthermore demonstrates the added value of visualizing cortical representations for understanding behavior. Crucially, while the influence of spatial orientation signals on direction estimation remained hidden in the participants' behavioral estimates, their effects were uncovered via the decoding of activity patterns in visual cortex. That is, while the behavioral histograms showed bimodality for

high-uncertainty conditions only, the decoded probabilistic cortical motion representation nonetheless revealed that orientation signals provide information about motion direction even when uncertainty is relatively low. These results point to the veiled intricacy of perceptual decision-making in a direction estimation task.

Furthermore, our results suggest that at even the earliest levels of cortical processing, multiple sources of evidence are combined to better represent the visual environment. It is well known that the visual system integrates multiple cues to infer visual properties for mid-level object properties, such as depth or object shape[1,4]. At a first glance, motion direction estimation may seem like a straightforward task, devoid of the need for additional evidence – after all, why would direction-sensitive neurons not provide sufficient information for direction estimation (see, e.g., [51], for a review)? However, as we show here, even for simple tasks the brain appears to utilize additional cues, such as spatial orientation, to reduce uncertainty. This highlights the fact that even simple tasks might be more intricate from the brain's perspective and that cue integration may be a ubiquitous feature of the human visual system.

## Methods

### Participants

This study complies with all relevant ethical regulations and was approved by the local ethics committee (Commissie Mensgebonden Onderzoek Regio Arnhem-Nijmegen, The Netherlands; Protocol CMO2014/288). Participants provided written informed consent before participation and received monetary compensation for their participation. 18 participants (aged 18-32, ten female, based on self-report) with normal or corrected to normal vision participated in the study. We did not test for differences in effect across gender, as it is unlikely that this factor will underlie differences in low-level visual cortical processing.

### fMRI data acquisition

MRI data were acquired using a Siemens 3 T MAGNETOM PrismaFit MR scanner with a 32-channel head coil located at the Donders Center for Cognitive Neuroimaging. For each participant and each session, a high-resolution T1-weighted magnetization-prepared rapid gradient echo anatomical scan (MPRAGE, FOV 256 × 256, 1-mm isotropic voxels) was collected at the start of the session. Functional imaging data were acquired using T2*-weighted gradient-echo echoplanar imaging covering the whole brain (68 slices, TR 1500 ms, TE 38.60 ms, FOV 210 × 210, slice thickness 2 mm, in-plane resolution 2.019 × 2.019 mm).

### Experimental design and stimuli

**fMRI experiment.** The fMRI experiment was run using an ASUS GL502V laptop (OS Kubuntu 17.04) connected to a luminance-calibrated projector EIKI LC - XL100 (resolution 1024 × 768 pixels, refresh rate 60 Hz). Participants viewed the visual display through a mirror mounted on the head coil. The stimuli were generated, and the experiment was controlled, using MATLAB and the Psychophysics Toolbox[52–54].

The stimulus consisted of dots coherently moving in a pseudo-randomly chosen direction (i.e., to ensure an even sampling of directions in each run, 18 evenly-spaced directions were selected from the full 360 deg. range with a random offset and were presented in random order during the run) within a circular aperture centered at the fixation point (inner radius 1.5 degrees of visual angle, dva; outer radius 7.5 dva; dot contrast reduced to 0 over the outer and inner 0.5 dva radius of the aperture). Each dot was white and had a Gaussian envelope with SD = 0.03 dva. There were 530 dots in total, resulting in an average density of approximately 3 dots/dva². The dot density was uniform within the aperture. Each dot was moving at 7 dva/s and had a limited lifetime of 10 to 14 frames (167 to 233 ms, randomly chosen for each dot). At the end of a dot's lifetime, it was pseudo-randomly repositioned in such a way that uniform dot density was maintained.

Participants were required to maintain fixation on a bull's eye target (diameter 0.5 dva) throughout the experiment. Each run consisted of an initial fixation period (12 s), followed by 18 trials (12.5 s each with a 4-second inter-trial interval) and a final fixation period (12 s). Each trial began with the disappearance of the fixation target, which reappeared after 100 ms. After another 400 ms, the stimulus was presented and remained on the screen for 1500 ms. This was followed by a 6 s fixation interval, after which a black line (length 0.9 dva) appeared at fixation (Supplementary Fig. 15). The participants reported the direction of motion of the dots by rotating this line. They did this by pressing the upper buttons on a Current Designs' HHSC-2×4-C fMRI response pad with the index fingers of the right and left hands. The response window was 4.5 s in duration, and the line began to dim after 3.5 s to indicate the approaching end of this window. Participants received no trial-by-trial feedback about the accuracy of their judgments.

The participants completed 39–49 stimulus runs during three experimental sessions on separate days. Before the experiment, the participants additionally participated in a 30-minute training session outside the scanner to ensure that they understood the task. Each scan session also included two visual localizer runs, which were used to select voxels that responded to the retinotopic location of the stimulus. The localizer stimulus consisted of moving dots presented within a circular aperture (described by the same parameters as the main experiment; it did not include the retinotopic area in which the response bar appeared). The dots were presented in seven 12-s intervals ("stimulus interval"), interleaved with fixation intervals of equal duration. During stimulus intervals, dot motion direction changed every 1.5 s, resulting in 8 directions of motion per interval. The 8 directions were chosen pseudo-randomly in the same way as during the main task. To ensure that participants paid attention to the localizer stimulus, they were asked to press a response button when the stimulus briefly dimmed to 50% contrast. Dimming events lasted 500 ms and appeared at random intervals with 2 to 7 seconds between events.

Retinotopic maps of the visual cortex were acquired in a separate scanning session using conventional retinotopic mapping procedures[55–57]. To determine the cortical boundaries of hMT +, we used two functional localizers based on a combination of approaches from previous studies[50,58–60]. Each localizer was repeated 3 to 7 times, either within a separate session or combined with the retinotopy session. For the first localizer, the participants viewed coherently moving dots presented in seven 12-s intervals, interleaved with seven 12-s intervals in which randomly-moving dots were presented. For the second localizer, we contrasted dots moving inwards or outwards from the fixation point (an optic flow pattern) with a static dot pattern, again presented in interleaved fashion with 12-s intervals. For both localizers, the dots had the same parameters as in the main fMRI experiment (i.e., dot color: white; Gaussian dot envelope with SD = 0.03 dva; 530 dots; dot density uniform; approximately 3 dots/dva²; each dot moving at 7 dva/s; limited lifetime of 10 to 14 frames (167 to 233 ms), randomly chosen for each dot). The dots were presented within an aperture with a radius of 8.7 dva and no inner window (dot contrast reduced linearly to 0 over the 0.5 dva outer radius of the aperture). During the coherent motion presentation (first localizer), the dots' direction changed every 1.5 s (8 evenly-spaced directions were selected from the full 360 deg. range with a random offset, and were presented in random order for each 12-s interval). During the random motion presentation, each dot direction was selected randomly. For the optic flow pattern (second localizer), motion direction (inward or outward) changed every 1.5 s. For the static dot pattern, the dots were generated using the same parameters as for the other localizers but did not move. The dot pattern was generated anew every 1.5 s. For both localizers, the same attention probes as for the within-session localizer were used: participants were asked to press a

response button when the stimulus briefly dimmed to 50% contrast (dimming events lasted 500 ms and appeared at random intervals with 2 to 7 seconds between events).

**Follow-up psychophysical experiment.** The follow-up behavioral study was run using a luminance-calibrated LaCie Electron 22blue II CRT display using the same laptop and software as in the fMRI experiment. A chinrest was used to stabilize the participant's head and reduce motion. The experiment was run in a dark, soundproof room with the display as the only light source. The task, run and trial structure, and stimuli were the same as in the main fMRI experiment, except that only 18% of the dots (randomly selected) on each trial moved in a single direction, while the directions of the remaining dots were distributed uniformly (i.e., randomly sampled from a uniform distribution). Participants responded using the left and right arrow keys on a keyboard. Each participant completed 12-20 runs within a single session.

## Data pre-processing

**Behavioral data.** Cardinal biases (Supplementary Fig. 16) were removed from the behavioral data by fitting four $4^{th}$-degree polynomials to each observer's behavioral errors as a function of motion direction. Specifically, we first determined bias direction by fitting two models that described either attraction or repulsion from cardinal directions. For the model that describes attraction biases, the behavioral errors are expected to be close to zero at cardinal directions, hence trials were split into four 90-degree bins centered at cardinal ({0, 90, 180, 270} degrees) directions. Dividing trials into bins enabled us to model the discontinuity that arises from repulsive biases around cardinal (see subject B in Supplementary Fig. 16 for an example; see also[17,61]). For each bin, we then fitted a regression model with $4^{th}$-degree orthogonal polynomials of motion direction (computed relative to the bin center) as independent variables and behavioral error as the dependent variable using the *GAMLSS* package in *R*[62]. To account for the heterogeneity of responses across motion direction (e.g., the oblique effect), the standard deviation of the behavioral errors was allowed to vary with distance to the polynomial's center. Accordingly, for each bin, we predicted the mean and standard deviation of participant errors as a function of the distance to the bin center. For the repulsion biases, on the other hand, the errors are expected to be close to zero at oblique directions. Hence, trials were split into four 90-degree bins centered at oblique ({45, 135, 225, 315} degrees) directions, but all remaining steps were identical. Both polynomial models were fitted to the data of each bin using maximum likelihood estimation. To remove the bias, the best fitting model (i.e., either attraction or repulsion bias) as indicated by their likelihood was selected. We used the residuals of these fits in subsequent analyses. We verified that our conclusions remain the same if no bias correction is applied. Errors that were larger than ±3 times the predicted standard deviation (obtained from the regression models described above) were considered outliers and not included in subsequent analyses (0.7% of all trials).

**fMRI data.** Functional images were motion corrected using FSL's MCFLIRT[63] and passed through a high-pass temporal filter with a cut-off period of 50 s to remove slow drifts in the BOLD signal. Residual motion-induced fluctuations in the BOLD signal were removed through linear regression, based on the alignment parameters generated by MCFLIRT. Functional volumes were aligned to an unbiased within-subject anatomical template, which was created using the participant's anatomical templates as obtained in each scanning session and FreeSurfer's longitudinal processing stream[64–66].

Regions of interests (ROIs) were defined using standard retinotopic procedures[55–57] (visual areas V1, V2, V3AB, and hV4) and a functional localizer (hMT+). Specifically, for each individual participant, hMT+ was delineated manually on the inflated cortical surface as the area that included voxels responding more strongly to both 1) moving

(i.e., optic flow patterns) rather than static dots ($p < .05$, FDR-corrected), and 2) coherent rather than random motion ($p < .05$, FDR-corrected; see Supplementary Fig. 17 for an example participant). Unless otherwise specified, individual ROIs were combined into a single ROI for the main analyses.

Voxels that responded to the retinotopic location of the stimulus were selected based on the within-session stimulus localizer. Specifically, within the native space of each participant, we selected all voxels within the ROI that were activated by the within-session stimulus localizer at a lenient threshold of $p < .01$ (uncorrected). The BOLD response of each voxel and time point within a given trial was z-normalized using the corresponding time points of all trials within the run. Activation patterns for each trial were defined by averaging together the first 4.5 s (3 TRs) of each trial, after adding a 3-s (2 TRs) temporal shift to account for the hemodynamic delay (Supplementary Fig. 18). Control analyses verified that the results were similar for individual visual areas (Supplementary Fig. 6), and not strongly affected by changes in the number of voxels selected for analysis (Supplementary Fig. 19). We furthermore confirmed that the selected time window was close to the peak of the hemodynamic response function (Supplementary Fig. 18).

For the control analyses involving head motion (Supplementary Fig. 5), we calculated for each participant and at each time step, the squared root of the sum of squares (i.e., the Euclidean norm) of the temporal derivatives of the realignment parameters as estimated by the motion correction algorithm; this quantity reflects the amount of head motion per time step. To obtain a measure of head motion per trial, the data were subsequently averaged across the trial's first 4.5 s, similar to our main analysis.

## Decoding analysis

We used a generative model-based method for estimating the degree of uncertainty in the cortical representation. This method (called TAFKAP[7,17,24]) computes the posterior distribution of motion direction from a given cortical response as measured with fMRI.

**Generative model.** The TAFKAP decoding algorithm assumes that BOLD activity varies randomly from trial to trial around a fixed stimulus-dependent mean that is different for each voxel:

$$b_i = f_i(s) + \varepsilon_i \tag{2}$$

where $b_i$ is the response of voxel $i$, $f_i(s)$ is the voxel's mean response to stimulus $s$, and $\varepsilon_i$ reflects random noise.

The mean response of the $i$-th voxel as a function of stimulus $s$ (i.e., the voxel's "tuning function," $f_i(s)$) is modeled as a weighted sum of $K = 8$ bell-shaped basis functions:

$$f_i(s) = \sum_k^K W_{ik} g_k(s) \tag{3}$$

$$g_k(s) = \max(0, \cos(s - \varphi_k))^5 \tag{4}$$

where $\varphi_k$ is the preferred motion direction of the $k$-th basis function (in radians), and the $K$ $\varphi_k$'s are spread evenly across the $2\pi$ motion space.

Around its tuning function $f_i(s)$, each voxel is assumed to fluctuate randomly due to Normally distributed noise $\varepsilon_i$. This noise is described by covariance matrix $\mathbf{\Omega}$, such that $\varepsilon \sim \mathcal{N}(0, \mathbf{\Omega})$. The probability of cortical activity pattern $\mathbf{b} = \begin{bmatrix} b_i \end{bmatrix}^T$ is therefore given by

$$p(\mathbf{b} \mid s, \boldsymbol{\theta}) = \mathcal{N}(\boldsymbol{f}(\boldsymbol{s}), \mathbf{\Omega})$$

where $\boldsymbol{\theta} = \{\mathbf{W}, \mathbf{\Omega}\}$ describes the model's free parameters (determined by the data), and $\boldsymbol{f}(s) = \begin{bmatrix} f_i(s) \end{bmatrix}$ are the voxel tuning functions (via $\mathbf{W}$ also determined by the data).

The covariance matrix of this multivariate Normal distribution was obtained as follows. Ideally, we would have used the sample covariance. However, when the number of voxels is larger than the number of trials, the estimation of the sample covariance matrix is non-invertible. To improve the estimation of the covariance matrix, TAFKAP therefore uses a concept called "shrinkage." Specifically, the model's covariance matrix $\mathbf{\Omega}$ is modeled as the sample covariance matrix $\mathbf{\Omega}_{\text{sample}}$ "shrunk" towards a parametrized theoretical covariance matrix $\mathbf{\Omega}_0$:

$$\mathbf{\Omega} = \lambda\mathbf{\Omega}_{\text{sample}} + (1-\lambda)\mathbf{\Omega}_0 \tag{5}$$

where $\lambda$ is a shrinkage parameter. The sample covariance is defined as follows:

$$\mathbf{\Omega}_{\text{sample}} = \frac{1}{N_{\text{train}}}\left(\mathbf{B} - \hat{\mathbf{W}}\mathbf{G}\right)\left(\mathbf{B} - \hat{\mathbf{W}}\mathbf{G}\right)^{\mathsf{T}}$$

And given TAFKAP's assumptions, the theoretical matrix $\mathbf{\Omega}_0$ is given by:

$$\mathbf{\Omega}_0 = \sigma^2\mathbf{W}\mathbf{W}^{\mathsf{T}} + (1-\rho)\text{diag}(\tau^2) + \rho\boldsymbol{\tau}\boldsymbol{\tau}^{\mathsf{T}} \tag{6}$$

where the first component ($\sigma^2\mathbf{W}\mathbf{W}^{\mathsf{T}}$) describes variance shared among similarly-tuned voxels, the second component ($(1-\rho)\text{diag}(\boldsymbol{\tau}^2)$) describes independent sources of variance, and the third component ($\rho\boldsymbol{\tau}\boldsymbol{\tau}^{\mathsf{T}}$) captures noise shared globally across all voxels. The $\tau_i^2$ parameter in TAFKAP is given by:

$$\tau_i^2 = \lambda_{\text{var}}\text{median}\left(\boldsymbol{\tau}'^2\right) + (1-\lambda_{\text{var}})\tau_i'^2 \tag{7}$$

where $\lambda_{\text{var}}$ is another shrinkage parameter. Please see ref. 17 for the derivation of the theoretical covariance matrix, and ref. 7 for further detail and rationale regarding TAFKAP's shrinkage estimation of the model's parameters.

**Training and testing.** Model parameters were estimated for each individual participant in a leave-one-run-out cross-validation procedure. That is, the model's free parameters were first estimated from the data of all but one run, and then the remaining run was used to decode posterior distributions on a trial-by-trial basis. Each run was used as a test set once. While training the model, TAFKAP uses "bootstrap aggregating" or "bagging" to take the uncertainty of model parameters into account. Specifically, trials in the training set were resampled many times (with replacement) to generate resampled data sets, each of which had the same number of trials as the original set. Model parameters were estimated for each set using ordinary least squares (see details in[7]). For each trial in the test set, the posterior distribution over motion direction was subsequently computed, conditioned on the fitted model parameters for a given bootstrapped training sample. The posterior distribution was obtained using Bayes' rule:

$$p\left(s \mid \mathbf{b}, \hat{\boldsymbol{\theta}}\right) = \frac{p\left(\mathbf{b} \mid s, \hat{\boldsymbol{\theta}}\right)p(s)}{\int p\left(\mathbf{b} \mid s, \hat{\boldsymbol{\theta}}\right)p(s)ds} \tag{8}$$

where the prior $p(s)$ was flat (reflecting the uniform distribution of motion directions in the experiment) and the normalizing constant $\int p(\mathbf{b} \mid s, \hat{\boldsymbol{\theta}})p(s)ds$ was estimated numerically. The posterior distribution was then averaged across each of the bootstrapping iterations to obtain one posterior per test trial. We took the circular mean of the decoded distribution as the estimated motion direction, and its entropy as a measure of uncertainty.

## Statistical procedures
**Benchmarking analyses.** When analyzing decoding accuracy, we computed the circular correlation coefficient between the decoded

and the true direction for each participant. We applied the Fisher transformation to individual coefficients and computed a Bayesian $t$-test on the transformed values. We used the standard (recommended) conservative priors in our Bayesian statistical analyses[67,68], both here and in the remaining analyses. The mean correlation coefficient across observers and its confidence intervals were computed on Fisher-transformed individual coefficients and the resulting values were transformed back to the correlational scale for reporting.

For analyses relating trial-by-trial uncertainty to behavioral variability, we used a Bayesian hierarchical regression with the *brms*[67] library in *R*. The analysis across motion directions included the bias-corrected squared behavioral error as the dependent variable. Trial-by-trial uncertainty (demeaned across all trials for each individual participant) was used as an independent variable, both as a population-level (fixed) effect and as a participant-level (random) effect along with participant-level (random) intercepts. This design allowed us to estimate the within-subject effect of uncertainty on behavioral variability while accounting for individual differences between participants. In the second set of analyses, we additionally controlled for differences between motion directions by including the oblique effect (distance to the nearest cardinal direction) in the model, both at the population and participant levels. In control analyses, we log-transformed the squared behavioral errors to account for the non-normality of their distribution; this did not change any of our conclusions.

**Analyses of the shape of the decoded distribution.** To test the predictions about the number of peaks in the decoded posterior (see Results; Bayesian observer model), we first fitted, for each subject, a descriptive bimodal model to both the mean posterior across trials and to single-trial posteriors. The model enabled us to estimate the location of the distribution's two peaks without any additional assumptions about the relationship between these peaks. The model is a mixture of two von Mises distributions and a circular uniform distribution:

$$\begin{aligned} f(x; \lambda, \alpha, \mu_1, \kappa_1, \mu_2, \kappa_2) = (1-\lambda)\big(\alpha f_{VM}(x; \mu_1, \kappa_1) \\ + (1-\alpha)f_{VM}(x; \mu_2, \kappa_2)\big) + \lambda U_{circular} \end{aligned} \tag{9}$$

where $\lambda$ is the weight of the uniform component, $\alpha$ is the relative weight of the first von Mises component, and $\mu_i$ and $\kappa_i$ are the mean and precision of the respective component. The probability density function $f_{VM}$ is a von Mises distribution and has two parameters, location (mean, $\mu_i$) and precision ($\kappa_i$):

$$f_{VM}(x; \mu, \kappa) = \frac{e^{\kappa\cos(x-\mu)}}{2\pi I_0(\kappa)} \tag{10}$$

where $I_0$ is a modified Bessel function of order 0. Because the component labels in this model are arbitrary, we disambiguate the components based on which one is larger (i.e., is higher at the maximum) or which one is located closer to the true direction of motion.

The model was fitted by minimizing the Jensen-Shannon divergence (JSD, a symmetrized version of the Kullback–Leibler divergence) between the decoded posterior and the model. The parameters were minimally constrained to avoid degenerate solutions: $\kappa_i \in [0.001, 100]$, $\alpha \in [1\times10^{-5}, 1-1\times10^{-5}]$, $\lambda \in [0, 0.9]$. To avoid local minima and assess the uniqueness of the solutions, we ran the optimization algorithm 100 times for each trial using random starting parameters, and computed the circular standard deviation of the estimated component locations across optimization runs. We found the solutions to be fairly unique. The SD across the estimated locations, averaged across trials and participants, was 0.15° for the larger component and 1.23° for the smaller component, for solutions with the JSD up to 5% larger than the optimal solution to allow for small numerical errors. The results were similar when the components were disambiguated based on the closeness to

the true direction rather than their height. Thus, while some trials did not result in a unique description of the decoded posterior (e.g., this might happen for a uniform posterior), most decoded posteriors were consistent with a unique description of component locations.

For the analyses of the mean posterior distribution, we averaged the posterior distribution across trials for each observer. We then estimated the best-fitting parameters for Eq. 9 as described above and computed the mean across observers and confidence intervals for both peak locations.

For analyses of peak location on a trial-by-trial basis, we did not categorize the trials into unimodal or bimodal based on the number of peaks in the single-trial posteriors. Instead, we use peak location to have objective, quantitative criteria for posterior shape analyses. That is, any goodness-of-fit measure is based on a statistical model (which describes how the data is generated so that the model's likelihood can be estimated). However, it is not clear how best to describe the statistical model for a mixture of functions that are fitted to the single-trial decoded posterior. This is why we tested for peak location instead, as the statistical model for peak location is much better understood. Specifically, we first fitted the von Mises mixture model (Eq. 9) to the trial-by-trial decoded posteriors for each individual participant. This gave us two peak locations for each trial, which are plotted as a joint probability distribution of peak locations across trials in Fig. 4c. For comparison, Supplementary Fig. 7 shows the predicted probability distributions for the fMRI data assuming that only orientation, only velocity or both signals are used. We then fitted two bivariate von Mises mixture models to the joint distribution of peak locations using the *BAMBI* package in $R^{69}$. The first model assumed that all location pairs belong to the same bivariate distribution (that is, a single cluster of trials is present) while the second assumed that they are best described as a mixture of two distributions (two clusters are present). We compared the models fits using the Watanabe–Akaike information criterion (WAIC).

To analyze the relationship between behavioral errors and the locations of the first and second peak in the decoded posterior, we first selected trials for which one of the peaks was closer to the true (−90° to 90°) and the other was closer to the opposite (90° to 270°) direction of motion (this selection criterion is conservative as it excludes trials for which both peaks correspond to approximately the same direction, that is, the "unimodal" trials). The peak locations were then transformed as $\mu' = \sin\left(\frac{\pi}{90}\mu\right)$ to account for the non-linear circular relationship predicted by the Bayesian observer model. In other words, we transformed the axes because of a non-linearity in the model predictions, which arises because of the circularity of the motion space. One (standard) way to linearize a circular variable is to apply a sine- and cosine-transformation (this is, for example, also done in a standard circular-circular regression). Because the model predictions are linear in the sine-transformed space (as shown in Fig. 2c), the transformation simplified our subsequent analyses. Next, we estimated the relationship between the transformed peak locations and the behavioral errors using a Bayesian hierarchical regression model that included behavioral errors as the dependent variable and the two peak locations as independent variables at both population-level (fixed) and participant-level (random) effects, as well as participant-level (random) intercepts.

**Follow-up psychophysical experiment.** In the analyses of the follow-up behavioral experiment, we compared two models fitted to the behavioral errors of each participant. We expected that if participants use both orientation and velocity signals, then there should be two peaks in the error distribution at the true and the opposite direction of motion. Accordingly, we fitted a von Mises mixture model (Eq. 9) to the behavioral error distribution with peak locations constrained to the true ($\mu_1 = 0°$) and opposite direction ($\mu_2 = 180°$). Alternatively, if observers do not use spatial orientation signals in their decision, there should be just one peak at the true direction of motion. For this

alternative hypothesis, we fitted a single-peak von Mises model ($\alpha = 1$ in Eq. 9). Both models included a uniform noise component. Because we fitted the model to the behavioral errors (rather than the decoded posterior distribution), we used a maximum likelihood (MLE) approach with the *DEoptim* package in $R^{70}$ instead of the JSD-based approach. The models were fitted to the data of each individual participant, and Bayesian information criterion (BIC) differences were summed across participants for the group-wise inference.

## Eye tracking data
Eye movements were recorded during the main fMRI experiment using an SR Research Eyelink 1000 eye tracker with 1000 Hz sampling rate and used for control analyses (Supplementary Figs. 5 and 8). After removing blinks, four variables were computed for the first 4.5 s of each trial. First, we computed gaze position as the absolute Euclidian distance between the fixation point (the screen center) and the mean gaze position within this time period. Second, we computed gaze position variability as the mean absolute Euclidian distance between point-by point gaze position and the mean position within this time period. Third, we computed the circular mean of saccade direction. Finally, we computed the mean axis of saccade direction as the circular average of all saccade directions wrapped in a 180-degree space.

## Bayesian observer models
The goal is to infer the direction of motion from noisy sensory signals. We consider two observer models for this task. The Bayesian observer model uses both the velocity and the spatial orientation measurements to estimate motion direction. The velocity-only model bases the judgments on velocity signals alone. Both models are described in three steps. First, we define the generative model that describes how the stimulus generates the velocity and orientation measurements of the observers. Second, we describe how each observer infers the range of motion directions that are likely given their measurement(s) – that is, how they compute the posterior distribution. The Bayesian model performs inference using both cues, whereas the velocity-only model only uses the velocity signals and ignores the spatial orientation measurements altogether. Finally, each observer selects their response given the computed posterior distribution combined with a cost function that determines how "bad" or costly potential errors are.

**Generative model (both models).** To infer the direction of motion $s$ of the stimulus, the observer measures its velocity ($x_V$) and spatial orientation ($x_O$) signals. These measurements are noisy, and are therefore best described as being drawn from a probability distribution; $p(x_V | s)$ and $p(x_O | s)$ for the velocity and spatial orientation signals, respectively. For velocity, we define the measurement probabilities as a von Mises (*VM*) distribution:

$$p(x_V | s) = f_{VM}(x_V; s, \kappa_V) \tag{11}$$

$$f_{VM}(x; s, \kappa) = \frac{e^{\kappa \cos(x-s)}}{2\pi I_0(\kappa)} \tag{12}$$

where $I_0$ is a modified Bessel function of order 0, and $\kappa$ is a precision parameter. Note that higher precision corresponds to lower circular standard deviation, $\sigma$:

$$\sigma = \sqrt{-2 \log\left(\frac{I_1(\kappa)}{I_0(\kappa)}\right)} \tag{13}$$

For spatial orientation, the measurement distribution $p(x_O | s)$ is similar to the velocity measurement distribution, but wrapped in the

180° orientation space. It is defined as follows:

$$p(x_O \mid s) = f_{VM}(x_O; s, \kappa_O) + f_{VM}(x_O; s + \pi, \kappa_O) \quad (14)$$

for any $x_O \in [0, \pi)$. Note that this function is unimodal in the orientation space, but bimodal in direction of motion. While this particular shape of the distribution is chosen to simplify the later analytical derivations, in principle any bell-shaped circular distribution can be used with qualitatively similar predictions.

Finally, the probability distribution of the stimuli (i.e., the prior distribution $p(s)$) is assumed to be a circular uniform distribution, $f_{UC}(s)$, corresponding to the uniform distribution of motion directions used in our task:

$$p(s) = f_{UC}(s) = \frac{1}{2\pi} \quad (15)$$

Together, $p(s)$, $p(x_M \mid s)$ and $p(x_O \mid s)$ define the generative model of how the moving stimulus gives rise to the measurements in our task.

**Inference (Bayesian model).** To infer the motion direction of the stimulus from the noisy sensory measurements, the Bayesian observer inverts the generative model. In other words, this observer estimates the most likely causes for the observed measurements by calculating the likelihood $L(s \mid x_V, x_O)$ of different stimulus values given the velocity and orientation measurements. The likelihood function given the velocity measurement alone is computed as follows:

$$L_V(s \mid x_V) = p(x_V \mid s) = f_{VM}(s; x_V, \kappa_V) \quad (16)$$

while the likelihood given the orientation measurement is:

$$L_O(s \mid x_O) = f_{VM}(s; x_O, \kappa_O) + f_{VM}(s; x_O + \pi, \kappa_O) \quad (17)$$

with the locations of the two peaks separated by 180 degrees ($\pi$ radians). Note that a horizontal orientation measurement is equally likely to be caused by a stimulus moving left and by a stimulus moving right. Hence, in the motion feature space, the likelihood becomes bimodal.

The Bayesian observer estimates the posterior distribution of motion direction s, $p(s \mid x_V, x_O)$, by computing the product of the stimulus likelihood $L(s \mid x_V, x_O)$ and the prior distribution $p(s)$:

$$p(s \mid x_V, x_O) \propto L(s \mid x_V, x_O)p(s) \quad (18)$$

Given that in our case the prior $p(s)$ is uniform, it can be subsumed under the proportionality sign:

$$p(s \mid x_V, x_O) \propto L(s \mid x_V, x_O) \quad (19)$$

In words, the posterior distribution is proportional to the likelihood of stimulus motion direction given the two measurements. Assuming that the velocity and orientation measurements are independent (the results are qualitatively similar if they are correlated, Supplementary Fig. 1), the likelihood $L(s \mid x_V, x_O)$ is:

$$L(s \mid x_V, x_O) = L_V(s \mid x_V)L_O(s \mid x_O) \quad (20)$$

Given the bimodality of the orientation-based likelihood, the likelihood given both measurements is bimodal, as well:

$$L(s \mid x_V, x_O) = f_{VM}(s; x_V, \kappa_V)\left(f_{VM}(s; x_O, \kappa_O) + f_{VM}(s; x_O + \pi, \kappa_O)\right) \quad (21)$$

Using Eq. 18 and 21 and properties of the von Mises distribution (see details in Supplementary Methods), we reformulate the posterior distribution as a weighted sum of two von Mises probability density functions, A and B, with weights $(w_A, w_B)$, means $(\theta_A, \theta_B)$ and precision

$(\kappa_A, \kappa_B)$ depending on the precision of the velocity and orientation components and the distance between their locations:

$$p(s \mid x_V, x_O) = w_A f_{VM}(s; \theta_A, \kappa_A) + w_B f_{VM}(s; \theta_B, \kappa_B) \quad (22)$$

Eq. 22 shows how the posterior can be described as a mixture of two components, which is useful when we quantify the shape of the posterior on a trial-by-trial basis (Eq. 9). While the equations specifying the parameters are given in the Supplementary Methods, we highlight three specific cases to provide an intuition about the posterior. First, when the variance of the velocity measurements is relatively low (that is, $\kappa_V$ is high), the weight of the second component $w_B$ approaches zero, so that the posterior becomes a unimodal von Mises distribution. In contrast, if the variance of the velocity measurements is high ($\kappa_V$ is approaching zero) and the variance of the spatial orientation measurements is relatively low ($\kappa_O$ is high), the posterior becomes bimodal of shape with two identical peaks at the true and opposite motion direction. Finally, when both velocity and orientation measurements are highly variable (both $\kappa_O$ and $\kappa_V$ are close to zero), the posterior becomes close to uniform (Supplementary Fig. 2).

**Decision-making (Bayesian model).** To judge the stimulus' direction of motion, the Bayesian observer estimates the relative cost associated with each response, as defined by the cost function combined with the posterior distribution. Which cost function is sensible for the decision about motion direction? An object moving in a given direction cannot simultaneously move in the opposite direction, hence if the posterior is bimodal, only one of the peaks would correspond to the true direction of motion, while another is just a by-product of the orientation signals. This suggests that a sensible strategy would be to use a delta cost function, selecting the most probable direction according to the posterior:

$$\hat{s}_{MAP} = \text{argmax}(p(s \mid x_O, x_V)) \quad (23)$$

Note that the mean squared-error cost function (which corresponds to taking the mean of the posterior) would be somewhat problematic for high-noise scenarios in which bimodality is observed, because the mean of a bimodal distribution falls in between the two peaks – it would create a paradoxical situation in which the true stimulus is never chosen as the response. Please also note that if the peaks in the posterior are well-separated, the MAP estimate matches a heuristic two-step strategy, by which observers first select the orientation peak that is more probable given the velocity peak location, and then estimate the true direction by combining this orientation peak and the velocity peak with any symmetric cost function (e.g., squared error). In other words, if observers use velocity estimates to disambiguate the orientation signals, and velocity signals provide relatively precise information, then the resulting estimate is the same as the maximum a posteriori (MAP) from the full posterior. The same results are also obtained when the velocity and orientation likelihoods are computed in a 180-degree space, and a separate binary variable (obtained from the velocity measurement $x_V$) indicates which half of the 360-degree motion space most likely contains the true direction of motion.

Given this decision strategy, what shape should we expect for the distribution of behavioral responses over trials? The distribution of maximum a-posteriori estimates is linked to the posterior distribution. When the velocity measurements have relatively low variance, the weight of the component corresponding to the opposite motion direction in the posterior ($w_B$) will approach zero and become negligible. The posterior distribution is then just the product of two unimodal likelihoods computed from the velocity and orientation measurements. In this case, the distribution of MAP estimates can be approximated by a von Mises distribution

(Murray & Morgenstern, 2010) with:

$$\hat{s}_{MAP} = x_V - \operatorname{atan2}\left(\sin(x_M - x_O), \frac{\kappa_V}{\kappa_O} + \cos(x_V - x_O)\right) \quad (24)$$

$$\mu_{MAP} = s \quad (25)$$

$$\kappa_{MAP} = \kappa_O + \kappa_V \quad (26)$$

That is, on a trial-by-trial basis, the maximum a posteriori estimate depends on the velocity and orientation measurements ($x_M$ and $x_O$) and the variability of these direction estimates is inversely related to precision parameters $\kappa_O$ and $\kappa_V$. Across trials, the distribution of the estimates is centered on the true stimulus ($s$) and its precision $\kappa_{MAP}$ is equal to the sum of the motion and orientation precision parameters. However, if the velocity measurements are highly variable, this approximation no longer holds, and simulations are necessary to assess the across-trial distribution of responses and its relationship with the parameters of the posterior.

**Inference (velocity-only model).** The velocity-only model follows the same inference steps but relies only on velocity estimates. First, the observer estimates the most likely causes for the observed measurements by calculating the likelihood $L_V(s|x_V)$ of different stimulus values given the velocity measurements (Eq. 16, repeated here for convenience):

$$L_V(s|x_V) = p(x_V|s) = f_{VM}(s; x_V, \kappa_V)$$

The observer then estimates the posterior distribution of motion direction s $p(s|x_V)$ by computing the product of the stimulus likelihood $L_V(s|x_V)$ and the prior distribution $p(s)$:

$$p(s|x_V) \propto L(s|x_V)p(s) \quad (27)$$

Given that in our case the prior $p(s)$ is uniform, it can be subsumed under the proportionality sign:

$$p(s|x_V) \propto L(s|x_V) \quad (28)$$

In words, for the velocity-only model, the posterior distribution is proportional to the likelihood of the stimulus given the velocity measurements.

**Decision-making (velocity-only model).** For the velocity-only model, any symmetric cost function (e.g., squared error or delta function) would result in the same decision. For consistency, we used the delta cost function, as we did for the Bayesian observer model:

$$\hat{s}_{velocity} = \operatorname{argmax}(p(s|x_V)) \quad (29)$$

The maximum of the posterior for the von Mises distribution lies at the measurement value, and the across-trial distribution of the MAP estimates is a von Mises distribution centered at the stimulus with precision $\kappa_V$.

**Simulations.** To obtain the predictions shown in Fig. 1b, we first simulated the posterior distribution for all of the possible combinations of the velocity and orientation standard deviation parameters spanning the range from 3° to 100° in 8 steps, $\sigma_V, \sigma_O \in \{3,5,10,20,30,40,60,100\}$. For each of the combinations, 10,000 trials were simulated using the measurement distributions for velocity and orientation (Eqs. 11, 14). The posterior distribution was calculated as described above (Eq. 21). For each trial, we obtained the MAP (maximum a-posteriori, i.e. the observer's judgment of motion direction) estimate by locating the maximum of the generated posterior on a 0.5°-step grid. For the velocity-only

model, the same simulated measurements were used but the inference was based on the velocity measurements alone (Eqs. 28, 29). The same simulated data was used for Fig. 2a and d, with the results split into low and high levels of uncertainty in the velocity likelihood. Fig. 2a compares low levels of uncertainty (i.e., using a 30° cutoff for $\sigma_V$) with high levels of uncertainty (i.e., $\sigma_V > 30°$). Fig. 2d shows the results for high levels of uncertainty (i.e., $\sigma_V > 30°$).

To facilitate a direct comparison with the posterior distribution decoded from the brain data (see Fig. 2b and c), in a second set of simulations, we simulated posteriors that are corrupted by additional (i.e., non-neuronal) sources of noise due to the fMRI measurements. MRI noise was modeled as independent noise on the observer's internal measurements, drawn from a von Mises distribution centered on 0, and with precision parameters $\kappa'_V$ and $\kappa'_O$ for the velocity and orientation measurements, respectively. This resulted in the following "decoded" likelihood:

$$L(s|x_V, x'_V, x_O, x'_O) = f_{VM}(s; x_V + x'_V, \kappa_V + \kappa'_V)\left(f_{VM}(s; x_O + x'_O, \kappa_O + \kappa'_O)\right. \\ \left. + f_{VM}(s; x_O + x'_O + \pi, \kappa_O + \kappa'_O)\right)$$

$$(30)$$

where $x'_V$ and $x'_O$ refer to random noise offsets caused by the fMRI measurements. Importantly, fMRI noise affected only the decoded likelihood and MRI measurements; the observer's measurements were unaffected by this particular form of noise.

For these simulations (Fig. 2b and c), the observer's measurements were drawn independently from two von Mises distributions (one for velocity and one for orientation). Because neural uncertainty varies on a trial-by-trial basis, the precision parameters of these von Mises distributions fluctuated across trials. Namely, on each trial $\kappa_V$ and $\kappa_O$ were drawn independently from a log-normal distribution with $\mu_{neur} = 3.8$ and $\sigma_{neur} = 0.6$. Parameter values were chosen such that the predicted distribution of behavioral responses (Eq. 24) matched the variability of the human participants' responses across trials (as estimated from the empirical data). These parameters were used in the model to predict the model's behavioral responses. The additional noise in the fMRI measurement of the observer's cortical representation ($\kappa'_V$, $\kappa'_O$) was also drawn randomly from a log-normal distribution with $\mu_{MRI,velocity} = 0.9$ and $\sigma_{MRI,velocity} = 1.1$ for velocity, and $\mu_{MRI,orientation} = 1.4$ and $\sigma_{MRI,orientation} = 0.7$ for orientation. These parameter values were chosen so as to match the actual posterior distribution decoded from the brain data (Eq. 30, obtained via searching on a parameter grid with 0.1 step for all four parameters). For the velocity-only model, the same simulated measurements were used, but the observer's decision was based only on the likelihood computed from the velocity signals (Eqs. 28, 29).

### Reporting summary
Further information on research design is available in the Nature Portfolio Reporting Summary linked to this article.

## Data availability
Preprocessed behavioral and fMRI data of individual participants generated in this study have been deposited in the Donders Repository database: https://doi.org/10.34973/yk4k-tp41[71]. This includes the data necessary to reproduce the figures. These data are available open access. The raw fMRI data are protected and are available upon request from the last author (Janneke F.M. Jehee) due to data privacy regulations. Requests for data will be answered within a reasonable timeframe (1 month).

## Code availability
Custom code for data analysis can be obtained via the Donders Repository: https://doi.org/10.34973/yk4k-tp41[71]. Custom code for the probabilistic decoding technique can also be found at https://github.com/jeheelab/[72].

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

## Acknowledgements

We would like to thank P. Gaalman for MRI support. This work was supported by European Research Council Starting Grant No. 677601 (to J.F.M.J.) and a Radboud Excellence Initiative fellowship (to A.C.). The funder had no role in study design, data collection and analysis, decision to publish or preparation of the manuscript.

## Author contributions

Conceptualization: A.C. and J.F.M.J.; Data curation: A.C. and J.F.M.J.; Formal analysis: A.C. and J.F.M.J.; Funding acquisition: A.C. and J.F.M.J.; Investigation: A.C.; Methodology: A.C. and J.F.M.J.; Project administration: A.C. and J.F.M.J.; Resources: J.F.M.J.; Software: A.C.; Supervision: J.F.M.J.; Validation: A.C.; Visualization: A.C. and J.F.M.J.; Writing – original draft: A.C. and J.F.M.J.; Writing – review & editing: A.C. and J.F.M.J.

## Funding

## Competing interests

The authors declare no competing interests.
