## [Peer Review File · Nature Communications]

Motion direction is represented as a bimodal probability distribution in the human visual cortexREVIEWER COMMENTS

Reviewer #1 (Remarks to the Author):

SUMMARY

To solve the mechanisms of motion direction estimation, visual neuroscientists have predominantly investigated computational models of velocity integration [1,2,3]. Geisler [4] proposed a novel and complementary idea: humans can exploit additional information, such as oriented motion streaks produced by motion temporal processing property, to compute motion direction. In this paper, the authors attempted to investigate the visual cortical mechanisms that integrate velocity signals with oriented motion streak signals, which they formalized as a Bayesian process. In agreement with the probabilistic population code framework [5, 6], they postulated that the probabilistic distributions of motion directions that are decoded from visual cortical responses to moving stimuli should reflect Bayesian posteriors. They used two experiments to test this hypothesis. First, eighteen subjects were asked to estimate the direction of motion of random dot kinematograms, in a very low noise condition (100% of the dots moved coherently in the same directions), while the authors collected fMRI BOLD responses from several visual cortical areas. They found that subjects' response variability increased with the uncertainty in subjects' decoded posteriors and that posteriors were bimodal as prescribed by their Bayesian model. In a separate psychophysics experiment, they collected subjects' behavioral estimates in response to high-noise kinematograms, in which a small proportion of the dots moved in the same direction. They found that subject trial distributions had the same bimodal shape as predicted by the Bayesian model. The authors concluded that human observers integrate motion-elicited spatial orientation signals with velocity signals in agreement with the normative Bayesian theory to improve motion direction estimation.

The question addressed by the authors is of significant interest to visual neuroscientists. Although the main question and findings of the study are clear, I describe below the reasons why I believe that the authors do not provide convincing evidence to support their claim.

MAJOR

1. To demonstrate that the decoded representations are Bayesian posteriors, the authors must show that the posteriors best fitted to subjects' estimates provide a better qualitative and quantitative fit to the decoded distributions than competing models that do not integrate the two distributions (e.g., a switching observer [7], the authors' null velocity-only observer or a constant response bias model). I suggest that they also show that the Bayesian model provides a best fit for most, if not all, subjects.
 - I do not find surprising the finding of bimodality. Separate populations of orientation-tuned neurons and velocity-tuned neurons coexist in visual cortical areas. Thus, a motion stimulus that elicits an explicit orientation signal, such as motion streak, will simultaneously elicit separate responses from both populations, yielding a decoded bimodal distribution.
 - (P5, L38): A Switching observer [7] that uses separate representations of spatial orientation and velocity in visual cortical areas would make the same predictions that the authors chose to test for their Bayesian model. The "Motion aftereffect" [8, 9] a tendency to report the direction opposite to the true motion direction seems to be an alternative hypothesis for their findings. I suggest that the authors rule out these hypotheses.
 - (p5, l36): The authors do not sufficiently test the qualitative predictions of the Bayesian model against their data to make for a convincing argument. The main prediction of Bayesian theory is that subjects should rely more on spatial orientation when velocity is noisy and more on velocity when the motion streak is weak. The authors could test these predictions to rule out a fixed response bias model, for example.
2. (p1, l40): There is some "direct neural evidence for streak-based computations in human motion direction is currently lacking". The authors should cite a relevant recent paper by Apthorp et al. [10]

who has shown that there is cortical neural evidence for the representation of motion streak.

3. Given the stimulus size, the decoded posteriors could correlate with the directions of subjects' saccades. I suggest that the authors rule out this confound.

4. The stimulus parameters used replicate the conditions required to reproduce motion streaks as described in the literature. The fMRI task and run designs, data collection and preprocessing pipelines and parameters, number of subjects used are consistent with those validated by past studies to decode motion direction probabilistic representations [5, 11]. The authors also provide sufficient details for replication and adequate references to these past studies. But the authors do not seem to discard the BOLD signals measured in the foveal region of the visual cortical areas before decoding. These signals are elicited by the response bar within the small aperture where the bar appears. Given that BOLD responses typically take 12 secs to completely come back to baseline and that the stimulus is displayed shortly (4.5 secs) after the end of the previous response phase, visual responses to the bar could be a confound to stimulus-elicited responses in individual trials. I suggest that the authors remove the foveal BOLD responses.

5. The authors should clarify whether motion coherence described in the psychophysics-only session ("follow-up") is 18% as stated (p9, l4) or 36% as described (p13, l15).

6. I suggest that the authors add the task figure as a panel to their first figure to describe it early. It enables the reader to concretely link the model prescriptions and the expected behavioral responses to the task.

7. Fig. 1b.

- (p4, l7): The authors should be more explicit in their description: they increased the circular std of the von mises from xxx to yyy values.

- (p4, l22-26): This figure does not demonstrate that cue combination improves motion direction estimation. To show that, the authors would need to plot the average estimate error on the y-axis. Estimate error - the difference between the model's maximum a posteriori readout and the true motion direction - is composed of two sources of errors: 1) a systematic bias away from the true motion direction and 2) random fluctuations around the average estimate. One can think of a scenario where estimate variability becomes smaller while systematic bias becomes much larger, increasing an observer's average estimate error.

8. Fig. 1c.

- (p4, l32): The authors write that "when the velocity signals provide no information at all (the velocity likelihood function is flat), the posterior becomes fully proportional to a bimodal orientation likelihood function". It is hard to understand. How can motion streak occur at all if there is no coherent motion direction at all, as all dots move in completely random directions? This scenario seems implausible.

- (p4, l30): The statement « for low levels of entropy» is too vague to enable replication and situate the described condition in panel b. The author should indicate the exact "noise" values, the circular standard deviation, that they used to simulate velocity measurement distributions.

- To show that their result is robust, the authors should demonstrate that bimodality holds for all displayed motion directions, which is a prescription of their Bayesian model.

9. Fig. 1d.

- (p4, paragraph l44): The authors describe fitting a mixture of two von mises to each trial posterior to determine trial posteriors' peaks. They do not describe the objective (quantitative) criteria that they used to decide whether a posterior is unimodal, bimodal, or uniform. Do they compare the goodness of fit (e.g., AIC) of a uniform, unimodal and bimodal distribution to each trial posterior to decide? If they do, I suggest that they explain it in the paragraph, in the method's "Simulation" section and in the figure's text legend.

- (p4, paragraph l44): The authors do not describe the noise condition (the exact circular std used) used to produce this panel. It would make sense for them to generate predictions here for the low-noise condition that they will test in fMRI (100% coherence). But given the large proportion of apparent bimodal trials on the plot, it is likely that they used a very high-noise condition. Is that correct?

10. Fig 1e.

- This panel introduces an important model prediction that is used to test whether decoded posterior bimodality is behaviorally relevant. However, the panel is very hard to understand from the figure legend's text and from the main text. They should use some of (p8, l3-7)'s description in paragraph (p5, l15) to make it clearer. Also, wouldn't it more intuitive and simpler to show that, for each given true motion direction, posterior peaks correlate with estimate distribution peaks?

- (p5, l17): The statement that "Figure 1e shows the behavioral estimates of direction of motion" is not consistent with the description made in the figure and with the previous paragraph's previous sentence. The y-axis or the figure's text legend should state that it is the (clockwise or counterclockwise) difference between the model's maximum a posteriori estimate and the true motion direction.

- I do not understand what the legend means. It indicates "peak at 0" for the blue line while the peak location changes on the x-axis. How can the peak be fixed at 0 and change at the same time?

- It is not intuitive why the authors sine-transform the x-axis. I recommend that they keep the axis in degrees, which is more intuitive and enables the comparison of the error on the y-axis with the peak location on the x-axis.

- For consistency, the y-axis should indicate "behavioral error". The behavioral response is the reported estimate not the error. It is currently confusing.

11. Fig 2a. The figure's text legend should indicate that this is the low-noise condition with 100% motion coherence to make Fig. 2a self-sufficient.

12. I suggest that the authors explain how to interpret BF (p6, l16,19), for example what BF value indicates sufficient evidence for significance. Also, I suggest that they motivate the use of the hierarchical Bayesian regression in the main text (p6, l37). Why is it the most adequate analysis?

13. Fig 2c. The authors chose to use the oblique effect [12] to test their first model prediction, that is that behavioral variability increases with posterior uncertainty. This restricts their ability to measure that relationship solely within a narrow range of uncertainties (about 7 to 8 bits), which differs from the range explored in their simulation (2 to 6 bits, Fig. 1b). As a result, we cannot compare Fig. 2c to Fig. 1b). This condition does not allow them to verify that that relationship is exponential (Fig. 1b). Critically, the data shown do not allow to disambiguate between a velocity-only and a cue-combination Bayesian observer that combines both cues. Both predicts the same qualitative relationship. The author would need to compare their goodness of fit to the data. But in that narrow range of uncertainty, that comparison is unlikely to be informative. To best test that hypothesis, the authors should have used a preliminary calibration procedure to measure the relationship between coherence and posterior uncertainty. In that procedure, subject undergo a similar task, in which different motion coherences are used at each trial (e.g., 6%, 24%, 100%). The authors then fit their Bayesian model to the reported estimates and select the coherences that yield the posterior uncertainties displayed in Fig. 1b's x-axis. That set of coherences should be used in their main task to demonstrate that, as coherence decreases, posterior uncertainty and estimate variability increase. They should compare the goodness of fit of the Bayesian model to competing models and show that their Bayesian model provides a better qualitative and quantitative account of the relationship between response variability and posterior uncertainty.

14. Fig 3.

- While their model predicts unimodal posteriors in low-noise conditions, the authors chose to decode posteriors during the fMRI experiment with 100% coherence motion stimuli. The uncertainty is the

lowest in that condition, particularly for the long stimulus duration used (1.5 sec). The decoded bimodality thus seems inconsistent with their model prediction. One hypothesis for why they would decode bimodal posteriors at 100% coherence is that they do not decode an integrated motion direction representation which shape depends on motion streak and motion coherence strengths. They decode separate coexisting orientation and velocity responses to the motion stimuli. A second hypothesis is that the decoded peak at the direction opposite to the true motion direction reflects motion aftereffect [4].

- The authors claim that the decoded distributions are bimodal but do not use quantitative model comparison to demonstrate the bimodality of the decoded distribution within and across subjects (same as comment 9.a).

- It is unclear why their decoded distributions are so smooth compared to [5]. Could bimodality be explained by noise or a noise filtering technique that they apply to the decoded distribution?

15. Fig 3c. It is not clear how statistically significant is the correlation, particularly the second peak. Also, for consistency the two panels of the corresponding figure in simulation should have the same legends (first peak and second peak, which are less confusing than the "peak at 0" and "peak at 180" legends).

16. (p7, l3): Contrary to the authors claim, decoding bimodal posterior does not reflect an "advantageous" estimation process if cues likelihood are not integrated multiplicatively according to the Bayesian rule, and particularly if they are not integrated at all. For example, choosing between the two with a fixed response bias would be particularly suboptimal.

17. Fig. 4: The authors should have collected fMRI responses in their follow-up condition. That is the condition where the Bayesian model predicts stronger posterior bimodality. Comparing the posterior in that high-noise condition with the posterior in the low-noise condition (100% coherence) will provide crucial evidence in support with Bayesian integration: if the ratio of the two peak amplitudes changes when coherence decreases as prescribed by Bayesian theory, it strengthens the Bayesian hypothesis.

18. Many supplementary figures are not of publishing quality.

References

- (1) Pasternak and Tadin, "Linking Neuronal Direction Selectivity to Perceptual Decisions About Visual Motion."
- (2) Adelson EH, Movshon JA. Phenomenal coherence of moving visual patterns. *Nature*. 1982;300:523-525.
- (3) Wilson HR, Ferrera VP, Yo C. A psychophysically motivated model for two-dimensional motion perception. *Vis Neurosci*. 1992.
- (4) Geisler WS. Motion streaks provide a spatial code for motion direction. *Nature*. 1999;400:65-69
- (5) van Bergen et al., "Sensory Uncertainty Decoded from Visual Cortex Predicts Behavior."
- (6) Ma, W., Beck, J., Latham, P. et al. Bayesian inference with probabilistic population codes. *Nat Neurosci* 9, 1432-1438 (2006). <https://doi.org/10.1038/nn1790>
- (7) Laquitaine, S., & Gardner, J. L. (2018). A switching observer for human perceptual estimation. *Neuron*, 97(2), 462-474.
- (8) Anstis, S., Verstraten, F. A., & Mather, G. (1998). The motion aftereffect. *Trends in Cognitive Sciences*, 2(3), 407 111-117.
- (9) Bae, G. Y., & Luck, S. J. (2021). Perception of opposite -direction motion in random dot kinem atograms. 412 *PsyArXiv*. <https://doi.org/10.31234/osf.io/uf3vd>
- (10) Apthorp et al., "Direct Evidence for Encoding of Motion Streaks in Human Visual Cortex."
- (11) van Bergen and Jehee, "Modeling Correlated Noise Is Necessary to Decode Uncertainty."
- (12) Gros, B. L., Blake, R., & Hiris, E. (1998). Anisotropies in visual motion perception: a fresh look. *JOSA A*, 15(8), 2003-2011.

Reviewer #2 (Remarks to the Author):

Re: Motion direction is represented as a bimodal probability distribution in the human visual cortex

In this article, the authors investigated the hypothesis that motion direction in the visual cortex is represented by integrating velocity signal with "orientation" signal (i.e., motion streak) resulting from a moving stimulus. The authors presented their hypothesis with detailed simulations of a Bayesian ideal observer model, and reported interesting empirical evidence from both fMRI and behavioral studies.

This article extended a previously proposed fMRI decoding method (Van Bergen et al., 2015; Van Bergen & Jehee, 2021) to the domain of motion. The authors showed that they can decode both participants' perceptual estimates and uncertainty of the motion direction. The authors observed a bimodal posterior distribution in their decoding results, which they interpreted as a neural signature of the motion streak signal in the visual cortex. However, I believe more substantial evidence needs to be provided to show the behavioral relevance of this bimodality. As they currently stand, the connections between the "neural bimodality" in experiment 1 and the "behavioral bimodality" in experiment 2 are rather indirect. Additional experiments need to be conducted to show a direct correspondence between the behavioral and neural bimodality.

I will detail my considerations below:

When a motion stimulus (e.g., random dot motion) is presented, the authors hypothesize that it results in two streams of evidence: Velocity, which provides a unimodal likelihood over motion direction; and "temporal orientation", which provides a bimodal likelihood over direction. Crucially though, the authors propose that they are combined in a Bayesian optimal way to form our percept of motion direction.

In the first experiment, the authors showed that the posterior distributions of motion decoded from the visual cortex are bimodal, with a smaller peak in the opposite direction. This is indeed evidence supporting the existence of something similar to "temporal orientation" in the visual cortex, which will result in bimodality. However, a tighter link to behavior needs to be established to demonstrate the behavioral relevance of this bimodality, and that subjects are indeed integrating velocity with temporal orientation. To this end, the authors showed that the location of both peaks in the bimodal posterior can predict behavioral error (Fig. 3c).

I am not sure if this is sufficient, as it seems possible that the correlation can be observed even when there is no integration. This could be the case due to a combination of many factors:

- 1) The noise is correlated between velocity and temporal orientation encoding. For example, due to shared retinal encoding noise (e.g., Angueyra & Rieke, 2013).
- 2) The overall uncertainties (e.g., variance of the noise) in the encoding of both processes are jointly modulated by common factors, such as attention effect (e.g., Goris, Movshon & Simoncelli, 2014). They both are also subject to the oblique effect, and have lower uncertainty close to the cardinal direction.
- 3) Lastly, correlated noise due to the BOLD signal itself can also introduce additional correlation in the fMRI decoding that was not due to integration.

The simulations in Fig. S6 partially address 1), but did not take 2) and 3) into account. Therefore, I am not sure if the observed correlation itself is sufficient to show that there is integration. On a side note, could the authors also provide some intuitions why they observed a negative correlation in some of the simulated conditions?

Perhaps equally importantly, since the motion coherence of the stimuli is 100% in the fMRI experiment, the resulting velocity likelihood should be very tight. Thus, one would expect the posterior to be unimodal (as shown by the authors in Fig S7, right column). I assume this can be tested by simulating the Bayesian observer with the noise level set to match the behavioral performance of subjects. If that is the case, why should we expect to see bimodality in the posterior decoded from voxel activity in the first place?

The authors did demonstrate a behavioral bimodality in the second experiment. It would be a much stronger result, for example, if on a trial-by-trial basis, the reverse in the perceived motion direction in experiment 2 can be predicted from the voxel activity. For example, experiment 1 could have been conducted using stimuli with a lower motion coherence, which would allow for analyses along this line. But as they currently stand, the connections between experiment 1 and experiment 2 seem weak.

Some other minor comments:

1) The authors cite previous literature regarding the idea of "motion streak", but it was rather brief (page 1, line 39 - 41). Since this idea is central to the paper, more background and a summary of previous work should be provided for the relevant context of the current study.

2) In the caption of Fig. 1, the authors used the terms "neural bimodality" and "noise bimodality", but it was not obvious why they are defined that way.

3) In Fig. 2c, is the effect of motion direction regressed out from the plotted behavioral variability (y-axis)? The caption mentioned "controlling for motion direction".

4) One might expect the "velocity" and "motion streak" information to be encoded in different visual areas. In that case, the likelihood should be unimodal in some areas, and highly bimodal in some other areas. Was there any evidence regarding such a separation?

Reviewer #3 (Remarks to the Author):

Review for the manuscript entitled "Motion direction is represented as a bimodal probability distribution in the human visual cortex" by Andrey Chetverikov, Janneke Jehee

This study combines computational modeling, behavioral and fMRI experiments to investigate the internal representation of visual motion in the human visual cortex, as well as its behavioral consequences. Using computational modeling, the authors found that the representation of motion should be bimodal. This bimodality should be reflected in the averaged posterior, the trial-to-trial likelihood function, and the overall behavioral error distribution. They then tested the model predictions using fMRI experiments. Leveraging Bayesian decoding techniques, the authors found evidence supporting the bimodality of the internal representation of motion direction both at the trial-averaged level and the individual-trial level. A follow-up behavioral experiment confirmed a further prediction of the modeling, i.e., the behavior estimates of a given motion direction should follow a bimodal distribution when the noise is large.

Overall, this study is well executed. The results are impressive and are presented in a logical fashion. It is relatively easy for the readers to follow the study. I enjoyed reading this paper and consider it a useful contribution. The paper will likely be of substantial interest to many in the field of cognitive and

computational neuroscience.

Having said that, I do have a number of concerns that I hope the authors could address or clarify in a revision.

1. My most serious concern is about the removal of motion biases in the analysis. Fig. S8 explained the procedure for removing these biases. I have multiple concerns. First, the procedure seems rather post-hoc and arbitrary. What's the rationale for removing this bias? The bias is an important and integrative part of the behavior error that contributes to the trial-by-trial error. Given the well-known bias-variance tradeoff, removing the bias may cause issues in the interpretation of the variance of the residual errors. If the bias was not removed, would that change any of the results in the paper? This would be important to know in order to give a more faithful interpretation of the data. The second question is more technical. Why was the estimated bias not a continuous function motion? Fig S8a shows the bias curve is discontinuous. Please clarify.

2. The following statement needs to be toned down.

"More broadly, our findings indicate that the cortical representation of low-level visual features, such as motion direction, can be far more complex than generally appreciated. "

It is generally appreciated that some of the V1 neurons are orientation-selective, while a portion of other neurons is direction-selective. Conceptually, some of the findings reported here could be interpreted as following from these electrophysiological observations. In addition, it would be useful to cite and discuss relevant neurophysiological studies on motion and direction selectivity in early visual processing (e.g., in V1).

3. The dependence of the velocity and spatial orientation signal needs an even more careful discussion. The authors have assumed the two to be independent in the modeling while acknowledging that the two should be dependent in reality. It would be useful if the authors could carry out additional simulations to demonstrate how to deal with the dependence between the two. Should the dependence be modeled as signal correlations or noise correlations? The authors seemed to indicate that they should be considered as noise correlations, but it also seems reasonable to think that they should be treated as signal correlations. A more careful treatment or discussion of these points in the context of the authors' model would be useful.

4. The results in the paper assumed a MAP estimate when performing Bayesian inference. This should be made clear in Line 16 of page 5 of the main text. Also, would using a different Bayesian estimator such as the posterior mean substantially change the results?

Other comments:

** Fig. S3c shows that the effect is not significant for the individual brain region. This should be mentioned in the main text. Furthermore, please indicate the effect size, R^2 , or the p-value in Fig 3c. More

** Line 30 of page 5. Why the difference in the entropy of the decoder between the cardinal and oblique directions is much smaller than the difference in precision? This needs to be addressed. If this means that there is substantial noise in the decoding, it should be stated so.

** It would be useful to report a scatter plot showing the observers' behavior in fMRI experiment (similar to Fig 4a). One possibility would be to place it in Fig 2. Another possibility is to make it a SI figure.

**Line 31 & 39 of page 5. What does variable b mean in these places?

**SI part 2. For the equation in Line 14, it would be helpful if the meaning of each variable could be explicitly defined.

** Fig S6, the explanation between lines 7- 20 is difficult to follow. Please get this better organized.

** In panel b of Fig S4, please change the color scheme so that the color changes gradually with the change of the number of voxels.

Reviewer #4 (Remarks to the Author):

There have been a number of studies over the past two decades directed at the hypothesis that spatial orientation signals (motion streaks) created by temporal integration in the early visual system provide orientation cues that are used in estimating motion direction, in addition to the more standard velocity cues extracted by cortical-neuron receptive fields that are oriented in space-time. Past studies have shown clearly that motion-streak cues are a highly useful source of motion-direction information, especially at higher retina speeds, and that motion-streak signals exist in the visual cortex of human and non-human primates. While it is highly likely that evolution would exploit such a useful source of information, it has not been convincingly demonstrated that motion-streak cues are actually combined with velocity cues in the human visual system to estimate motion direction. By measuring and analyzing human fMRI and behavioral responses to moving dot stimuli, this paper provides stronger evidence that the motion-streak and velocity cues are combined to estimate motion direction. The paper also uses the Bayesian cue-combination framework to model motion direction perception, which I also think has not been done before. Thus, I am quite positive. The paper is also clearly written. Nonetheless, I think there are some ways in which the paper could be improved.

1. In the initial modeling section, the authors assume that the motion streak and velocity cues are statistically independent, which is unlikely to hold exactly. The authors discuss this issue in the Discussion section, but something should be said up front, especially since a case can be made that they may be largely independent (e.g., the neurons that respond best to motion streaks are largely different from those that respond best to the sign of motion direction; although there are neurons that are sign selective for motion parallel to the receptive field orientation; Geisler et al. 2001).

2. The authors also assume a flat prior on motion direction. This is likely true for the sign of the motion direction, but not for the orientation. Adding a more realistic prior would not affect the conclusions of the paper. Again, something should be said about this up front.

3. I think a slightly different Bayesian formulation would be sensible to consider, although I have not tried to see if it is equivalent. Specifically, it makes sense to estimate motion direction as an estimate in the range of 0-180 (th), and separately estimate the sign of the motion direction (k), because the two cues (v and o) have different relative reliabilities for the 0-180 direction and for the sign of the direction. I will represent the estimates as \hat{th} and \hat{k} . For independent cues, the two estimates are: $\hat{th} = (r_{\rightarrow v} th^v + r_{\rightarrow o} th^o) / (r_{\rightarrow v} + r_{\rightarrow o})$ and $\hat{k} = (r_{\rightarrow v} k^v + r_{\rightarrow o} k^o) / (r_{\rightarrow v} + r_{\rightarrow o})$. (The r_{\rightarrow} subscript variables represent reliabilities). The reliability of the sign estimation for the streak signal is 0, so $\hat{k} = k^v$. The MAP estimate of the direction between 0 and 360 is thus $\hat{th} + 180$ if $\hat{k} = -1$, else it is \hat{th} if $\hat{k} = 1$. It might be best to estimate \hat{th} first, and then compare that estimate to $\hat{th} - 180$ to get the sign. This is a relative minor modeling difference, but I think it

would be worth checking out. It might be conceptually simpler for the readers.

4. The authors should make it clear to the reader that the relative reliability of the motion-streak signals to the velocity signals change with feature speed and feature size. They should state clearly that the results reported depend on the specific dot size and on the chosen speed of 7 deg/s. A future direction of study would be to vary dot speed and see how the relative reliability of the two cues in the fMRI responses varies and whether behavior follows along.

5. The chosen dot speed (7 deg/s) is a sensible choice, because it will produce both strong motion-streak and velocity signals in visual cortex. It would strengthen the paper to talk about what the study means for perception in the real world. By my calculation, if a person with eyes 5 feet about the ground plane is walking at 3 ft/sec (2 miles per hour), the optic flow in the ground plane will be 7 deg/s at about 24 ft to the left and right and at about 12 ft to the front. Ground plain features that are closer will have even greater speed (and greater relative reliability for the motion-streak signals). This, would make the case that the experimental stimuli are highly relevant for the kinds of optical flow signals that exist under real world conditions. I recommend double checking these calculations or carrying out similar calculations and including them in the paper.

6. It is a very nice result that motion direction over 360 degrees can be decoded from the fMRI responses.

7. The demonstration that humans make judgements of motion direction opposite to the true direction in high noise is also a very nice result demonstrating that motion streaks are actually used in estimation motion direction. I did not see if subjects were given feedback. Probably better if not. It would be good to state explicitly whether they were or were not.

8. That motion direction responses have a bimodal distribution in visual cortex, with a higher peak in the direction of the correct sign, is not at all surprising. That these correlate with behavior, even on a trial-by-trial basis is an important finding that provides more direct evidence that motion streaks play a functional role in motion perception/estimation and are not an epiphenomenon of temporal integration in the early visual system (retina).

9. If one could record from the whole population of V1 neurons with orientation preference perpendicular to the direction of motion, one would find that the population response those cells (the velocity-cue cells) would also have a bimodal population response with a weaker peak in the opposite direction (at least for gratings). This is because most V1 neurons have a second smaller peak in the opposite to the preferred direction. This might not hold for dot patterns but I am not sure it has been tested. Might be worth discussion this a bit.

10. p. 9 l. 4 Is it 18% or 30%. The methods indicates 30%.

11. p. 10 l. 14 "This strongly suggests that the observer's behavioral choices are based on the decoded neural representations." Does it? Can you predict the trial-by-trial from just thv? Is the prediction better for both cues together?

12. p. 10 l. 46 "Altogether, this strongly suggests that the human visual system uses spatial orientation signals for determining the direction of moving objects, and reveals the hidden complexity of probabilistic feature representations in cortex." The version of the Bayesian model I suggested above doesn't really have a complex probabilistic feature representation.

13. There is some relevant literature that should be included and perhaps discussed (see following list). One important one is a recent study (Tohmi M et al., 2021) showing that in the mouse there are populations of neurons with velocity selectivity that peaks at high at speeds in the direction parallel to the static orientation tuning. This is pretty strong evidence that motion steak signals are used for

encoding and decoding in the mouse cortex. These motion selective cells are most selective to streak signals and would not be there if they were not being used for motion direction coding. Because of the low spatial resolution of the mouse visual system, only large features are encoded and hence motion streaks only occur at high speeds. Thus, the streak cells in mouse are tuned to much higher speeds. But given their cruising speed and low eye height from the ground, the streak cells still encode optic flow signals over a substantial ground-plane region around the cruising mouse.

Apthorp, D., Cass, J., & Alais, D. (2011). The spatial tuning of "motion streak" mechanisms revealed by masking and adaptation. *Journal of Vision*, 11(7):17, 1–16, <http://www.journalofvision.org/content/11/7/17>, doi:10.1167/11.7.17.

Apthorp D, Schwarzkopf DS, Kaul C, Bahrami B, Alais D, Rees G. (2013) Direct evidence for encoding of motion streaks in human visual cortex. *Proc R Soc B* 280: 20122339. <http://dx.doi.org/10.1098/rspb.2012.2339>

Barlow HB and Olshausen BA (2004) Convergent evidence for the visual analysis of optic flow through anisotropic attenuation of high spatial frequencies. *Journal of Vision* 4, 415-426.

Burr, D., and Thompson, P. (2011). Motion psychophysics: 1985-2010. *Vision Res.* 51, 1431–296 1456.

Rasch, M.J., Chen, M., Wu, S., Lu, H.D., and Roe, A.W. (2013). Quantitative inference of population response properties across eccentricity from motion-induced maps in macaque V1. *J. Neurophysiol.* 109, 1233–1249.

Tohmi M, Tanabe S & Cang J (2021) Motion streak neurons in mouse visual cortex, *Cell Reports* 34, 108617.

Just for fun, I attached a photo I just took a couple of days ago, here is Colorado. The camera has a pretty fast shutter but one still gets motion streaks for the snowflakes, which are like the random dots in the authors experiment. One can see how the streaks convey the direction of motion. Feel free to use this if you want.

Bill Geisler

Reviewer #1 (Remarks to the Author):

SUMMARY

To solve the mechanisms of motion direction estimation, visual neuroscientists have predominantly investigated computational models of velocity integration [1,2,3]. Geisler [4] proposed a novel and complementary idea: humans can exploit additional information, such as oriented motion streaks produced by motion temporal processing property, to compute motion direction. In this paper, the authors attempted to investigate the visual cortical mechanisms that integrate velocity signals with oriented motion streak signals, which they formalized as a Bayesian process. In agreement with the probabilistic population code framework [5, 6], they postulated that the probabilistic distributions of motion directions that are decoded from visual cortical responses to moving stimuli should reflect Bayesian posteriors. They used two experiments to test this hypothesis. First, eighteen subjects were asked to estimate the direction of motion of random dot kinematograms, in a very low noise condition (100% of the dots moved coherently in the same directions), while the authors collected fMRI BOLD responses from several visual cortical areas. They found that subjects' response variability increased with the uncertainty in subjects' decoded posteriors and that posteriors were bimodal as prescribed by their Bayesian model. In a separate psychophysics experiment, they collected subjects' behavioral estimates in response to high-noise kinematograms, in which a small proportion of the dots moved in the same direction. They found that subject trial distributions had the same bimodal shape as predicted by the Bayesian model. The authors concluded that human observers integrate motion-elicited spatial orientation signals with velocity signals in agreement with the normative Bayesian theory to improve motion direction estimation.

The question addressed by the authors is of significant interest to visual neuroscientists. Although the main question and findings of the study are clear, I describe below the reasons why I believe that the authors do not provide convincing evidence to support their claim.

MAJOR

1. To demonstrate that the decoded representations are Bayesian posteriors, the authors must show that the posteriors best fitted to subjects' estimates provide a better qualitative and quantitative fit to the decoded distributions than competing models that do not integrate the two distributions (e.g., a switching observer [7], the authors' null velocity-only observer or a constant response bias model). I suggest that they also show that the Bayesian model provides a best fit for most, if not all, subjects.

We agree that comparing different models is important to demonstrate that the decoded representations reflect both motion and orientation signals, and that both sources of information are used in behavior. This is why in the manuscript, we compare both qualitatively and quantitatively the predictions of the Bayesian model with both a velocity-only observer (p. 8) and a model that assumes that orientation information is present in the brain but not used for behavior (p. 9).

Using quantitative procedures, we show that for almost all subjects (i.e., 16 out of 18) the second peak of the decoded distribution is located at the opposite motion direction, rather than a random location. Importantly, this is a pattern of results that the velocity-only observer cannot capture. We furthermore show that this holds both when averaged across trials and on a trial-by-trial basis. In addition, the second peak of the decoded distribution predicts the behavioral errors of our participants. Crucially, the velocity-only model cannot capture these results.

Importantly, note that we do, in fact, quantitatively compare between the models using the Watanabe–Akaike Information Criterion (WAIC), which is the gold standard in the field. We kindly refer the reviewer to our response to comment 9 for further rationale and detail behind our quantitative approach to model fitting and comparison.

To further address the reviewer’s concern, we additionally considered a constant response bias model and a switching model, as suggested by the reviewer. We provide the predicted behavioral response distributions for these models in Fig. S14. As can be seen in this figure, neither of these models make predictions consistent with our data.

That is, as far as we are aware, a constant response bias is usually discussed in the context of a forced-choice task, where observers might have a propensity to select a given response regardless of the presented stimulus. For example, observers might have a tendency to always select the presented stimulus in the left over the one presented in the right visual field. This cannot be applied here, however, as we do not use a forced-choice task. Observers might also be biased in an estimation task, as we use here. For example, the observer’s behavioral responses might always be biased away from the presented motion direction by around 5 degrees in a clockwise direction (see, e.g., Cicchini et al., 2021, *CurrBio*, for this use of the term). While this strategy would create biased responses, it cannot capture *bimodality* in the decoded posterior distribution, nor does it explain the bimodal behavioral response distribution as observed here. We now illustrate this model in Fig. S14.

The switching observer model was introduced by Laquitaine and Gardner (2018, *Neuron*). It assumes that an observer never integrates prior knowledge with the sensory likelihood, but rather switches between the two distributions when making a decision (i.e., a given response is based on prior *or* likelihood, but never both at the same time). Applied to our situation, such a strategy would have the observer switch between the orientation and velocity distributions. Much like the Bayesian model, it would predict that the behavioral response distribution is bimodal. However, contrary to the predictions of the Bayesian model, bimodality would *always* be observed for a switching model, regardless of the degree of uncertainty. Critically, this is not what we observe in our data, where bimodality in the behavioral response distribution clearly depends on stimulus reliability (Fig. S16). Only the combined model can capture these results.

We now also illustrate and discuss this in the text (p. 11, l. 38 onwards).

- I do not find surprising the finding of bimodality. Separate populations of orientation-tuned neurons and velocity-tuned neurons coexist in visual cortical areas. Thus, a motion stimulus that elicits an explicit orientation signal, such as motion streak, will simultaneously elicit separate responses from both populations, yielding a decoded bimodal distribution.

We believe our contributions are unexpected and novel in many ways. To the best of our knowledge, we are (1) the first to model motion perception within the Bayesian cue combination framework and (2) raise the possibility of a bimodal probabilistic representation in cortex. Not only do we make this surprising prediction, we also present (3) novel evidence to suggest that motion direction can, in fact, be extracted as a bimodal probability distribution from cortex. To date, no one has raised and investigated this possibility. In addition, we show that (4) this decoded distribution is linked in various ways to the behavioral judgments of the observer – such a direct link between the cortical motion response and human behavior had also not been observed before. Our computational work furthermore predicts (5) a new perceptual illusion that we also validated in experiments: participants sometimes perceive the stimulus as if it is moving in the direction *opposite* to the true direction of motion. As we show in the paper, this illusion is a direct consequence of the bimodal probabilistic nature of the cortical stimulus representation. Altogether, we believe that this presents the strongest evidence to date that not only velocity signals, but also spatial orientation signals are used in motion perception – a conclusion that stands in stark contrast to the dominant model of motion perception that assumes that motion direction is computed from direction-sensitive neurons alone (e.g., Rust et al., 2006, *Nature Neuroscience*).

- (P5, L38): A Switching observer [7] that uses separate representations of spatial orientation and velocity in visual cortical areas would make the same predictions that the authors chose to test for their Bayesian model. The “Motion aftereffect” [8, 9] a tendency to report the direction opposite to the true motion direction seems to be an alternative hypothesis for their findings. I suggest that the authors rule out these hypotheses.

We agree with the reviewer that it is important to test alternative models and now also consider the switching observer model, as discussed above. Importantly, this model cannot explain our findings (see above).

A model based on the motion aftereffect assumes that observers experience and report aftereffects after viewing the stimulus, which results in behavioral responses that are opposite to the true direction of motion. However, such a model would predict stronger aftereffects with greater motion coherence in the stimulus (so lower uncertainty) because the strength of the aftereffect is positively related to signal strength (e.g., Keck et al., 1976, *Vis. Res.*; Nishida et al., 1997, *Vis. Res.*). Therefore, greater bimodality in behavior is expected with low uncertainty – this is opposite to what we find.

We now also discuss this in the text (p. 12, l. 4-10).

- (p5, l36): The authors do not sufficiently test the qualitative predictions of the Bayesian model against their data to make for a convincing argument. The main prediction of Bayesian theory is that subjects should rely more on spatial orientation when velocity is noisy and more on velocity when the motion streak is weak. The authors could test these predictions to rule out a fixed response bias model, for

example.

We believe it is important to consider the research questions of the current paper here. That is, we asked whether spatial orientation signals are combined with velocity signals in motion perception, but never intended to show (nor do we mention anywhere in the text) that these cues are integrated *optimally* (i.e., weighted by their uncertainty). In fact, we believe it may be difficult, if not impossible, to test for optimality in this particular situation.

The main reason for this is that it will be difficult to infer the likelihood for the spatial orientation and velocity-based signals alone. That is, in a typical cue integration experiment, the likelihood of each cue is manipulated by the experimenter and therefore (roughly) known. This makes it possible to predict what the behavioral response should be for both the optimal integration strategy and alternative strategies that ignore uncertainty. For the integration problem considered here, however, the likelihoods are not a priori known to the experimenter, and would have to be inferred from brain data. An fMRI voxel, however, reflects the aggregate response of many neural populations, where the responses from the individual populations are unknown (and cannot be inferred from voxel activity). Without knowledge of the individual signals for spatial orientation and velocity, their likelihoods cannot be calculated, which makes it impossible to predict and compare the integration strategies.

We now also discuss this in the text (p. 12, l. 11-23).

To acknowledge that our results are also consistent with an integration strategy that ignores uncertainty, we now discuss in the manuscript the theoretical predictions of a model that uses random integration weights (i.e., the response is given as a weighted sum of the spatial orientation-based estimate and velocity-based estimate with random weights, see Figure S14). Please note, however, that the random-weight model predicts a large drop in performance when orientation signals are very noisy. This prediction is inconsistent with previous empirical results showing that observers perform relatively well at slow motion speeds when orientation information is presumably very noisy or even absent (e.g., Edwards & Crane, 2007, Vision Research). Together with our findings, this suggests, albeit indirectly, that human observers likely do consider uncertainty when integrating the two sources of evidence.

2. (p1, l40): There is some “direct neural evidence for streak-based computations in human motion direction is currently lacking”. The authors should cite a relevant recent paper by Apthorp et al. [10] who has shown that there is cortical neural evidence for the representation of motion streak.

We agree that Apthorp et al. (2013) is relevant here. Apthorp et al. showed that a 2-way classifier trained on cortical responses for static orientation stimuli can reliably discriminate the direction of motion of the stimulus. However, this held for signals from area V2 only and not for any of the other areas that were tested (so no evidence for orientation signals in V1, V3, and hMT+). In addition, decoding performance in V2 was generally low (~60% for a 2-way classifier), and a classifier trained on motion showed either no reliable discrimination (fast motion) or even a tendency for below-chance discrimination (slow motion, Figure 5c in Apthorp et al.). This altogether casts some doubt on the strength of their evidence. In addition, they demonstrated no clear relationship with the observer's

behavior, enhancing the possibility that some of their results may have reflected artifacts in the measurements. Thus, while some of their results may have been consistent with streak-based signals, the presented evidence was mixed at best. We now discuss this in the text (p. 11, l. 27 onwards).

3. Given the stimulus size, the decoded posteriors could correlate with the directions of subjects' saccades. I suggest that the authors rule out this confound.

Following the reviewer's suggestion, we ran additional control analyses involving the direction of the saccades made by the participants. Specifically, we tested whether saccade direction could explain the relationship between the decoded posterior and the direction and magnitude of the behavioral errors (Fig. 3c). For each trial, we computed the mean saccade direction over time using the temporal window corresponding to the presentation of the stimulus and used this quantity as a control variable. We found that the addition of this control variable had virtually no effect on our results. In addition, we ran the same test for saccades along the motion direction axis (i.e., in orientation space) and for saccade direction and orientation transformed from circular to linear space using the same sine-transformation as for the peak locations. However, these control analyses again showed that saccades cannot explain the relationship between the peak locations and behavioral errors. We now discuss this in the text (p. 9, l. 19 onwards) and have added the corresponding figures (Figure S15) to the supplement.

4. The stimulus parameters used replicate the conditions required to reproduce motion streaks as described in the literature. The fMRI task and run designs, data collection and preprocessing pipelines and parameters, number of subjects used are consistent with those validated by past studies to decode motion direction probabilistic representations [5, 11]. The authors also provide sufficient details for replication and adequate references to these past studies. But the authors do not seem to discard the BOLD signals measured in the foveal region of the visual cortical areas before decoding. These signals are elicited by the response bar within the small aperture where the bar appears. Given that Bold responses typically take 12 secs to completely come back to baseline and that the stimulus is displayed shortly (4.5 secs) after the end of the previous response phase, visual responses to the bar could be a confound to stimulus-elicited responses in individual trials. I suggest that the authors remove the foveal BOLD responses.

We agree that the BOLD response elicited by the response bar can be a confound, which is why we focused all of our analyses on a narrow time window *before* the bar appeared on the screen (see p. 16 for a description and Figure S5). In addition, voxels were selected based on localizers that did not include the aperture in which the bar appeared (p. 14). In other words, we already excluded response bar-related signals from our analysis and do not see how the response bar could have affected any of our conclusions. Please also note that this analysis pipeline fully replicates that of our earlier work (van Bergen et al., 2015; van Bergen & Jehee, 2019, cited by the reviewer as [5,11] above; Geurts et al., 2022, Nature Human Behaviour).

5. The authors should clarify whether motion coherence described in the psychophysics-only session (“follow-up”) is 18% as stated (p9, l4) or 36% as described (p13, l15).

18%, fixed.

6. I suggest that the authors add the task figure as a panel to their first figure to describe it early. It enables the reader to concretely link the model prescriptions and the expected behavioral responses to the task.

We thank the reviewer for the suggestions. We have revised this figure and added a photo of real-world motion streaks to the figure (courtesy of reviewer 4). We hope this better illustrates the link between the model’s predictions and behavior.

7. Fig. 1b.

- (p4, l7): The authors should be more explicit in their description: they increased the circular std of the von mises from xxx to yyy values.

We now describe these details in the next paragraph as follows: “We analyzed the relationship between behavioral variability and uncertainty for each model observer with velocity and orientation standard deviation parameters spanning the range from 3° to 100°” (p. 4, l. 26).

- (p4, l22-26): This figure does not demonstrate that cue combination improves motion direction estimation. To show that, the authors would need to plot the average estimate error on the y-axis. Estimate error - the difference between the model’s maximum a posteriori readout and the true motion direction - is composed of two sources of errors: 1) a systematic bias away from the true motion direction and 2) random fluctuations around the average estimate. One can think of a scenario where estimate variability becomes smaller while systematic bias becomes much larger, increasing an observer’s average estimate error.

The reviewer suggests a scenario where the posterior is (on average) not centered on 0 (i.e., on the true stimulus), creating a systematic bias in the observer’s behavioral responses. However, this is different from our situation, in which the expected mean of the posterior is, in fact, centered on 0 (so, on the true stimulus).

To see why, please note that there are two likelihoods, one computed from orientation signals and another computed from velocity signals. The ‘velocity’ likelihood is (on average) centered on the true motion direction, so it does not bias the expected mean of the posterior. Because of this, the behavioral responses of the velocity-only model are not biased, and their mean is centered on the true stimulus.

The same rationale holds for the first peak of the ‘orientation’ likelihood. The second peak of the ‘orientation’ likelihood is centered (on average) on the opposite motion direction. Because the space is circular, however, it does not pull the mean of the posterior in a specific direction. So, when combined, the two likelihoods create a posterior that is (on average) still centered on the true motion direction. Consequently, the behavioral response distribution of the combined (Bayesian) model is not biased and still centered on the true stimulus.

To further illustrate this, we simulated both the mean behavioral error and the mean absolute behavioral error across trials for the two models. As shown in the figure below, the mean error across trials for both models is zero (so no systematic bias). For completeness, we also include the mean absolute error; note that its pattern is the same as that obtained using SD in figure 1b (because the expected value of the average absolute error is equal to SD).

Figure R1. Mean absolute error and mean error as a function of uncertainty for the full model and velocity-only model.

8. Fig. 1c.

- (p4, l32): The authors write that “when the velocity signals provide no information at all (the velocity likelihood function is flat), the posterior becomes fully proportional to a bimodal orientation likelihood function”. It is hard to understand. How can motion streak occur at all if there is no coherent motion direction at all, as all dots move in completely random directions? This scenario seems implausible.

Here, we discuss a hypothetical scenario in which the internal representation strongly deviates from the physical stimulus. For example, a low coherence stimulus combined with strong attentional effects on motion streaks might create a representation that is dominated by orientation signals. Of course, our example is an extreme, but we believe discussing such extreme scenarios may help some readers to better understand the model.

To make our text more accessible to a broad readership, we have nonetheless revised our wording and hope the current version is more clear.

- (p4, l30): The statement « for low levels of entropy» is too vague to enable replication and situate the described condition in panel b. The author should indicate the exact “noise” values, the circular standard deviation, that they used to simulate velocity measurement distributions.

Agreed. We simulated the model predictions (see the Simulations section in Methods, p. 24) for all possible combinations of the velocity and orientation variance parameters spanning the range from 3° to 100° (in 8 steps) to ensure that our predictions hold for different parameter settings. For Figures 1c and 1f, the results are split into low and high levels of uncertainty in the velocity likelihood using a 30° cutoff for σ_V (i.e., $\sigma_V > 30^\circ$ corresponds to high uncertainty and $\sigma_V \leq 30^\circ$ for low uncertainty). We mention this in the figure legend and in the text on p. 4, l. 41.

- To show that their result is robust, the authors should demonstrate that bimodality holds for all displayed motion directions, which is a prescription of their Bayesian model.

Since these simulations do not include any prior for motion direction, they are independent of the specific choice of motion direction. So, they indeed hold for all motion directions. For completeness, we include a plot of the posterior distributions across motion directions for the posteriors decoded from the human fMRI data below.

Figure R2. Average posterior distribution decoded from the fMRI data for different motion directions. Color shows the posterior probability for each decoded direction and each stimulus motion direction (in 100 equally-sized bins). Higher posterior probability for directions 180 degrees apart from the true ones is clearly visible.

9. Fig. 1d.

- (p4, paragraph l44): The authors describe fitting a mixture of two von mises to each trial posterior to

determine trial posteriors' peaks. They do not describe the objective (quantitative) criteria that they used to decide whether a posterior is unimodal, bimodal, or uniform. Do they compare the goodness of fit (e.g., AIC) of a uniform, unimodal and bimodal distribution to each trial posterior to decide? If they do, I suggest that they explain it in the paragraph, in the method's "Simulation" section and in the figure's text legend.

We assume that the reviewer refers to the paragraph in the text that describes Fig. 1d, along with the figure caption. Please note that this figure presents simulated data.

Unfortunately, the proposed statistical measures are of limited use when analyzing simulated results. This is because the analysis compares between nested models (the unimodal model is nested within the bimodal), so the more complex model will always fit better due to overfitting. For empirical datasets, this issue is easily resolved by using goodness-of-fit measures that penalize for model complexity, such as the Akaike Information Criterion (AIC). However, given the incredibly large number of trials in a simulated dataset (100,000), penalizing for complexity won't solve the issue: the more complex model will always win, no matter how the data were generated (so even when the simulated posterior was, in fact, unimodal). This is why, for the simulated data, we instead focused on the qualitative pattern of predictions, which clearly differs between the models. We have included a figure to show this below. (Please note that we did use quantitative tests when analyzing the empirical data.)

Please also note that we did not categorize the trials into unimodal or bimodal based on the *number* of peaks in the single-trial posteriors. Instead, we use peak *location*. The reason for this is precisely because of the need for objective, quantitative criteria as mentioned by the reviewer. That is, any goodness-of-fit measure is based on a statistical model (which describes how the data is generated so that the model's likelihood can be estimated). However, it is not clear how best to describe the statistical model for a mixture of functions that are fitted to the decoded posterior. This is why, in the empirical data, we tested for peak location instead, as the statistical model for peak location is much better understood.

Specifically, when analyzing the empirical data, we fitted two bivariate von Mises mixture models to the locations of the peaks across trials and compared between the models using the Watanabe–Akaike Information Criterion (WAIC) (as described on p. 19, l. 34 in Methods). Note also that the joint distribution of peak locations predicted by the velocity-only, orientation-only, and combined observer models are very different (see below, we now also include the plots for simulated data in the manuscript, Figure S17), so this procedure enabled us to not only quantitatively but also qualitatively adjudicate between the observer models in the empirical data. We now discuss this more thoroughly in the text (p. 8, p. 19).

Figure R3. A comparison of the joint probability density for the trial-by-trial distribution of peak locations for the data simulated with the generative BOLD model and the empirical data. The plots for the simulated data shown here are now included in Figure S17.

- (p4, paragraph I44): The authors do not describe the noise condition (the exact circular std used) used to produce this panel. It would make sense for them to generate predictions here for the low-noise condition that they will test in fMRI (100% coherence). But given the large proportion of apparent bimodal trials on the plot, it is likely that they used a very high-noise condition. Is that correct?

These bimodal trials arise because of noise in the fMRI measurements (see equation 30 and p. 24, line 15). Specifically, for the simulations of Fig. 1d, 1e, the observer's measurements were drawn independently from two von Mises distributions (one for velocity and one for orientation). Because neural uncertainty varies on a trial-by-trial basis, the precision parameters of these von Mises distributions fluctuated across trials. That is, on each trial, κ_V and κ_O were drawn independently from a log-normal distribution with $\mu_{\text{neur}} = 3.8$ and $\sigma_{\text{neur}} = 0.6$. Parameter values were chosen such that the predicted distribution of behavioral responses (Eq. 26) matched the variability of the participants' behavioral responses across runs (as estimated from the data).

The additional noise in the fMRI measurement of the observer's cortical representation (κ'_V , κ'_O , see equation 30) was also drawn randomly from a log-normal distribution with $\mu_{\text{MRI,velocity}} = 0.9$ and $\sigma_{\text{MRI,velocity}} = 1.1$ for velocity, and $\mu_{\text{MRI,orientation}} = 1.4$ and $\sigma_{\text{MRI,orientation}} = 0.7$ for orientation. These parameter values were chosen so as to match the actual posterior distribution decoded from the brain data (Eq. 30, obtained via searching on a parameter grid with 0.1 step for all four parameters). For the velocity-only model, the same simulated measurements were used, but the observer's decision was based only on the posterior computed from the velocity estimates (Eqs. 28, 29).

Please note that it is well-known that fMRI scanner noise and various physiological sources of noise contribute to the BOLD signal (as also assumed by our decoding model, see e.g., van Bergen et al, Nature Neuroscience 2015; van Bergen & Jehee, NeuroImage 2018).

We now better describe this in the text (p. 5 l. 5, p. 24), and additionally added two figures to further illustrate how neural variance combined with fMRI noise gives rise to bimodal probability distributions at the level of fMRI voxels (Figure S10, S11).

10. Fig 1e.

- This panel introduces an important model prediction that is used to test whether decoded posterior bimodality is behaviorally relevant. However, the panel is very hard to understand from the figure legend's text and from the main text. They should use some of (p8, l3-7)'s description in paragraph (p5, l15) to make it clearer. Also, wouldn't it more intuitive and simpler to show that, for each given true motion direction, posterior peaks correlate with estimate distribution peaks?

We thank the reviewer for the suggestion, and have expanded the description of 1e in both the caption and main text (p. 5). We hope the current version is more clear.

- (p5, l17): The statement that "Figure 1e shows the behavioral estimates of direction of motion" is not consistent with the description made in the figure and with the previous paragraph's previous sentence. The y-axis or the figure's text legend should state that it is the (clockwise or counterclockwise) difference between the model's maximum a posteriori estimate and the true motion direction.

Fixed.

- I do not understand what the legend means. It indicates "peak at 0" for the blue line while the peak location changes on the x-axis. How can the peak be fixed at 0 and change at the same time?

We have now updated the labels to make this clear. The peak locations were used to classify the peaks on each trial into the peak that is closer to the true stimulus direction ("around 0") or closer to the opposite direction ("around 180").

- It is not intuitive why the authors sine-transform the x-axis. I recommend that they keep the axis in degrees, which is more intuitive and enables the comparison of the error on the y-axis with the peak location on the x-axis.

We transformed the axes because of a non-linearity in the model predictions, which arises because of the circularity of the motion space. One (standard) way to linearize a circular variable is to apply a sine- and cosine-transformation (this is, for example, also done in a standard circular-circular regression). Because the model predictions are linear in the sine-transformed space (as shown in this figure), the transformation greatly simplified our subsequent analyses, which is why we applied it. We prefer to stay close to our analyses and plot the data in the same transformed space. We now thoroughly explain (the rationale behind) these procedures in the Methods section (p. 19) as well as the figure's caption, and hope this will make our application of the transformation more clear.

- For consistency, the y-axis should indicate "behavioral error". The behavioral response is the reported estimate not the error. It is currently confusing.

Fixed.

11. Fig 2a. The figure's text legend should indicate that this is the low-noise condition with 100% motion coherence to make Fig. 2a self-sufficient.

Fixed.

12. I suggest that the authors explain how to interpret BF (p6, l16,19), for example what BF value indicates sufficient evidence for significance. Also, I suggest that they motivate the use of the hierarchical Bayesian regression in the main text (p6, l37). Why is it the most adequate analysis?

Thank you for the suggestions. We now also motivate our use of the Bayesian hierarchical model in the main text. We have also added further information on Bayes factors, as per the reviewer's suggestions (p. 6).

13. Fig 2c. The authors chose to use the oblique effect [12] to test their first model prediction, that is that behavioral variability increases with posterior uncertainty. This restricts their ability to measure that relationship solely within a narrow range of uncertainties (about 7 to 8 bits), which differs from the range explored in their simulation (2 to 6 bits, Fig. 1b). As a result, we cannot compare Fig. 2c to Fig. 1b). This condition does not allow them to verify that that relationship is exponential (Fig. 1b). Critically, the data shown do not allow to disambiguate between a velocity-only and a cue-combination Bayesian observer that combines both cues. Both predicts the same qualitative relationship. The author would need to compare their goodness of fit to the data. But in that narrow range of uncertainty, that comparison is unlikely to be informative. To best test that hypothesis, the authors should have used a preliminary calibration procedure to measure the relationship between coherence and posterior

uncertainty. In that procedure, subject undergo a similar task, in which different motion coherences are used at each trial (e.g., 6%, 24%, 100%). The authors then fit their Bayesian model to the reported estimates and select the coherences that yield the posterior uncertainties displayed in Fig. 1b's x-axis. That set of coherences should be used in their main task to demonstrate that, as coherence decreases, posterior uncertainty and estimate variability increase. They should compare the goodness of fit of the Bayesian model to competing models and show that their Bayesian model provides a better qualitative and quantitative account of the relationship between response variability and posterior uncertainty.

The reviewer appears to suggest that the critical aspect of figure 1b is that it shows an exponential relationship between uncertainty and behavioral variability. However, this is not what we meant to illustrate with this figure. Instead, the figure intends to show that both uncertainty and behavioral variability are lower when velocity signals are combined with orientation signals, as also explained in the figure's caption and the main text (p. 4, l. 24). It also demonstrates that uncertainty will always be linked to behavioral variability, no matter its value. The specific shape of the relationship (exponential) is identical for both models – so, contrary to the suggestions of the reviewer, this prediction cannot be used to dissociate between the models.

It is also important to realize that fMRI will add additional variance to the uncertainty estimates, while areas downstream of visual cortex will add additional variance to behavior. Because of this, the relationship between uncertainty and behavioral variability can only be tested qualitatively, and never quantitatively, contrary to the suggestions of the reviewer. This altogether means that it will be impossible to dissociate between the two models based on the observed values of uncertainty and behavioral variability alone. So, unfortunately, and to the best of our knowledge, there is no way that the suggested calibration procedures could help in this matter.

Finally, we wish to thank the reviewer for bringing a mistake to our attention. Due to a technical error, the entropy in Figure 1b was presented in nats instead of bits, so the units were different from the units in Figure 2c. With the proper units, Figure 1b encompasses the range shown in Figure 2c (note, however, that a direct comparison with the empirical data is not informative here because of the additional variance introduced by fMRI as explained above). We have updated the figure in the manuscript, and hope this will convince the reviewer of the validity of the uncertainty values used in our simulations.

14. Fig 3.

- While their model predicts unimodal posteriors in low-noise conditions, the authors chose to decode posteriors during the fMRI experiment with 100% coherence motion stimuli. The uncertainty is the lowest in that condition, particularly for the long stimulus duration used (1.5 sec). The decoded bimodality thus seems inconsistent with their model prediction. One hypothesis for why they would decode bimodal posteriors at 100% coherence is that they do not decode an integrated motion direction representation which shape depends on motion streak and motion coherence strengths. They decode separate coexisting orientation and velocity responses to the motion stimuli. A second hypothesis is that the decoded peak at the direction opposite to the true motion direction reflects motion aftereffect [4].

We would like to bring the reviewer's attention to simulations presented in Supplement 1 (Fig. S17) which show that, when decoded from fMRI signals, the posterior distribution should, in fact, become bimodal when the stimulus is presented at 100% coherence. This is because fMRI adds additional noise to the neural responses; it is this added noise that results in bimodality at the level of the voxels. Similarly, the predictions of Fig. 1d, 1e were obtained while assuming additional noise due to the fMRI recordings. We now discuss this more prominently in the main text (p. 5).

We agree with the reviewer that area V1 likely represents the two cues separately (using e.g. orientation-tuned and velocity-tuned neurons, Hubel & Wiesel, 1962, J Physiol; An et al., 2012, J. Neurosci.). In fact, given known tuning properties of V1 neurons, it is quite likely that not V1, but rather areas downstream of V1 integrate the two signals, similar to what has been found for e.g. multisensory perception (e.g., Rohe & Noppeney, 2016, Curr. Bio.). Indeed, our simulations are based on this premise (see, e.g., Eqs. S6-7). We now discuss this in the text (p. 12).

Contrary to the suggestions of the reviewer, however, the presence of these orientation and velocity responses is not at all incompatible with our hypothesis – quite the contrary, we would even argue that it is further evidence for our model, as it shows that orientation-tuned neurons respond to motion streaks. We demonstrate this in Figure S10 and Supplement 1.

It is perhaps also important to mention here that we find a reliable positive relationship between the second peak of the decoded distribution (i.e., the one around 180°) and the observer's behavioral responses (see Fig 3c, which was predicted by Figure 1e). This finding suggests that the spatial orientation and velocity signals are integrated (presumably in downstream areas) at least before the decision is made.

Finally, the alternative hypothesis proposed by the reviewer does not hold, as also explained above. This is because the motion aftereffect hypothesis would predict stronger motion aftereffects with greater coherence (lower uncertainty), so greater bimodality in behavior for low levels of noise or uncertainty, which is opposite to what we find (see Figure 4).

- The authors claim that the decoded distributions are bimodal but do not use quantitative model comparison to demonstrate the bimodality of the decoded distribution within and across subjects (same as comment 9.a).

We appreciate the reviewer's thorough approach to data analysis and model fitting. However, as explained in our response to comment 9.a, we do in fact use a quantitative statistical model to demonstrate bimodality. Specifically, we computed the Watanabe–Akaike Information Criterion to compare between the unimodal and bimodal models when analyzing peak locations. We have updated the description of our statistical procedures to clarify this (p. 19).

- It is unclear why their decoded distributions are so smooth compared to [5]. Could bimodality be explained by noise or a noise filtering technique that they apply to the decoded distribution?

Please note that the posteriors shown in figure 3a are averaged across trials. Of course, when taken from single trials, the posteriors are much noisier (please see below for some examples). We have updated the figure legend to make this clear.

15. Fig 3c. It is not clear how statistically significant is the correlation, particularly the second peak. Also, for consistency the two panels of the corresponding figure in simulation should have the same legends (first peak and second peak, which are less confusing than the “peak at 0” and “peak at 180” legends).

We updated the legend for Figure 1 per the reviewer’s suggestion. Furthermore, “at 0” and “at 180” now reads “First peak (around 0)” and “Second peak (around 180)”. The statistical tests (including Bayes factors) are mentioned in the figure caption, as well as the main text.

16. (p7, l3): Contrary to the authors claim, decoding bimodal posterior does not reflect an “advantageous” estimation process if cues likelihood are not integrated multiplicatively according to the Bayesian rule, and particularly if they are not integrated at all. For example, choosing between the two with a fixed response bias would be particularly suboptimal.

Agreed. We have rephrased the text to state that the motion and orientation signals should be combined rather than just used. Given the context of this sentence, we believe it will be sufficiently clear that the combination process referred to here is a Bayesian one.

17. Fig. 4: The authors should have collected fMRI responses in their follow-up condition. That is the condition where the Bayesian model predicts stronger posterior bimodality. Comparing the posterior in that high-noise condition with the posterior in the low-noise condition (100% coherence) will provide crucial evidence in support with Bayesian integration: if the ratio of the two peak amplitudes changes when coherence decreases as prescribed by Bayesian theory, it strengthens the Bayesian hypothesis.

We thank the reviewer for the suggestion. Regrettably, however, we fail to see what theoretical question such an experiment would address. That is, decreasing coherence would merely increase the width of the underlying two likelihoods in visual cortex. This should result in a more bimodal decoded posterior at the level of the voxels, but would not demonstrate that downstream areas integrate the evidence via likelihood multiplication (as the underlying likelihoods themselves are unknown, we simply cannot test this prediction noninvasively in humans; see our response to comment 1, above). In other words, to the best of our knowledge, the suggested experimental manipulation would only serve to demonstrate that the decoded posterior reflects uncertainty – a conclusion we already reached via Figures 2b and 2c of our manuscript and in our previous work (van Bergen et al., 2015, *Nature Neuroscience*; van Bergen & Jehee, 2018, *NeuroImage*; Geurts et al., 2022, *Nature Human Behaviour*).

We would of course be more than happy to reconsider if the reviewer would clarify their rationale for this experiment.

18. Many supplementary figures are not of publishing quality.

We have uploaded high-resolution figures along with the manuscript for the revision. These can be obtained from the Nature Communications reviewer website.

References

- (1) Pasternak and Tadin, "Linking Neuronal Direction Selectivity to Perceptual Decisions About Visual Motion."
- (2) Adelson EH, Movshon JA. Phenomenal coherence of moving visual patterns. *Nature*. 1982;300:523–525.
- (3) Wilson HR, Ferrera VP, Yo C. A psychophysically motivated model for two-dimensional motion perception. *Vis Neurosci*. 1992.
- (4) Geisler WS. Motion streaks provide a spatial code for motion direction. *Nature*. 1999;400:65–69
- (5) van Bergen et al., "Sensory Uncertainty Decoded from Visual Cortex Predicts Behavior."
- (6) Ma, W., Beck, J., Latham, P. et al. Bayesian inference with probabilistic population codes. *Nat Neurosci* 9, 1432–1438 (2006). <https://doi.org/10.1038/nn1790>
- (7) Laquitaine, S., & Gardner, J. L. (2018). A switching observer for human perceptual estimation. *Neuron*, 97(2), 462-474.
- (8) Anstis, S., Verstraten, F. A., & Mather, G. (1998). The motion aftereffect. *Trends in Cognitive Sciences*,

2(3), 407 111–117.

(9) Bae, G. Y., & Luck, S. J. (2021). Perception of opposite -direction motion in random dot kinem atograms. 412 PsyArXiv. <https://doi.org/10.31234/osf.io/uf3vd>

(10) Apthorp et al., “Direct Evidence for Encoding of Motion Streaks in Human Visual Cortex.”

(11) van Bergen and Jehee, “Modeling Correlated Noise Is Necessary to Decode Uncertainty.”

(12) Gros, B. L., Blake, R., & Hiris, E. (1998). Anisotropies in visual motion perception: a fresh look. *JOSA A*, 15(8), 2003-2011.

Reviewer #2 (Remarks to the Author):

Re: Motion direction is represented as a bimodal probability distribution in the human visual cortex

In this article, the authors investigated the hypothesis that motion direction in the visual cortex is represented by integrating velocity signal with “orientation” signal (i.e., motion streak) resulting from a moving stimulus. The authors presented their hypothesis with detailed simulations of a Bayesian ideal observer model, and reported interesting empirical evidence from both fMRI and behavioral studies.

This article extended a previously proposed fMRI decoding method (Van Bergen et al., 2015; Van Bergen & Jehee, 2021) to the domain of motion. The authors showed that they can decode both participants’ perceptual estimates and uncertainty of the motion direction. The authors observed a bimodal posterior distribution in their decoding results, which they interpreted as a neural signature of the motion streak signal in the visual cortex. However, I believe more substantial evidence needs to be provided to show the behavioral relevance of this bimodality. As they currently stand, the connections between the “neural bimodality” in experiment 1 and the “behavioral bimodality” in experiment 2 are rather indirect. Additional experiments need to be conducted to show a direct correspondence between the behavioral and neural bimodality.

I will detail my considerations below:

When a motion stimulus (e.g., random dot motion) is presented, the authors hypothesize that it results in two streams of evidence: Velocity, which provides a unimodal likelihood over motion direction; and “temporal orientation”, which provides a bimodal likelihood over direction. Crucially though, the authors propose that they are combined in a Bayesian optimal way to form our percept of motion direction.

In the first experiment, the authors showed that the posterior distributions of motion decoded from the visual cortex are bimodal, with a smaller peak in the opposite direction. This is indeed evidence supporting the existence of something similar to “temporal orientation” in the visual cortex, which will result in bimodality. However, a tighter link to behavior needs to be established to demonstrate the behavioral relevance of this bimodality, and that subjects are indeed integrating velocity with temporal orientation. To this end, the authors showed that the location of both peaks in the bimodal posterior can predict behavioral error (Fig. 3c).

I am not sure if this is sufficient, as it seems possible that the correlation can be observed even when there is no integration. This could be the case due to a combination of many factors:

- 1) The noise is correlated between velocity and temporal orientation encoding. For example, due to shared retinal encoding noise (e.g., Angueyra & Rieke, 2013).*
- 2) The overall uncertainties (e.g., variance of the noise) in the encoding of both processes are jointly modulated by common factors, such as attention effect (e.g., Goris, Movshon & Simoncelli, 2014). They both are also subject to the oblique effect, and have lower uncertainty close to the cardinal direction.*
- 3) Lastly, correlated noise due to the BOLD signal itself can also introduce additional correlation in the*

fMRI decoding that was not due to integration.

The simulations in Fig. S6 partially address 1), but did not take 2) and 3) into account. Therefore, I am not sure if the observed correlation itself is sufficient to show that there is integration. On a side note, could the authors also provide some intuitions why they observed a negative correlation in some of the simulated conditions?

We thank the reviewer for the thoughtful response. We addressed the three points raised by the reviewer by simulating each scenario's predictions and testing these in the empirical data.

As confirmed by the reviewer, the simulations of Figure S6 address scenario 1. Please note that the neural populations that respond to motion streaks are likely very different from the ones that respond to motion direction (e.g., Geisler, 2001; Gur & Snodderly, 2007; An et al., 2012, 2014), as also pointed out by reviewer 4, suggesting that only a small fraction of noise is shared due to common retinal factors. We now mention this in the text (p. 2 l. 1, p. 11 l. 7).

As to the remaining points 2 & 3: We simulated (2) uncertainty values that are correlated between orientation and velocity, and (3) noise that is correlated at the level of fMRI voxels but independent from underlying neural activity. Interestingly, we observe that neither type of correlation can explain the observed results. Specifically, when the uncertainty values are correlated, the model predictions remain the same regardless of the correlation level, indicating that joint variations in orientation and velocity uncertainty are not important for our conclusions. When fMRI noise is correlated, the predictions remain largely the same, except when the correlation in fMRI noise is close to one (i.e., fMRI noise is fully identical for the velocity and orientation signals, which seems rather unrealistic for real data). So, all in all, it seems unlikely that these additional factors could underlie the observed correlation between the peak locations and the behavioral errors.

Importantly, the only model that can predict the observed positive relationship between the second peak of the decoded distribution and behavior, is the Bayesian model that combines the velocity and orientation signals. This suggests that human observers use both sources of evidence to estimate direction of motion.

We have included these simulations and their discussion in the text (p. 11, Figure S7, S8).

Regarding the side note: the negative relationship arises because voxels in the visual cortex reflect a combination of velocity and orientation signals. When these signals are combined, the two peaks of the orientation likelihood are `pulled` towards the peak of the velocity likelihood in the resulting posterior distribution. This creates a negative circular correlation between the location of the posterior's second peak and the peak location of the velocity likelihood. For example, when the velocity peak shifts clockwise relative to the true stimulus, the second posterior peak shifts towards it, counterclockwise. When the observer's behavioral response is based on velocity signals alone, this behavioral response then also has this negative correlation with the second peak of the integrated posterior. We now describe this in the text (p. 5 l. 46; see also Figure R4 below).

Figure R4. The velocity peak location is negatively correlated with the location of the second peak in the decoded posterior distribution. When orientation and motion signals are combined, the two peaks of the orientation likelihood are ‘pulled’ towards the peak of the velocity likelihood in the resulting posterior distribution. This creates a negative circular correlation between the location of the posterior’s second peak and the peak location of the velocity likelihood. For example, when the velocity peak shifts counterclockwise relative to the true stimulus (left panel), the second posterior peak shifts towards it, clockwise. When the velocity peak shifts clockwise relative to the true stimulus (right panel) the second posterior peak again shifts towards it, which means that it shifts counterclockwise relative to the opposite motion direction. When the observer’s behavioral response is based on velocity signals alone, this behavioral response then also has this negative correlation with the second peak of the integrated posterior.

Perhaps equally importantly, since the motion coherence of the stimuli is 100% in the fMRI experiment, the resulting velocity likelihood should be very tight. Thus, one would expect the posterior to be unimodal (as shown by the authors in Fig S7, right column). I assume this can be tested by simulating the Bayesian observer with the noise level set to match the behavioral performance of subjects. If that is the case, why should we expect to see bimodality in the posterior decoded from voxel activity in the first place?

The reviewer is correct that, when estimated directly from neural responses alone, the likelihood should be very narrow for 100% motion coherence stimuli – however, this is not what happens with fMRI. That is, the BOLD signal also reflects non-neuronal sources of noise (e.g., fMRI scanner noise and various physiological sources of noise), and these additional forms of noise substantially broaden the likelihoods. We have presented simulations to illustrate this in Supplement 1. These simulations show that when fMRI noise is added to neural activity, the decoded posterior becomes bimodal even when the stimulus is presented at 100% motion coherence (see Figure S17, S18).

To facilitate a direct comparison between the modelled predictions and the posterior distribution decoded from the empirical data, we included this fMRI noise in our simulations of Figure 1d, 1e and S6 (see Methods, Ideal observer models, Simulations, p. 24).

We apologize for the confusion and have extensively revised both text and figure captions to clarify this (e.g., p. 5).

The authors did demonstrate a behavioral bimodality in the second experiment. It would be a much stronger result, for example, if on a trial-by-trial basis, the reverse in the perceived motion direction in experiment 2 can be predicted from the voxel activity. For example, experiment 1 could have been conducted using stimuli with a lower motion coherence, which would allow for analyses along this line. But as they currently stand, the connections between experiment 1 and experiment 2 seem weak.

We thank the reviewer for the interesting suggestion. To address this comment, we started by simulating decoded posterior distributions and behavioral responses for low-motion-coherence stimuli to see what this scenario would predict.

Specifically, we first fitted the Bayesian model to the behavioral estimates of the human observers in experiment 2 (low coherence stimuli); this gave us an estimate of the noise parameters of the underlying orientation and velocity likelihoods. We additionally estimated the amount of fMRI noise in experiment 1 by fitting the model's parameters to the decoded posteriors and the observers' behavioral estimates in this experiment. We then used these values in our simulations, following the same procedures as discussed in the text (see Methods, 'Bayesian observer models'). We simulated both the decoded posterior distribution and the observer's behavioral response on 1,800,000 trials (100,000 trials for each of 18 observers).

We divided these simulated trials over two bins: one for which the largest peak of the decoded posterior fell 'opposite' to the presented direction of motion (i.e., 170-190 degrees away from the presented direction of motion), and another bin for all the other trials. Within each bin, we then calculated the fraction of trials on which the model observer gave the 'opposite' behavioral response.

We did this for both the velocity-only model and the Bayesian model that combines the two likelihoods.

The results are presented in the table below. While the Bayesian model predicts a larger fraction of opposite behavioral responses than the velocity-only model, both models predict that there is a relationship between the opposite peak and the behavioral response. Interestingly, the increase in the fraction of opposite behavioral responses is even larger for the velocity-only model (0.032/0.004 vs. 0.136/0.023, see below).

Decoded posterior	Velocity-only	Combined
1: largest peak elsewhere	0.004	0.023
2: largest peak opposite	0.032	0.136

In other words, this analysis won't enable us to further adjudicate between the two models.

The reason why both models predict a relationship with behavior is that the velocity-only observer also sometimes makes large errors (i.e., when the MAP falls on the opposite direction, simply because of noise), especially when coherence levels are low, and uncertainty is high.

Given these results, we are not quite sure what theoretical question the suggested analysis would address, but we would be happy to reconsider if the reviewer disagrees and would clarify this.

Some other minor comments:

1) The authors cite previous literature regarding the idea of “motion streak”, but it was rather brief (page 1, line 39 - 41). Since this idea is central to the paper, more background and a summary of previous work should be provided for the relevant context of the current study.

We thank the reviewer for the suggestion, and now extensively discuss previous results on motion streaks in the Discussion section (p. 11).

2) In the caption of Fig. 1, the authors used the terms “neural bimodality” and “noise bimodality”, but it was not obvious why they are defined that way.

We are sorry to hear this was unclear. We have now changed the labels to ‘MAP readout’ and ‘Velocity-only readout’, and we hope this will make the figure clearer.

3) In Fig. 2c, is the effect of motion direction regressed out from the plotted behavioral variability (y-axis)? The caption mentioned “controlling for motion direction”.

Yes, that is correct. We have changed the text to make this clear.

4) One might expect the “velocity” and “motion streak” information to be encoded in different visual areas. In that case, the likelihood should be unimodal in some areas, and highly bimodal in some other areas. Was there any evidence regarding such a separation?

As Figure S3 shows, we observe bimodality in all areas. Because the stimulus-to-noise ratio likely differs between the areas (which will affect the degree to which bimodality is observed, see Figure S18), it is difficult to directly compare between the cortical areas. That said, it does seem likely that all of the tested areas should show at least some degree of bimodality, as they all contain both orientation and velocity sensitive neurons (e.g., Albright, 1984; An et al., 2012). We now discuss this in the text (p. 12).

Reviewer #3 (Remarks to the Author):

Review for the manuscript entitled “Motion direction is represented as a bimodal probability distribution in the human visual cortex” by Andrey Chetverikov, Janneke Jehee

This study combines computational modeling, behavioral and fMRI experiments to investigate the internal representation of visual motion in the human visual cortex, as well as its behavioral consequences. Using computational modeling, the authors found that the representation of motion should be bimodal. This bimodality should be reflected in the averaged posterior, the trial-to-trial likelihood function, and the overall behavioral error distribution. They then tested the model predictions using fMRI experiments. Leveraging Bayesian decoding techniques, the authors found evidence supporting the bimodality of the internal representation of motion direction both at the trial-averaged level and the individual-trial level. A follow-up behavioral experiment confirmed a further prediction of the modeling, i.e., the behavior estimates of a given motion direction should follow a bimodal distribution when the noise is large.

Overall, this study is well executed. The results are impressive and are presented in a logical fashion. It is relatively easy for the readers to follow the study. I enjoyed reading this paper and consider it a useful contribution. The paper will likely be of substantial interest to many in the field of cognitive and computational neuroscience.

We thank the reviewer for the kind words.

Having said that, I do have a number of concerns that I hope the authors could address or clarify in a revision.

1. My most serious concern is about the removal of motion biases in the analysis. Fig. S8 explained the procedure for removing these biases. I have multiple concerns. First, the procedure seems rather post-hoc and arbitrary. What’s the rationale for removing this bias? The bias is an important and integrative part of the behavior error that contributes to the trial-by-trial error. Given the well-known bias-variance tradeoff, removing the bias may cause issues in the interpretation of the variance of the residual errors. If the bias was not removed, would that change any of the results in the paper? This would be important to know in order to give a more faithful interpretation of the data.

The biases are removed for several reasons. First, when analyzing trial-by-trial changes in uncertainty (Fig. 2c), we remove the biases so as to estimate trial-by-trial fluctuations in uncertainty when the effect of motion direction is controlled for (i.e., we were worried that our results could otherwise be driven by direction-dependent effects, as both decoded uncertainty and behavioral variability change as a function of motion direction, see Fig. 2b). Please note that the same approach was used in previous work (e.g. van Bergen et al., 2015; Geurts et al, 2022). The trial-by-trial correlation with uncertainty remains significant (i.e., the effect of uncertainty (entropy) on behavioral variability is above zero)

without correction, $b = 5.05$, 95% HPDI = [0.15, 9.98], $BF = 7.22$ (compared to $b = 5.93$, 95% HPDI = [1.98, 9.96], $BF = 157$ after correction; the latter result is reported in the paper).

Second, we removed biases in the behavioral estimates (Figure S12) so as to accurately estimate behavioral variability. Because the behavioral response distribution is clearly not centered at 0 for some directions of motions (and for some observers, see Figure S12), we could have artificially inflated our estimates of behavioral variability and induced an artificial link between behavioral variability and decoded uncertainty (Fig 2b), had we not corrected for these biases. To address the reviewer's concern, we nonetheless reran the analysis without correction for these direction-dependent biases and found similar results. That is, there clearly is a link between behavioral variability and decoded uncertainty, even when we do not correct for direction-dependent behavioral biases ($b = 8.93$, 95% HPDI = [3.13, 14.53], $BF = 7.78 \times 10^3$ vs. $b = 9.17$, 95% HPDI = [4.73, 13.65], $BF = 8.54 \times 10^5$ after the correction).

Third, direction-dependent biases in the behavioral estimates could similarly (artificially) inflate the relationship between peak location and behavioral error (Fig. 3c) if both are somehow linked to the motion direction of the presented stimulus. This is why we performed the analysis on bias-corrected data in the paper. To nonetheless address the reviewer's concern, we ran an additional analysis to ensure that the correction is not affecting our results. Indeed, our results were very similar between the two analyses: $r = 0.06$, $b = 0.63$, 95% HPDI = [0.40, 0.88] (corrected) vs. $r = 0.05$, $b = 0.57$, 95% HPDI = [0.33, 0.80] (raw) for the first peak, $r = 0.03$, $b = 0.30$, 95% HPDI = [0.10, 0.50] (corrected) vs. $r = 0.03$, $b = 0.27$, 95% HPDI = [0.07, 0.46] (raw) for the second peak.

We now also mention this in the text (p. 15).

The second question is more technical. Why was the estimated bias not a continuous function motion? Fig S8a shows the bias curve is discontinuous. Please clarify.

The biases are modeled as a combination of discontinuous functions because cardinal biases in orientation and motion perception are often repulsive in an estimation task (see e.g. van Bergen et al., 2015, Nature Neuroscience; Wei & Stocker, 2015, Nature Neuroscience). That is, the behavioral responses are pushed away from the cardinals, creating a gap in the function. Please see subject B in Figures S12 for an example of this. Fitting a continuous function here would underestimate the biases, which could contaminate the analyses (see above). We now clarify this in the text (p. 15).

2. The following statement needs to be toned down.

"More broadly, our findings indicate that the cortical representation of low-level visual features, such as motion direction, can be far more complex than generally appreciated."

It is generally appreciated that some of the V1 neurons are orientation-selective, while a portion of other neurons is direction-selective. Conceptually, some of the findings reported here could be interpreted as following from these electrophysiological observations. In addition, it would be useful to cite and discuss relevant neurophysiological studies on motion and direction selectivity in early visual processing (e.g., in V1).

We revised the sentence (abstract) and added a discussion of the relevant neurophysiological studies as suggested by the reviewer (p. 12).

3. The dependence of the velocity and spatial orientation signal needs an even more careful discussion. The authors have assumed the two to be independent in the modeling while acknowledging that the two should be dependent in reality. It would be useful if the authors could carry out additional simulations to demonstrate how to deal with the dependence between the two. Should the dependence be modeled as signal correlations or noise correlations? The authors seemed to indicate that they should be considered as noise correlations, but it also seems reasonable to think that they should be treated as signal correlations. A more careful treatment or discussion of these points in the context of the authors' model would be useful.

We thank the reviewer for the suggestion. In the context of our work, we define signal correlations and noise correlations as follows: signal correlations are described by the tuning curves of the voxels. If two voxels have identical tuning curves, their signal correlation is 1. Noise correlations are described by the voxel covariance structure. This structure describes the trial-by-trial response fluctuations that are independent of the stimulus (but still shared between voxels). For example, global fluctuations in cortical response across repeated presentations of the same stimulus are modelled by this voxel covariance structure.

Considering these definitions, we modeled the dependence of the velocity and spatial orientation signals as a noise correlation (and not a signal correlation), as it refers to trial-by-trial fluctuations in the response when the stimulus is held constant. This is how we obtained the predictions of Figures 1e and S6 (i.e., repeated presentations of the same stimulus, and only internal noise is allowed to vary), and it is also how we analyzed our data (i.e., we removed stimulus effects, resulting in errors and peak locations centered around 0).

Having said that, we have also now added additional analyses to show that the effect of correlated fMRI noise and correlated variation in neural noise (Figure S7, S8) cannot explain our results. We now discuss this in the text (p. 11).

4. The results in the paper assumed a MAP estimate when performing Bayesian inference. This should be made clear in Line 16 of page 5 of the main text. Also, would using a different Bayesian estimator such as the posterior mean substantially change the results?

We followed the reviewer suggestions and now mention the MAP estimate in the main text (line 10, p 4). As to the use of other estimators: for low-noise scenarios that result in a single-peak posterior, using a different estimator would not change the results. This is because MAP, mean, and similar estimators all select the peak of the distribution. For higher-noise scenarios where bimodality is observed, however, it would be somewhat problematic to take the mean of the distribution, because the mean of a bimodal distribution falls in between the two peaks. Thus, although this strategy does minimize the

squared error cost function, it would nonetheless lead to a paradoxical situation in which the true stimulus would never be chosen as a response. Following this line of reasoning, we settled on using the MAP as an estimator. We now more thoroughly discuss this in the text (p. 22).

Other comments:

*** Fig. S3c shows that the effect is not significant for the individual brain region. This should be mentioned in the main text. Furthermore, please indicate the effect size, R^2 , or the p-value in Fig 3c. More*

We have changed the text to indicate that the effect is significant only when the regions are combined. We have included the appropriate statistic in the figure's caption (we use Bayesian statistics for our analyses, so this is a BF or HPDI intervals and not a p-value). The effect size for the regression model is indicated by the regression coefficient, b , which directly reflects how much the dependent variable is changed by a unit step in the independent variable.

*** Line 30 of page 5. Why the difference in the entropy of the decoder between the cardinal and oblique directions is much smaller than the difference in precision? This needs to be addressed. If this means that there is substantial noise in the decoding, it should be stated so.*

Please note that the difference in behavioral variability (3.2 deg.) is rather comparable to the difference in entropy. That is, for a univariate Gaussian, a change in entropy from 7.40 to 7.49 bits reflects a 3.1 deg. increase in sigma.

*** It would be useful to report a scatter plot showing the observers' behavior in fMRI experiment (similar to Fig 4a). One possibility would be to place it in Fig 2. Another possibility is to make it a SI figure.*

We have included a scatter plot of the behavioral responses of one example observer, along with a probability density plot of the response distribution (computed across observers), in the supplement (Fig S16).

***Line 31 & 39 of page 5. What does variable b mean in these places?*

The variable b is the regression coefficient. It captures the change in the dependent variable with a unit step in the independent variable. We have added this to the text (p. 7, line 15).

***SI part 2. For the equation in Line 14, it would be helpful if the meaning of each variable could be explicitly defined.*

We have added a description of the variables.

*** Fig S6, the explanation between lines 7- 20 is difficult to follow. Please get this better organized.*

We have clarified the explanation of this figure in the text.

*** In panel b of Fig S4, please change the color scheme so that the color changes gradually with the change of the number of voxels.*

We have changed the color scheme accordingly.

Reviewer #4 (Remarks to the Author):

There have been a number of studies over the past two decades directed at the hypothesis that spatial orientation signals (motion streaks) created by temporal integration in the early visual system provide orientation cues that are used in estimating motion direction, in addition to the more standard velocity cues extracted by cortical-neuron receptive fields that are oriented in space-time. Past studies have shown clearly that motion-streak cues are a highly useful source of motion-direction information, especially at higher retina speeds, and that motion-streak signals exist in the visual cortex of human and non-human primates. While it is highly likely that evolution would exploit such a useful source of information, it has not been convincingly demonstrated that motion-streak cues are actually combined with velocity cues in the human visual system to estimate motion direction. By measuring and analyzing human fMRI and behavioral responses to moving dot stimuli, this paper provides stronger evidence that the motion-streak and velocity cues are combined to estimate motion direction. The paper also uses the Bayesian cue-combination framework to model motion direction perception, which I also think has not been done before. Thus, I am quite positive. The paper is also clearly written. Nonetheless, I think there are some ways in which the paper could be improved.

1. In the initial modeling section, the authors assume that the motion streak and velocity cues are statistically independent, which is unlikely to hold exactly. The authors discuss this issue in the Discussion section, but something should be said up front, especially since a case can be made that they may be largely independent (e.g., the neurons that respond best to motion streaks are largely different from those that respond best to the sign of motion direction; although there are neurons that are sign selective for motion parallel to the receptive field orientation; Geisler et al. 2001).

We thank the reviewer for the suggestion, and now mention this on p. 3, with reference to the relevant literature on neural tuning, including the paper mentioned by the reviewer.

2. The authors also assume a flat prior on motion direction. This is likely true for the sign of the motion direction, but not for the orientation. Adding a more realistic prior would not affect the conclusions of the paper. Again, something should be said about this up front.

We now mention the prior and its effects on our predictions when discussing the models (p. 3). Indeed, as noted by the review, this would not change the conclusions of the paper.

3. I think a slightly different Bayesian formulation would be sensible to consider, although I have not tried to see if it is equivalent. Specifically, it makes sense to estimate motion direction as an estimate in the range of 0-180 (th), and separately estimate the sign of the motion direction (k), because the two cues (v and o) have different relative reliabilities for the 0-180 direction and for the sign of the direction. I will represent the estimates as th^{\wedge} and k^{\wedge} . For independent cues, the two estimates are: $th^{\wedge} = (r_{-vth} \times th^{\wedge v} + r_{-oth} \times th^{\wedge o}) / (r_{-vth} + r_{-oth})$ and $k^{\wedge} = (r_{-vk} \times k^{\wedge v} + r_{-ok} \times k^{\wedge o}) / (r_{-vk} + r_{-ok})$. (The r-subscript

variables represent reliabilities). The reliability of the sign estimation for the streak signal is 0, so $k^{\wedge} = k^{\wedge}v$. The MAP estimate of the direction between 0 and 360 is thus $th^{\wedge} + 180$ if $k^{\wedge} = -1$, else it is th^{\wedge} if $k^{\wedge} = 1$. It might be best to estimate th^{\wedge} first, and then compare that estimate to $th^{\wedge}-v$ to get the sign. This is a relative minor modeling difference, but I think it would be worth checking out. It might be conceptually simpler for the readers.

We thank the reviewer for the suggestion. If we understand correctly, the suggested model disambiguates the orientation signal using the velocity-based sign measurement ($k^{\wedge}v$ in the equations above) and then the disambiguated orientation signal is combined with the motion signal. However, it is not quite clear how the variable $k^{\wedge}v$ is computed; specifically, what determines the 0 point in motion space. If any fixed boundary is used (say, 0 corresponds to horizontal motion), then a stimulus at that boundary would generate uncertain $k = 0.5$ (since approximately one half of a likelihood would be on one side of the boundary, and the other half would be on the other side). This would in turn give bimodal posteriors even when all likelihoods for theta are precise. A flexible boundary would have to be tied to the velocity-based likelihood in the 360 space (as the true stimulus is unavailable to the observer). The ML estimates of this model are the same as for our Bayesian model if the peaks of the posterior (in our model) are well-separated (this strategy is similar to the two-step strategy discussed in the text, p. 23). If the peaks are not well-separated (that is, if motion uncertainty is low compared to orientation uncertainty), then the estimates will be different as the MAP response of the Bayesian observer will be influenced by the second peak. We now also discuss the suggested model in the text on p. 22.

4. The authors should make it clear to the reader that the relative reliability of the motion-streak signals to the velocity signals change with feature speed and feature size. They should state clearly that the results reported depend on the specific dot size and on the chosen speed of 7 deg/s. A future direction of study would be to vary dot speed and see how the relative reliability of the two cues in the fMRI responses varies and whether behavior follows along.

We have made clear in the text that the reported results depend on dot size and chosen speed (p. 6).

We thank the reviewer for suggesting this interesting direction for future research. In addressing this suggestion, we believe it might be helpful to start with a clarification. As the reviewer may know, fMRI voxels reflect the combined responses from many different populations of neurons. Because their individual signals are unknown (and cannot be deduced from voxel activity), the likelihood associated with each individual population alone cannot be computed (although it is possible to calculate a likelihood from the combined response, as we show in the paper). This makes it impossible to predict what the observer's behavioral response should be for the optimal integration strategy, as well as for alternative strategies that ignore uncertainty. We now also discuss it in the text on p. 12.

Thus, while we agree that manipulating the relative reliability of the two cues would be incredibly interesting and worthwhile if we had access to the underlying neural populations and likelihoods, given

the limitations of fMRI, it is currently unclear how this manipulation would enable us to further test the Bayesian model noninvasively in human cortex.

We would very much appreciate hearing the reviewer's response to this, should we have misunderstood or overlooked something.

5. The chosen dot speed (7 deg/s) is a sensible choice, because it will produce both strong motion-streak and velocity signals in visual cortex. It would strengthen the paper to talk about what the study means for perception in the real world. By my calculation, if a person with eyes 5 feet above the ground plane is walking at 3 ft/sec (2 miles per hour), the optic flow in the ground plane will be 7 deg/s at about 24 ft to the left and right and at about 12 ft to the front. Ground plane features that are closer will have even greater speed (and greater relative reliability for the motion-streak signals). This, would make the case that the experimental stimuli are highly relevant for the kinds of optical flow signals that exist under real world conditions. I recommend double checking these calculations or carrying out similar calculations and including them in the paper.

We thank the reviewer for the interesting and highly relevant suggestion. Our calculations give similar results: A person walking at 1.4 m/s (about 3 miles per hour, corresponding to the average walking speed) and with eyes 1.5 m above the ground (about 5 ft), looking straight ahead, will have an optic flow speed of 7 deg/s at 4.8 m (about 16 ft) for a point straight in front of them or at 6.4 m for a point 5 meter to the left or right. We now include this in the text along with a discussion (p. 12).

6. It is a very nice result that motion direction over 360 degrees can be decoded from the fMRI responses.

We thank the reviewer for the kind words. We now highlight this finding in the discussion (p. 10).

7. The demonstration that humans make judgements of motion direction opposite to the true direction in high noise is also a very nice result demonstrating that motion streaks are actually used in estimation motion direction. I did not see if subjects were given feedback. Probably better if not. It would be good to state explicitly whether they were or were not.

Our participants did not receive trial-by-trial feedback about their performance, and we now mention this in the text.

8. That motion direction responses have a bimodal distribution in visual cortex, with a higher peak in the direction of the correct sign, is not at all surprising. That these correlate with behavior, even on a trial-by-trial basis is an important finding that provides more direct evidence that motion streaks play a

functional role in motion perception/estimation and are not an epiphenomenon of temporal integration in the early visual system (retina).

We agree with the reviewer's assessment and now highlight the relationship between cortical responses and behavior, along with its implications, in the text (p. 10).

9. If one could record from the whole population of V1 neurons with orientation preference perpendicular to the direction of motion, one would find that the population response those cells (the velocity-cue cells) would also have a bimodal population response with a weaker peak in the opposite direction (at least for gratings). This is because most V1 neurons have a second smaller peak in the opposite to the preferred direction. This might not hold for dot patterns but I am not sure it has been tested. Might be worth discussion this a bit.

We thank the reviewer for bringing these bimodally-tuned direction selective cells to our attention. We ran simulations to see if their tuning properties could potentially explain our results. Specifically, using a realistic distribution of direction selectivity indices (taken from Gur & Snodderly, 2007, Fig. 5, using DI (direction index) > 0.5, as defined by the paper's authors), we simulated fMRI voxel responses for moving dot stimuli, decoded the posterior distribution of motion direction given each pattern of voxel responses, and averaged across trials. We found that the decoded posterior was always unimodal and never bimodal (see Figure S19). This further strengthens the hypothesis that the empirically observed posteriors reflect the responses of cells whose spatial orientation receptive field runs parallel to the presented motion direction. We now discuss this in the text (p. 12, Fig. S19).

10. p. 9 l. 4 Is it 18% or 30%. The methods indicates 30%.

Fixed. 18%.

11. p. 10 l. 14 "This strongly suggests that the observer's behavioral choices are based on the decoded neural representations." Does it? Can you predict the trial-by-trial from just thv? Is the prediction better for both cues together?

We have revised the sentence in question. Unfortunately, we cannot compute the velocity and orientation likelihoods as the individual neural signals for each cue cannot be extracted from the voxel response (see also our response above), but we can conclude that both peaks in the decoded posterior are related to the observer's behavioral choices and that the decoded posterior is therefore behaviorally relevant (as we believe the reviewer also agrees based on comment 8). We now discuss it in the text (p. 12).

12. p. 10 l. 46 “Altogether, this strongly suggests that the human visual system uses spatial orientation signals for determining the direction of moving objects, and reveals the hidden complexity of probabilistic feature representations in cortex.” The version of the Bayesian model I suggested above doesn’t really have a complex probabilistic feature representation.

We toned down the paragraph.

13. There is some relevant literature that should be included and perhaps discussed (see following list). One important one is a recent study (Tohmi M et al., 2021) showing that in the mouse there are populations of neurons with velocity selectivity that peaks at high speeds in the direction parallel to the static orientation tuning. This is pretty strong evidence that motion streak signals are used for encoding and decoding in the mouse cortex. These motion selective cells are most selective to streak signals and would not be there if they were not being used for motion direction coding. Because of the low spatial resolution of the mouse visual system, only large features are encoded and hence motion streaks only occur at high speeds. Thus, the streak cells in mouse are tuned to much higher speeds. But given their cruising speed and low eye height from the ground, the streak cells still encode optic flow signals over a substantial ground-plane region around the cruising mouse.

Apthorp, D., Cass, J., & Alais, D. (2011). The spatial tuning of “motion streak” mechanisms revealed by masking and adaptation. *Journal of Vision*, 11(7):17, 1–16,
<http://www.journalofvision.org/content/11/7/17>, doi:10.1167/11.7.17.

Apthorp D, Schwarzkopf DS, Kaul C, Bahrami B, Alais D, Rees G. (2013) Direct evidence for encoding of motion streaks in human visual cortex. *Proc R Soc B* 280: 20122339.
<http://dx.doi.org/10.1098/rspb.2012.2339>

Barlow HB and Olshausen BA (2004) Convergent evidence for the visual analysis of optic flow through anisotropic attenuation of high spatial frequencies. *Journal of Vision* 4, 415-426.

Burr, D., and Thompson, P. (2011). Motion psychophysics: 1985-2010. *Vision Res.* 51, 1431–296 1456.

Rasch, M.J., Chen, M., Wu, S., Lu, H.D., and Roe, A.W. (2013). Quantitative inference of population response properties across eccentricity from motion-induced maps in macaque V1. *J. Neurophysiol.* 109, 1233–1249.

Tohmi M, Tanabe S & Cang J (2021) Motion streak neurons in mouse visual cortex, *Cell Reports* 34, 108617.

We thank the reviewer for pointing us to these highly relevant papers, which we now discuss in the text (p. 11-12).

Just for fun, I attached a photo I just took a couple of days ago, here is Colorado. The camera has a pretty fast shutter but one still gets motion streaks for the snowflakes, which are like the random dots in the authors experiment. One can see how the streaks convey the direction of motion. Feel free to use this if you want.

We thank the reviewer very much for sharing the photo. We agree that this is a great illustration of motion streaks and are happy to include it in the manuscript (Fig 1).

Bill Geisler

REVIEWER COMMENTS

Reviewer #1 (Remarks to the Author):

Overall, I want to thank the authors for having substantially improved the clarity of the manuscript. I believe that they have convincingly demonstrated that motion direction representations in visual cortices are bimodal and are linked with the motion direction estimate bimodality. The authors have additionally made a commendable effort to test whether the observed bimodality results from a cue combination mechanism (optimal inference or not), or alternative mechanisms that completely forgo combination (e.g., binary decision process).

I have only a few remaining comments:

Novelty: I agree with the authors that their work presents the strongest evidence to date that not only velocity signals, but also spatial orientation signals contribute to the representation of motion direction in visual cortex; they have demonstrate that the representation of motion direction is bimodal in early and downstream visual cortical areas (Figure S3) and have linked that decoded representation with motion direction estimation behavior.

fMRI confounds: It is indeed well-known that scanner and physiological noise contribute to the BOLD signal but I am appreciative for the authors additional effort to explicitly link these remaining sources of noise (captured in a quantitative generative model in van Bergen et al, Nature Neuroscience 2015) to the bimodality discovered in their study. I also thank the authors for clarifying the fMRI design and analytical steps, identical to the previous studies. These steps were not entirely clear to me in the method section of their first draft of the paper.

Quantitative analyses: The authors have clarified in the text and providing convincing arguments for their choice of distinct quantitative approaches to assess the shape of their posteriors from the simulations and the voxel responses.

Alternative hypotheses that forgo cue combination: While I agree that cue combination (optimal or not) provides a good description of their data. I am not convinced that they have ruled out non-integrating hypotheses. I think that I have an issue here with how the authors interpret their data. In Laquitaine and Gardner (2018), the probability to switch is not fixed, contrary to how the authors implemented the model. In the original paper, switching actually depends on the ratio of the reliabilities of stimulus and prior. Applied to each cue in the current study, the model predicts that when velocity becomes very reliable, estimates near the first peak (0 degree, velocity peak) will become more probable than estimates near the second peak (180 degree, opposite orientation peak). In the extreme case, with virtually no uncertainty in velocity, estimate distributions become unimodal with a peak at 0 degree, as prescribed by their Bayesian MAP model (Fig S14), and accounting for their data, including the positive relationship between neural and behavioral uncertainty. I believe that cues never need to be "combined" into a single representation, contrary to what is stated by the authors, on several occasion in the text. They can coexist in separate visual cortical populations within each area, without ever being integrated (which I would say is supported by Figure S3), and be fed to e.g., the lateral intraparietal area (LIP) for a decision readout (a binary decision process, not a continuous inference process). That "segregated representations" model offers a simpler and computationally cheaper explanation to the authors' data. Therefore, I must disagree with statements such as "While our data suggest that observers combine velocity and orientation cues when inferring motion direction," (p12, l11).

I do agree though that the authors have demonstrated that the decoded representations reflect both motion and orientation signals that both coexist in visual cortical areas and that they have linked these signals to behavior. I also concur with the authors that a motion aftereffect explanation can be ruled out as it would produce greater bimodality in behavior with low uncertainty which does not explain their data. The constant bias model can also be ruled out.

Link between neural and behavioral variability: I thank the authors for their clear explanation of Figure 1b. They have very clearly shown that behavioral variability is only explained by random fluctuations around the average estimate here and not by systematic biases in their model.

Reviewer #2 (Remarks to the Author):

I appreciate the detailed response from the authors. They have now fully addressed the main concern in my original review.

I understood the authors' argument as follows:

The second peak in the decoded posterior (from the motion streak) is going to be correlated with behavior *only if* the behavioral estimate reflects the integration of velocity encoding and motion streak, unless the noise in the two processes is extremely correlated, which is unlikely.

Otherwise, the second peak actually tends to be *negatively* correlated with behavior, due to the effect of attraction towards the primary peak.

Thus, I now agree that the positive correlation observed in the data indeed supports the idea that there is integration.

I do have a minor follow up regarding the comment on experiment 2. My intention here is not to ask for any additional experiments, but to clarify some conceptual questions about the bimodality. Maybe the authors could simply add a sentence or two in the experiment 2 section to discuss this.

I agree that the relationship between the opposite peak and the behavior will be present for either low- or high-coherence stimuli. My original thought was that, if both the behavior and the decoding are bimodal, then maybe it is possible to show some stronger form of correspondence between the two (e.g., showing the decoder can predict which trial the behavioral report is on the opposite side)?

I might not have fully understood the authors' response here. For an observer model that uses velocity-only decoding, shouldn't be much more difficult to predict the opposite peak response using decoded posterior, as the observer will only produce a unimodal response, no matter how uncertain the stimulus is?

Reviewer #3 (Remarks to the Author):

I would like to thank the authors for their responses to my critiques. My concerns were addressed. The additional analyses and clarifications in the revised version have substantially improved the paper.

This work will be a useful contribution. I support the publication of this paper.

Reviewer #4 (Remarks to the Author):

The authors have done an admirable job of responding to the extensive comments from all the reviewers. The revisions have substantially improved the paper. I do not have much more to add except a general comment about how the results might transfer to more complex tasks. Specifically, the Bayesian model with independent signals makes perfect sense for the random-dot motion

estimation task, but it will not work easily for real-world motion estimation, where there are irrelevant static orientation signals everywhere. The use of orientation signals for motion perception at a given image location must be gated in some way by the simultaneous encoding of motion at that location. This fact is why a multiplicative interaction was postulated in Geisler (1999). Basically, the probability that a strong orientation response at some location corresponds to some motion direction becomes zero if there is no simultaneous motion response at that location. It might be good to mention in the discussion that a more realistic Bayesian model will have to take this into account. Nonetheless, this does not detract from the value of this study. The fMRI results and the fact that humans show 180-deg errors in motion direction estimation at low signal strength seems to be strong evidence that humans are using the orientation information to estimate motion direction.

We are glad that all reviewers found our study of interest, and thank them for their constructive comments, which have been very valuable to us in further strengthening the manuscript. In what follows, we respond to the remaining comments:

REVIEWER COMMENTS

Reviewer #1 (Remarks to the Author):

Overall, I want to thank the authors for having substantially improved the clarity of the manuscript. I believe that they have convincingly demonstrated that motion direction representations in visual cortices are bimodal and are linked with the motion direction estimate bimodality. The authors have additionally made a commendable effort to test whether the observed bimodality results from a cue combination mechanism (optimal inference or not), or alternative mechanisms that completely forgo combination (e.g., binary decision process).

I have only a few remaining comments:

Novelty: I agree with the authors that their work presents the strongest evidence to date that not only velocity signals, but also spatial orientation signals contribute to the representation of motion direction in visual cortex; they have demonstrated that the representation of motion direction is bimodal in early and downstream visual cortical areas (Figure S3) and have linked that decoded representation with motion direction estimation behavior.

fMRI confounds: It is indeed well-known that scanner and physiological noise contribute to the BOLD signal but I am appreciative for the authors additional effort to explicitly link these remaining sources of noise (captured in a quantitative generative model in van Bergen et al, Nature Neuroscience 2015) to the bimodality discovered in their study. I also thank the authors for clarifying the fMRI design and analytical steps, identical to the previous studies. These steps were not entirely clear to me in the method section of their first draft of the paper.

Quantitative analyses: The authors have clarified in the text and providing convincing arguments for their choice of distinct quantitative approaches to assess the shape of their posteriors from the simulations and the voxel responses.

Alternative hypotheses that forgo cue combination: While I agree that cue combination (optimal or not) provides a good description of their data. I am not convinced that they have ruled out non-integrating hypotheses. I think that I have an issue here with how the authors interpret their data. In Laquitaine and Gardner (2018), the probability to switch is not fixed, contrary to how the authors implemented the model. In the original paper, switching actually depends on the ratio of the reliabilities of stimulus and prior. Applied to each cue in the current study, the model predicts that when velocity becomes very reliable, estimates near the first peak (0 degree, velocity peak) will become more probable than estimates near the second peak (180 degree, opposite orientation peak). In the extreme case, with virtually no uncertainty in velocity, estimate distributions become unimodal with a peak at 0 degree, as prescribed by their Bayesian MAP model (Fig S14), and accounting for their data, including the positive relationship between neural and behavioral uncertainty. I believe that cues never need to be “combined” into a single representation, contrary to what is stated by the

authors, on several occasion in the text. They can coexist in separate visual cortical populations within each area, without ever being integrated (which I would say is supported by Figure S3), and be fed to e.g., the lateral intraparietal area (LIP) for a decision readout (a binary decision process, not a continuous inference process). That “segregated representations” model offers a simpler and computationally cheaper explanation to the authors’ data. Therefore, I must disagree with statements such as “While our data suggest that observers combine velocity and orientation cues when inferring motion direction,” (p12, l11).

We appreciate the reviewer’s strong focus on model comparison and thank them for bringing the disparity between the models to our attention. We have now updated our simulations to incorporate a switch that depends on the reliability of the two cues, as proposed by the reviewer and modeled by Laquitaine and Gardner (2018). Contrary to the suggestions of the reviewer, however, this updated model cannot capture our findings. This is because our participants made hardly any opposite behavioral responses for the high coherence stimuli in our main fMRI study.

That is, when the switch probability is governed by the precision ratio as in Laquitaine and Gardner (2018), the precision of the velocity estimates should not just be ‘high’, but rather extremely high *relative* to the precision of orientation estimates so as to capture these behavioral data. In our experiment, the observers made 11 responses with errors above 90 degrees (out of the more than 14000 trials obtained across observers), so even if we take a very lenient threshold and count all of these responses as switches (and not attention lapses), the proportion of trials on which a switch occurred is less than 0.001. Following the equation on p. e3 in Laquitaine and Gardner (2018), the probability of using the orientation likelihood is given by:

$$p_{\text{orientation}} = \kappa_{\text{orientation}} / (\kappa_{\text{orientation}} + \kappa_{\text{velocity}}).$$

So, given a 0.001 proportion of switches, κ_{velocity} would have to be about 1000 times higher than $\kappa_{\text{orientation}}$ to capture the behavioral responses of the observers. This essentially means that the orientation likelihood would have to be flat (uniform) – in other words, the spatial orientation signals provide no information at all. This is inconsistent with our fMRI results, which clearly show that bimodal distributions are represented in cortex (as also acknowledged by the reviewer), and that the underlying spatial orientation signals are used by the observers in their estimates of motion direction (Fig. 3c). Please note that this scenario is also inconsistent with the results of the follow-up behavioral study, as an already flat orientation likelihood cannot give rise to a bimodal behavioral response distribution with greater levels of uncertainty.

The Bayesian observer model, on the other hand, is the only model that can capture all of these results. This is because the integration (multiplication) of the velocity and orientation likelihoods reduces the second peak in the posterior. Consequently, it does not give rise to opposite responses when the orientation and motion signals are similarly precise (as we show in our simulations, see e.g., Fig S14).

For convenience, we present in the two tables below the proportion of opposite behavioral responses for the two models as a function of velocity and orientation uncertainty (using the same parameter values as used in the paper). Note how for the uncertainty-guided switching observer, the proportion of switches only lies around 0.001 in extreme cases. As noted above, these extreme parameter values are inconsistent with the fMRI results, and moreover, would not result in bimodality when uncertainty increases even further due to the lower levels of motion coherence used in the follow-up behavioral experiment.

We now discuss the uncertainty-guided switching model in the manuscripts (Figure S14).

Table 1. The proportion of opposite (more than 90 deg. away from the true direction) responses for the switching observer model.

		Orientation uncertainty (SD, deg.)							
		3	5	10	20	30	40	60	100
Velocity uncertainty (SD, deg.)	3	0.250	0.132	0.041	0.012	0.006	0.003	0.002	0.001
	5	0.367	0.249	0.101	0.032	0.016	0.010	0.005	0.002
	10	0.455	0.399	0.251	0.105	0.056	0.037	0.021	0.007
	20	0.489	0.469	0.395	0.249	0.163	0.117	0.072	0.024
	30	0.494	0.485	0.446	0.339	0.254	0.198	0.130	0.054
	40	0.495	0.488	0.465	0.391	0.318	0.267	0.196	0.101
	60	0.498	0.496	0.486	0.448	0.407	0.376	0.320	0.227
	100	0.499	0.499	0.498	0.493	0.483	0.481	0.466	0.427

Table 2. The proportion of opposite (more than 90 deg. away from the true direction) responses for the Bayesian MAP observer.

		Orientation uncertainty (SD, deg.)							
		3	5	10	20	30	40	60	100
Velocity uncertainty (SD, deg.)	3	0.000	0.000	0.000	0.000	0.000	0.000	0.000	0.000
	5	0.000	0.000	0.000	0.000	0.000	0.000	0.000	0.000
	10	0.000	0.000	0.000	0.000	0.000	0.000	0.000	0.000
	20	0.000	0.000	0.000	0.001	0.000	0.000	0.000	0.000
	30	0.006	0.007	0.007	0.016	0.019	0.013	0.007	0.006
	40	0.035	0.035	0.038	0.049	0.062	0.059	0.039	0.034
	60	0.142	0.143	0.146	0.160	0.177	0.183	0.164	0.142
	100	0.361	0.360	0.363	0.368	0.378	0.384	0.387	0.362

I do agree though that the authors have demonstrated that the decoded representations reflect both motion and orientation signals that both coexist in visual cortical areas and that they have linked these signals to behavior. I also concur with the authors that a motion aftereffect explanation can be ruled out as it would produce greater bimodality in behavior with low uncertainty which does not explain their data. The constant bias model can also be ruled out.

Link between neural and behavioral variability: I thank the authors for their clear explanation of Figure 1b. They have very clearly shown that behavioral variability is only explained by random fluctuations around the average estimate here and not by systematic biases in their model.

Reviewer #2 (Remarks to the Author):

I appreciate the detailed response from the authors. They have now fully addressed the main concern in my original review.

I understood the authors' argument as follows:

*The second peak in the decoded posterior (from the motion streak) is going to be correlated with behavior *only if* the behavioral estimate reflects the integration of velocity encoding and motion streak, unless the noise in the two processes is extremely correlated, which is unlikely.*

*Otherwise, the second peak actually tends to be *negatively* correlated with behavior, due to the effect of attraction towards the primary peak.*

Thus, I now agree that the positive correlation observed in the data indeed supports the idea that there is integration.

I do have a minor follow up regarding the comment on experiment 2. My intention here is not to ask for any additional experiments, but to clarify some conceptual questions about the bimodality. Maybe the authors could simply add a sentence or two in the experiment 2 section to discuss this.

I agree that the relationship between the opposite peak and the behavior will be present for either low- or high-coherence stimuli. My original thought was that, if both the behavior and the decoding are bimodal, then maybe it is possible to show some stronger form of correspondence between the two (e.g., showing the decoder can predict which trial the behavioral report is on the opposite side)?

I might not have fully understood the authors' response here. For an observer model that uses velocity-only decoding, shouldn't be much more difficult to predict the opposite peak response using decoded posterior, as the observer will only produce a unimodal response, no matter how uncertain the stimulus is?

We apologize for the confusion. The reviewer is correct that the velocity-only observer bases their response on a unimodal likelihood. But this unimodal likelihood is not always centered on 0. That is, the velocity measurement fluctuates across trials due to random noise. Because the measurement fluctuates, so does the location of the likelihood's peak, and consequently, the MLE (i.e., velocity-only readout). The trial-by-trial variance in the MLE is directly related to the variance in the observer's measurements and the width (variance) of the likelihood (see e.g., Eq. 11, 28). Thus, when uncertainty is high and the likelihood is sufficiently broad, the peak of the distribution (and associated behavioral response) will sometimes fall on the opposite direction of motion due to random noise (even though the mean of the behavioral response distribution, which is computed across trials, is 0).

In other words, the velocity-only observer will sometimes make errors that fall on (or are close to) the opposite motion direction.

The location of the velocity measurement strongly influences the shape of the posterior distribution that is decoded from the velocity and orientation measurements (Eq. 21, 30). Consequently, when

the velocity measurement falls close to the opposite direction of motion, the larger peak of the decoded posterior is likely located close to this direction as well, resulting in a relationship between the two.

Importantly, also the Bayesian model predicts that opposite behavioral responses can be predicted from the decoded posterior. Because both models predict that there should be a relationship between the location of the posterior's larger peak and opposite responses, simply testing for this relationship won't enable us to adjudicate between the models.

We hope this better explains why testing this relationship won't help. We now also discuss this in the text (p. 6, and Figure S20).

Reviewer #3 (Remarks to the Author):

I would like to thank the authors for their responses to my critiques. My concerns were addressed. The additional analyses and clarifications in the revised version have substantially improved the paper.

This work will be a useful contribution. I support the publication of this paper.

Reviewer #4 (Remarks to the Author):

The authors have done an admirable job of responding to the extensive comments from all the reviewers. The revisions have substantially improved the paper. I do not have much more to add except a general comment about how the results might transfer to more complex tasks. Specifically, the Bayesian model with independent signals makes perfect sense for the random-dot motion estimation task, but it will not work easily for real-world motion estimation, where there are irrelevant static orientation signals everywhere. The use of orientation signals for motion perception at a given image location must be gated in some way by the simultaneous encoding of motion at that location. This fact is why a multiplicative interaction was postulated in Geisler (1999). Basically, the probability that a strong orientation response at some location corresponds to some motion direction becomes zero if there is no simultaneous motion response at that location. It might be good to mention in the discussion that a more realistic Bayesian model will have to take this into account. Nonetheless, this does not detract from the value of this study. The fMRI results and the fact that humans show 180-deg errors in motion direction estimation at low signal strength seems to be strong evidence that humans are using the orientation information to estimate motion direction.

We thank the reviewer for this insight. Indeed, for real-world operations it will be important to appropriately bind the relevant signals together. We now briefly discuss this on p. 13.

REVIEWERS' COMMENTS

Reviewer #1 (Remarks to the Author):

I thank the authors for quantitatively demonstrating that the Switching model cannot capture all of their behavioral and fMRI results but that the Bayesian model can.

I understand their arguments as follows: orientation likelihood must be uniform for the Switching model to capture the behavioral responses observed in the fMRI experiment (with the 100% coherence stimulus). However, a Switching model with uniform orientation likelihood predicts that the distributions decoded from the subjects' fMRI Bold responses should be unimodal, which is inconsistent with the observed bimodality of the decoded distributions. Moreover, already flat with the high-noise stimulus, the likelihood should also be flat with greater levels of uncertainty as in the follow-up behavioral study (with 18% coherence stimuli). Such a model cannot produce the bimodal behavioral response distribution observed in that study. The Bayesian model can capture all of these results, thus providing a better explanation.

Reviewer #2 (Remarks to the Author):

I would like to thank the authors for their detailed responses. My concerns were fully addressed, and I support the manuscript to be published in its current form.

We thank all four reviewers for their constructive comments, which have been very valuable to us in further strengthening the manuscript.

Reviewer #1 (Remarks to the Author):

I thank the authors for quantitatively demonstrating that the Switching model cannot capture all of their behavioral and fMRI results but that the Bayesian model can.

I understand their arguments as follows: orientation likelihood must be uniform for the Switching model to capture the behavioral responses observed in the fMRI experiment (with the 100% coherence stimulus). However, a Switching model with uniform orientation likelihood predicts that the distributions decoded from the subjects' fMRI Bold responses should be unimodal, which is inconsistent with the observed bimodality of the decoded distributions. Moreover, already flat with the high-noise stimulus, the likelihood should also be flat with greater levels of uncertainty as in the follow-up behavioral study (with 18% coherence stimuli). Such a model cannot produce the bimodal behavioral response distribution observed in that study. The Bayesian model can capture all of these results, thus providing a better explanation.

That is correct.

Reviewer #2 (Remarks to the Author):

I would like to thank the authors for their detailed responses. My concerns were fully addressed, and I support the manuscript to be published in its current form.